# The Slingshot Effect: A Late-Stage Optimization Anomaly in Adam-Family of Optimization Methods

**Vimal Thilak** *vthilak@apple.com*
*Apple*

**Etai Littwin** *elittwin@apple.com*
*Apple*

**Shuangfei Zhai** *szhai@apple.com*
*Apple*

**Omid Saremi** *osaremi@apple.com*
*Apple*

**Roni Paiss** *paiss.roni@gmail.com*

**Joshua Susskind** *jsusskind@apple.com*
*Apple*

**Reviewed on OpenReview:** *https://openreview.net/forum?id=OZbn8ULouY*

## Abstract

Adam (Kingma & Ba, 2014) and Adam-family (Loshchilov & Hutter, 2017; Tieleman & Hinton, 2012) of adaptive gradient methods have become indispensable for optimizing neural networks, particularly in conjunction with Transformers (Vaswani et al., 2017; Dosovitskiy et al., 2020). In this paper, we present a novel optimization anomaly called the *Slingshot Effect*, which manifests during extremely late stages of training. We identify a distinctive characteristic of this phenomenon through cyclic phase transitions between stable and unstable training regimes, as evidenced by the cyclic behavior of the norm of the last layer's weights. Although the Slingshot Effect can be easily reproduced in more general settings, it does not align with any known optimization theories, emphasizing the need for in-depth examination. Moreover, we make a noteworthy observation that Grokking, as reported by Power et al. (2021), occurs predominantly during the onset of the Slingshot Effects and is absent without it, even in the absence of explicit regularization. This finding suggests a surprising inductive bias of adaptive gradient optimizers at late training stages, urging a revised theoretical analysis of their origin. Our study sheds light on an intriguing optimization behavior that has significant implications for understanding the inner workings of Adam-family of gradient methods.

## 1 Introduction

Adaptive optimizers (Kingma & Ba, 2014; Loshchilov & Hutter, 2017; Tieleman & Hinton, 2012) are widely used to train deep neural networks including ResNets (Wightman et al., 2021; Dosovitskiy et al., 2020) and Transformers (Vaswani et al., 2017; Dosovitskiy et al., 2020) as they empirically demonstrate strong

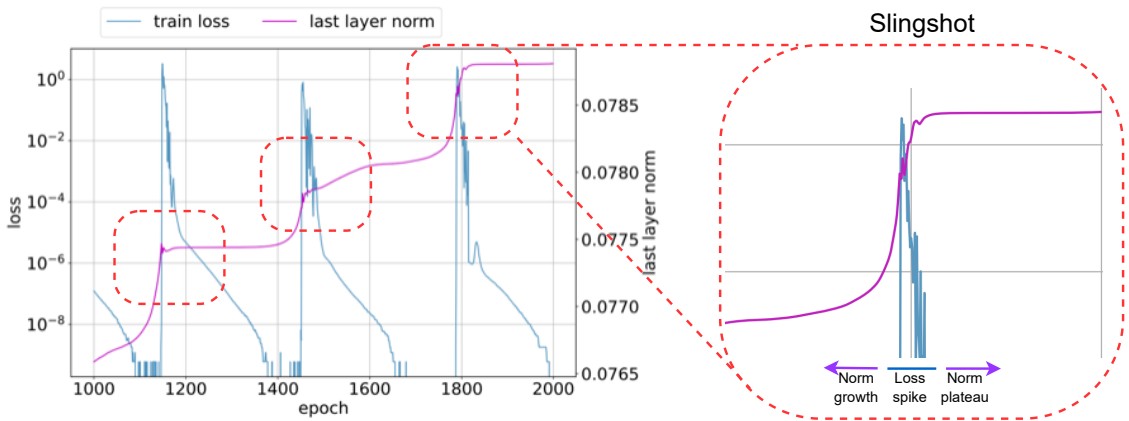

Figure 1: Slingshot Effects observed with a fully-connected ReLU network (FCN). The FCN is trained with 200 randomly chosen CIFAR-10 samples with Adam. Multiple Slingshot Effects occur in a cyclic fashion as indicated by the dotted red boxes. Each Slingshot Effect is characterized by a period of rapid growth of the last layer weights, an ensuing training loss spike, and a period of curtailed norm growth.

performance. Recently, several works (Chowdhery et al., 2022; Molybog et al., 2023; Wortsman et al., 2023; Courtois et al., 2023; Zhu et al., 2023) have reported optimization instability of Adam or AdamW (Kingma & Ba, 2014; Loshchilov & Hutter, 2017) when used to train deep neural networks. Specifically, the training instabilities are characterized by several spikes in the loss during optimization. This curious behavior has motivated studies that attempt to understand and explain the source of these instabilities (Molybog et al., 2023) or provide mitigation strategies to stabilize training (Chowdhery et al., 2022; Wortsman et al., 2023).

In this paper, we present an optimization anomaly that we call the *Slingshot Effect* wherein the training loss of neural networks optimized with Adam-family of adaptive gradient methods (see Algorithm 1 for a generic description of methods used in this work) exhibit loss spikes. Figure 1 shows an example of the training loss spikes observed during neural network optimization. As can be seen from Figure 1, the instabilities are observed at extremely late stages of training or in the so-called *Terminal Phase of Training* (TPT) . In detail, leveraging the basic setup in Power et al. (2021) with Adam (Kingma & Ba, 2014) and without weight decay, we make the following additional observations visualized in Figure 2:

1. During the TPT, training exhibits a cyclic behaviour between stable and unstable regimes. A prominent artifact of this behaviour can be seen in the norm of the classification layer's weights, which exhibits a cyclical behavior with distinct, sharp phase transitions that alternate between rapid growth and growth curtailment over the course of training.

2. The norm grows rapidly sometime after the model has perfect classification accuracy on training data. A sharp phase transition then occurs in which the model misclassifies training samples. This phase change is accompanied by a sudden spike in training loss, and a deceleration in the norm growth of the final classification layer.

3. The features (pre-classification layer) show rapid evolution as the weight norm transitions from rapid growth phase to growth curtailment phase, and change relatively little at the norm growth phase.

4. Phase transitions between norm growth and norm curtailment phases are typically accompanied by a sudden bump in generalization as measured by classification accuracy on a validation set, as observed in a dramatic fashion in Power et al. (2021).

5. It is empirically observed with the setup described above (Adam with no weight decay) that grokking as reported in Power et al. (2021) almost exclusively happens at the onset of *Slingshots*, and is absent without it.

We denote the observations above as the *Slingshot Effect* (or simply slingshot), which is defined to be the full cycle starting from the rapid norm growth phase, and ending in the curtailed norm growth phase. Empirically, a single training run typically exhibits multiple Slingshots. Moreover, while *grokking* as described in Power et al. (2021) might be data dependent, we find that the Slingshot Effect is pervasive, and can be easily reproduced in multiple scenarios, encompassing a variety of models (Transformers and MLPs) and datasets (both vision, algorithmic and synthetic datasets). Since we only observe Slingshots when training classification models with adaptive optimization methods such as Adam Kingma & Ba (2014) and RMSProp (Tieleman & Hinton, 2012), our work can be seen as empirically characterizing an implicit bias of such optimizers. Finally, we focus on Adam (Kingma & Ba, 2014) optimization method in the main paper, and relegate all experiments with additional optimizers to the appendix that suggest that our observations and conclusions hold for other methods under the Adam-family of optimizers [1]. We note here that we limit our empirical analysis of Slingshot Effects with Adam (Kingma & Ba, 2014), RMSProp (Tieleman & Hinton, 2012) and Adagrad (Duchi et al., 2011) while acknowledging that there are a vast number of other adaptive optimization methods including AMSGrad (Reddi et al., 2018), Adax (Li et al., 2020), AdaBound (Luo et al., 2019) and AdaBelief (Zhuang et al., 2020) that are not used in our analysis.

---

**Algorithm 1** Adam-family of Adaptive Gradient Methods

---

**Input:** $x_1 \in \mathcal{F}$ initial parameters,
**Input:** step size $\mu$,
**Input:** $\beta_1$ and $\beta_2 \in [0, 1)$,
**Input:** $\epsilon \in \mathbb{R}^+$,
**Output:** Optimized parameters $x_{T+1}$.

1  Initialize $m_0$ and $V_0$ to $\mathbf{0}$
2  **for** $t = 1..., T$ **do**
3       $g_t = \nabla f_t(x_t)$. (Get gradients $g$ w.r.t function $f$)
4       $m_t = \beta_1 m_{t-1} + (1 - \beta_1)g_t$
5       $m_t = \frac{m_t}{1 - \beta_1^t}$
6       $V_t = \beta_2 V_{t-1} + (1 - \beta_2)g_t^2$
7       $V_t = \frac{V_t}{1 - \beta_2^t}$
8       $x_{t+1} = x_t - \frac{\mu m_t}{\sqrt{V_t} + \epsilon}$ (Update parameters)

---

## 1.1 Implications of Our Findings

The findings in this paper have both theoretical and practical implications that go beyond characterizing the effect. A prominent feature of the Slingshot Effect is the repeating phase shifts between stable and unstable training regimes, where the unstable phase is characterized by extremely large gradients, and spiking training loss. Furthermore, we find that learning at late stages of training have a cyclic property, where non trivial feature adaptation only takes place at the onset of a phase shift. This abrupt feature learning characteristic is observed in a dramatic fashion when considering the grokking phenomena, where we observe the slingshot effect at the onset of grokking, pointing towards possible generalization benefits of the effect. From a theoretical perspective, this is contradictory to common assumptions made in the literature of convergence of adaptive optimizers, which typically require $L$ smooth cost functions, and bounded stochastic gradients, either in the $L_2$ or $L_\infty$ norm, decreasing step sizes and stable convergence (Zhang et al., 2020; Allen-Zhu et al., 2019; Barakat & Bianchi, 2021). The generalization behavior observed with the grokking setup (Power et al., 2021) casts doubt on the ability of current working theories to explain the Slingshot Effect.

Practically, our work presents additional evidence for the growing body of work indicating the importance of the TPT stage of training for optimal performance (Hoffer et al., 2017; Power et al., 2021; Papyan et al., 2020). In an era where the sheer size of models are quickly becoming out of reach for most practitioners, our work suggests focusing on improved methods to prevent excessive norm growth either implicitly through Slingshot Effects or through other forms of explicit regularization or normalization which we study in the appendix.

---

[1]We use the term "Adam-family" to refer to those methods whose update equations are defined in Kingma & Ba (2014)

## 2    Related Work

The Slingshot Effect we uncover here is reminiscent of the *catapult mechanism* described in Lewkowycz et al. (2020). Lewkowycz et al. (2020) show that loss of a model trained via gradient descent with an appropriately large learning rate shows a non-monotonic behavior —the loss initially increases and starts decreasing once the model "catapults" to a region of lower curvature —early in training. However, the catapult phenomenon differs from Slingshot Effects in several key aspects. The *catapult mechanism* is observed with vanilla or stochastic gradient descent unlike the Slingshot Effect that is seen with adaptive optimizers including Adam (Kingma & Ba, 2014) and RMSProp (Tieleman & Hinton, 2012). Furthermore, the *catapult phenomenon* relates to a large initial learning rate, and does not exhibit a repeating cyclic behavior. More intriguingly, Slingshot Effects only emerge late in training, typically long after the model reaches perfect accuracy on the training data.

Cohen et al. (2021) describe a "progressive sharpening" phenomenon in which the maximum eigenvalue of the loss Hessian increases and reaches a value that is at equal to or slightly larger than $2/\eta$ where $\eta$ is the learning rate. This "progressive sharpening" phenomenon leads the model to enter a regime Cohen et al. (2021) call *Edge of Stability* (EoS) where-in the model shows non-monotonic training loss behavior over short time spans. In a recent work, Cohen et al. (2022) demonstrate a similar phenomenon with full-batch (or with large batch empirically) adaptive optimizers called *Adaptive Edge of Stability* (AEoS) where-in the "raw" sharpness quantified by the maximum eigenvalue of the loss Hessian increases and oscillates around a certain value for a version of Adam Cohen et al. (2021) call "Frozen Adam". However, Cohen et al. (2022) observe that the 'raw' sharpness continues to increase during training with regular Adam (Kingma & Ba, 2014). Edge of Stability and Adaptive Edge of Stability are similar to Slingshot Effects in that the instablities are observed later on in training. However, while EoS Cohen et al. (2021) and AEoS (Cohen et al., 2022) aim to shed light on neural network training dynamics they do not show or analyze the cyclic behavior of the last layer norm weights and do not examine generalization that we observe in our experiments.

As noted above, the Slingshot Effect emerges late in training, typically longer after the model reaches perfect accuracy and has low loss on training data. The benefits of continuing to training a model in this regime has been theoretically studied in several works including (Soudry et al., 2018; Lyu & Li, 2019). Soudry et al. (2018) show that training a linear model on separable data with gradient using the logistic loss function leads to a max-margin solution. Furthermore, Soudry et al. (2018) prove that the loss decreases at a rate of $O(\frac{1}{t})$ while the margin increases much slower $O(\frac{1}{\log t})$, where $t$ is the number of training steps. Soudry et al. (2018) also note that the weight norm of the predictor layer increases at a logarithmic rate, i.e., $O(\log(t))$. Lyu & Li (2019) generalize the above results to homogeneous neural networks trained with exponential-type loss function and show that loss decreases at a rate of $O(1/t(\log(t))^{2-2/L})$ where $L$ is defined as the order of the homogenous neural network. Although these results indeed prove the benefits of training models, their analyses are limited to gradient descent. Moreover, the analyses developed by Soudry et al. (2018) do not predict any phenomenon that resembles the Slingshot Effect. Wang et al. (2021) show that homogenous neural networks trained with RMSProp (Tieleman & Hinton, 2012) or Adam (Kingma & Ba, 2014) without momentum converge in direction to the max-margin solution. However, none of these papers can explain the Slingshot Effect and specifically the cyclical behavior of the norm of the last layer weights.

**Understanding Grokking**    The grokking phenomenon (Power et al., 2021) has been studied in several recent works that aim to understand and explain the empirical findings by Power et al. (2021). Liu et al. (2022) use a toy model to study grokking and show four phases of learning and that feature learning occurs at a slower rate that leads to delayed generalization. In a follow-up work, Liu et al. (2023) analyze the train and test loss landscapes and suggest that grokking occurs due to mismatch between the train and test loss landscapes. Nanda et al. (2023) analyze the modular addition example proposed by Power et al. (2021) and identify three phases of learning and propose a "hidden measure" that track the development of structure in the weights of a model. Barak et al. (2022) analyzes the behavior of neural networks on subset parity task and show that a grokking-like phenomenon can be seen in that setup even with multi-layer perceptrons (MLPs). We empirically analyze grokking (Power et al., 2021; Barak et al., 2022) in a setup that does not include any explicit regualarization like weight decay unlike the other works that use weight decay.

**Relationship to Training Instability or Loss Spikes** Several papers (Chowdhery et al., 2022; Molybog et al., 2023; Wortsman et al., 2023; Zhu et al., 2023; Lobacheva et al., 2021) and references therein have reported optimization instability of Adam (Kingma & Ba, 2014) or AdamW (Loshchilov & Hutter, 2017) that is characterized by spikes in the loss during training. Molybog et al. (2023) theoretically analyze the unstable behavior observed while training large scale models with Adam (Kingma & Ba, 2014) and propose several mitigation approaches to stabilize training. Wortsman et al. (2023) also report loss spikes while training large scale vision models and observe that the loss spikes occur due to either an out-of-date second order gradient moment estimate or due to lower precision floating point (16-bit) training. Wortsman et al. (2023) provide mitigation strategies based on these observations. We note here that most literature related to training instability is focused on stabilizing optimization with the notable exception of Zhu et al. (2023) who provide empirical evidence that training instabilities may lead to better generalization. While Zhu et al. (2023) use stochastic gradient descent (SGD) in their work, we empirically analyze the optimization and generalization behavior of neural networks that exhibit training loss spikes with adaptive optimizers.

## 3 The Slingshot Effect

### 3.1 Experimental Setup

We use the training setup studied by Power et al. (2021) in the main paper as a working example to illustrate the Slingshot Effect. In this setup, we train decoder-only Transformers (Vaswani et al., 2017) on a modular division dataset (Power et al., 2021) of the form $a \div b = c$, where $a$, $b$ and $c$ are discrete symbols and $\div$ refers to division modulo $p$ for some prime number $p$, split into training and validation sets. The task consists of calculating $c$ given $a$ and $b$. The algorithmic operations and details of the datasets considered in our experiments are described in Appendix B. The Transformer consists of 2 layers, of width 128 and 4 attention heads with approximately 450K trainable parameters and is optimized by Adam (Kingma & Ba, 2014; Loshchilov & Hutter, 2017). For these experiments we set learning rate to 0.001, weight decay to 0, $\beta_1 = 0.9$, $\beta_2 = 0.98$, $\epsilon = 10^{-8}$, linear learning rate warmup for the first 10 steps and minibatch size to 512 which are in line with the hyperparameters considered in (Power et al., 2021).

### 3.2 Experimental Observations

Figure 2 shows the metrics of interest that we record on training and validation samples for modular division dataset. Specifically, we measure 1) *train loss*; 2) *train accuracy*; 3) *validation loss*; 4) *validation accuracy*; 5) *last layer norm*: denoting the norm of the classification layer's weights and 6) *feature change*: the relative change of features of the l-th layer ($h^l$) after the t-th gradient update step $\frac{\|h_{t+1}^l - h_t^l\|}{\|h_t^l\|}$. We observe from Figure 2b that the model is able to reach high training accuracy around step 300 while validation accuracy starts improving after $10^5$ steps as seen in Figure 2d. Power et al. (2021) originally showed this phenomenon and refer to it as grokking. We observe that while the validation accuracy does not exhibit any change until much later in training, the validation loss shown in Figure 2c exhibits a double descent behavior with an initial decrease, then a growth before rapidly decreasing to zero.

Seemingly, some of these observations can be explained by the arguments in (Soudry et al., 2018) and their extensions to adaptive optimizers (Wang et al., 2021). Namely, at the point of reaching perfect classification of the training set, the cross-entropy (CE) loss by design pressures the classification layer to grow in norm at relatively fast rate. Simultaneously, the implicit bias of the optimizer coupled with the CE loss, pushes the direction of the classification layer to coincide with that of the maximum margin classifier, albeit at a much slower rate.

These insights motivate us to measure the classifier's last layer norm during training. We observe in Figure 2a that once classification reaches perfect accuracy on the training set, the classification layer norm exhibits a distinct cyclic behavior, alternating between rapid growth and growth curtailment, with a sharp phase transition between phases. Simultaneously, the training loss retains a low value in periods of rapid norm growth, and then wildly fluctuates in time periods when there is a curtailment of norm growth. Figure 2e and Figure 2f shows the evolution of the relative change in features output by each layer in the Transformer. We observe that the feature maps are not updated much during the norm growth phase. However, at the

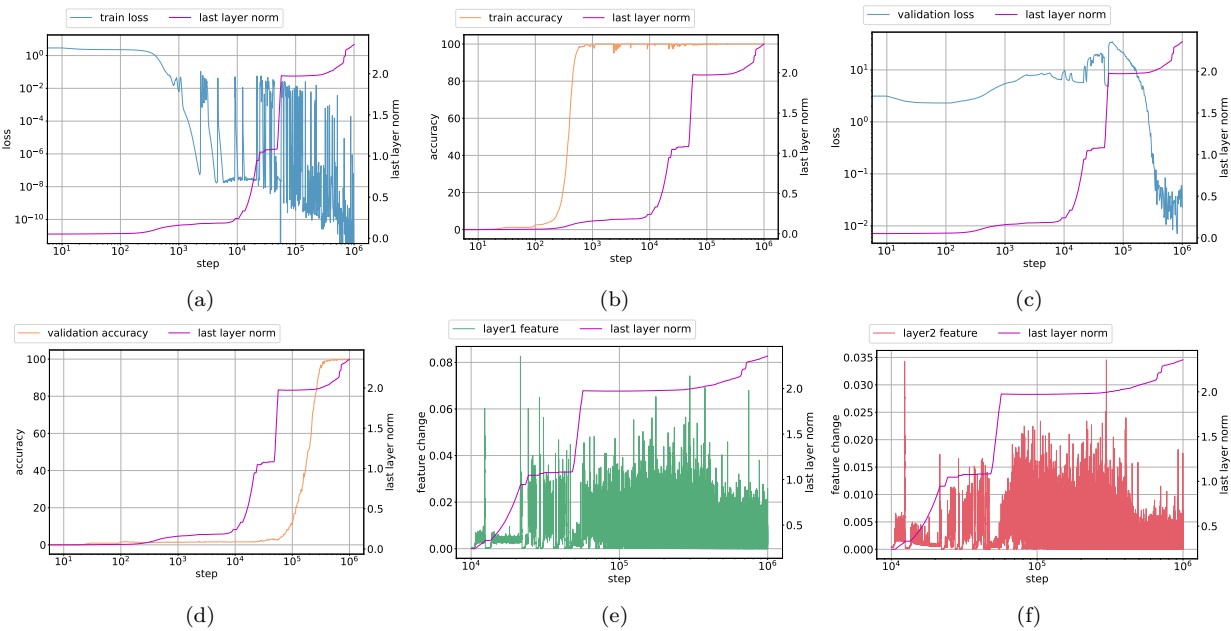

Figure 2: Division dataset: Last layer weight norm growth versus a) loss on training data b) accuracy on training data (c) loss on validation data d) accuracy on validation data e) normalized relative change in features of first Transformer layer (f) normalized relative change in features of second Transformer layer. Note that the feature change plots are shown starting at 10K step to emphasize the feature change behavior during rapid norm growth and norm growth curtailment phases, revealing that the features stop changing during the norm growth phase and resume changing during the curtailed norm growth phase.

phase transition, we observe that the feature maps receive a rapid update, which suggests that the internal representation of the model is updating.

**Are Slingshot Effects a general phenomenon?** In an attempt to ascertain the generality of Slingshot Effects as an optimization artifact, we run similar experiments with additional architectures, datasets, optimizers, and hyperparameters. We use all algorithmic datasets as proposed in (Power et al., 2021), as well as frequently used vision benchmarks such as CIFAR-10 (Krizhevsky, 2009), and even synthetic Gaussian dataset. For architectures, we use Transformers, MLPs and deep linear models (see figure 1). We find abundant evidence of Slingshot Effects in all of our experiments with Adam and RMSProp. We are unable to observe Slingshot Effects with Adagrad (Duchi et al., 2011) and also with stochastic gradient descent (SGD) or SGD with momentum, pointing to the generality of the effect across architectures and datasets. We refer the reader to Appendix A for the full, detailed description of the experiments.

**Why do Slingshot Effects happen?** We hypothesize that the norm growth continues until the curvature of the loss surface becomes large, effectively "flinging" the weights to a different region in parameter space as small gradient directions get amplified, reminiscent of the mechanics of a slingshot flinging a projectile. We attempt to quantify how far a model is flung by measuring the cosine distance between a checkpoint during optimization and initial parameters. Specifically, we divide the model parameters into representation (pre-classifier) parameters and classifier (last layer) parameters and calculate how far these parameters have moved from initialization. We show that checkpoints collected after a model experiences Slingshot have a larger representation cosine distance. We defer the reader to the appendix for further details.

By design, adaptive optimizers adapt the learning rate on a per parameter basis. In toy, convex scenarios, the $\epsilon$ parameter provably determines whether the algorithm will converge stably. To illustrate this, we take inspiration from Cohen et al. (2021), and consider a quadratic cost function $\mathcal{L}(A, B, C) = \frac{1}{2}x^\top A x + B^\top x + C, A \in \mathcal{R}^{d \times d}, x, B \in \mathcal{R}^d, C \in \mathcal{R}$, where we assume $A$ is symmetric and positive definite. Note that the global minimum of this cost is given by $x^\star = -A^{-1}B$. The gradient of this cost with respect to $x$ is

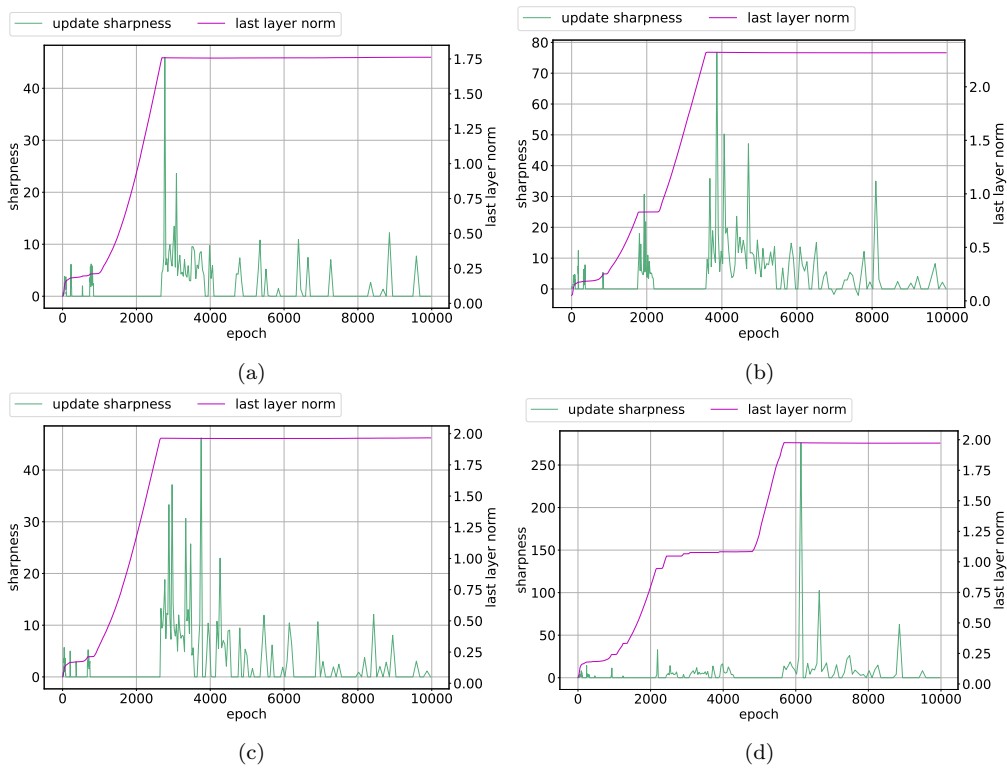

Figure 3: Curvature metric (denoted as "update sharpness") evolution vs norm growth on (a) addition, (b) subtraction, (c) multiplication, and (d) division dataset. Note the spike in the sharpness metric near the phase transitions between rapid and curtailed norm growth.

given by $g = Ax + B$. Consider optimizing the cost with adaptive optimization steps of the simple form $x_{t+1} = x_t - \mu \frac{g}{|g|+\epsilon} = x_t - \mu \frac{Ax_t+B}{|Ax_t+B|+\epsilon}$ where $\mu$ is a learning rate, and the division and absolute operations are taken element wise. Starting from some $x_0$, the error $e_t = x_t - x^\star$ evolves according to:

$$e_{t+1} = \left(I - \mu\,\mathrm{diag}(\frac{1}{|Ae_t|+\epsilon})A\right)e_t \overset{\mathrm{def}}{=} \mathcal{M}_t e_t \tag{1}$$

Note that the condition $\|A\|_s < \frac{2\epsilon}{\mu}$ where $\| \cdot \|_s$ denotes the spectral norm, implies that the mapping $\mathcal{M}_t$ is a contraction for all values of $t$, and hence convergence to the global optimum is guaranteed (This is in contrast to gradient descent, where the requirement is $\|A\|_s < \frac{2}{\mu}$). Note that the choice of $\epsilon$ crucially controls the requirement on the curvature of the cost, represented by the the spectrum of $A$ in this case. In other words, the smaller $\epsilon$, the more restrictive the requirements on the top eigenvalue of $A$. Cohen et al. (2021) observed that full batch gradient descent increases the spectral norm of the Hessian to its maximum allowed value. We therefore hypothesize that for deep networks, a small value for $\epsilon$ requires convergence to a low curvature local minimum, causing a Slingshot Effect when this does not occur. Moreover, we may reasonably predict that increasing the value of $\epsilon$ would lift the restriction on the curvature, and with it evidence of Slingshot Effects.

Figure 3 shows evidence consistent with the hypothesis that Slingshot Effects occur in the vicinity of high loss curvature, by measuring the local loss surface curvature along the optimization trajectory. Let $\mathcal{H}_t$ denote the local Hessian matrix of the loss, and $u_t$ the parameter update at time $t$ given the optimization algorithm of choice. We use the local curvature along the trajectory of the optimizer, given by $\frac{1}{\|u_t\|^2}u_t^\top \mathcal{H}_t u_t$, as a curvature measure. Across the arithmetic datasets from (Power et al., 2021), whenever the last layer weight norm growth is curtailed, the curvature measure momentarily peaks and settles back down.

**Varying** $\epsilon$   We next observe from Figure 2a that the training loss value also spikes up around the time step when the weight norm change transitions from rapid growth to curtailed growth phase. A low training loss

value suggests that the gradients (and their moments) used as inputs to the optimizer are small, which in turn can cause the $\epsilon$ hyperparameter value to play a role in calculating updates. Our hypothesis here is that the Slingshot Effect should eventually disappear with a sufficiently large $\epsilon$. To confirm this hypothesis, we run an experiment where we vary $\epsilon$ while retaining the rest of the setup described in the previous section.

Figure 4 shows the results for various values of $\epsilon$ considered in this experiment. We first observe that the number of Slingshot Effect cycles is higher for smaller values of $\epsilon$. Secondly, smaller values of $\epsilon$ cause grokking to appear at an earlier time step when compared to larger values. More intriguingly, models that show signs of grokking also experience Slingshot Effects while models that do not experience Slingshot Effects do not show any signs of grokking. Lastly, the model trained with the largest $\epsilon = 10^{-5}$ shows no sign of generalization even after receiving 500K updates.

### 3.3 Effects on Generalization

In order to understand the relationship between Slingshot Effects and neural networks generalization, we experiment with various models and datasets. We observe that models that exhibit Slingshot tend to generalize better, which suggests the benefit of training models for a long time with Adam (Kingma & Ba, 2014). More surprisingly, we observe that Slingshots and grokking tend to come in tandem.

**Transformers with algorithmic datasets** We follow the setting studied by Power et al. (2021) and generate several datasets that represent algorithmic operations and consider several training and validation splits. This dataset creation approach is consistent with the methodology used to demonstrate grokking (Power et al., 2021). The Transformer is trained with Adam (Kingma & Ba, 2014; Loshchilov & Hutter, 2017) with a learning rate of 0.001, weight decay set to 0, and with learning rate warmup for 500K steps. We consider $\epsilon$ of Adam as a hyperparameter in this experiment. Figure 5 summarizes the results for this experiment where the x-axis indicates the algorithmic operation followed by the training data split size. As can be seen in Figure 5, Slingshot Effects are seen with lower values of $\epsilon$ and disappear with higher values of $\epsilon$ which confirms the observations made in Section 3 with modular division dataset. In addition, models that exhibit Slingshot Effects and grokking (shown in green) tend to generalize better than models that do not experience Slingshot Effects and grokking (shown in red).

**ViT with CIFAR-10** For further validation of Slingshot Effects and generalization, we train a Vision Transformer (ViT) (Dosovitskiy et al., 2020) on CIFAR-10 (Krizhevsky, 2009). The ViT consists of 12 layers, width 384 and 12 attention heads trained on fixed subsets of CIFAR-10 dataset (Krizhevsky, 2009). The ViT model described above is trained with 10K, 20K, 30K, 40K and 50K (full dataset) training samples. We train the models with the following learning rates: 0.0001, 0.00031 and 0.001 and with a linear learning rate warmup for the 1 epoch of optimization. We consider multiple learning rates to study the impact of this hyperparameter on Slingshot taking inspiration from Power et al. (2021) where the authors report observing grokking over a narrow range of learning rates . Figure 6 shows a plot of the highest test accuracy for a set of hyperparameters (learning rate, number of training samples) as a function of the number of training samples from which we make the following observations. The best test accuracy for a given set of hyperparameters is typically achieved after Slingshot phase begins during optimization. The checkpoints that achieve the highest test accuracy are labeled as "post-slingshot" and shown in green in Figure 6. While post-Slingshot checkpoints seem to enjoy higher test accuracy, there are certain combinations of hyperparameters that lead to models that show better test accuracy prior to the start of the first Slingshot phase. We label these points as "pre-slingshot" (shown in blue) in Figure 6. The above observations appear to be consistent with our finding that training long periods of time may lead to better generalization seen with grokking datasets (Power et al., 2021).

**Non-Transformer models with subset parity** We train MLPs with Adam on the $(n, k)$ subset parity task. This family of tasks is notoriously challenging since it poses strict computational lower bounds on learning (see (Barak et al., 2022) for more details). We refer the reader to Appendix A.6 for a full description of the dataset and the model used in this setup. We observe multiple Slingshot Effects in Figure 13. Additionally, we observe an improvement in the test accuracy around the vicinity of each Slingshot effect until the loss stops changing deep into optimization. We empirically study the dynamics of the effective step size and

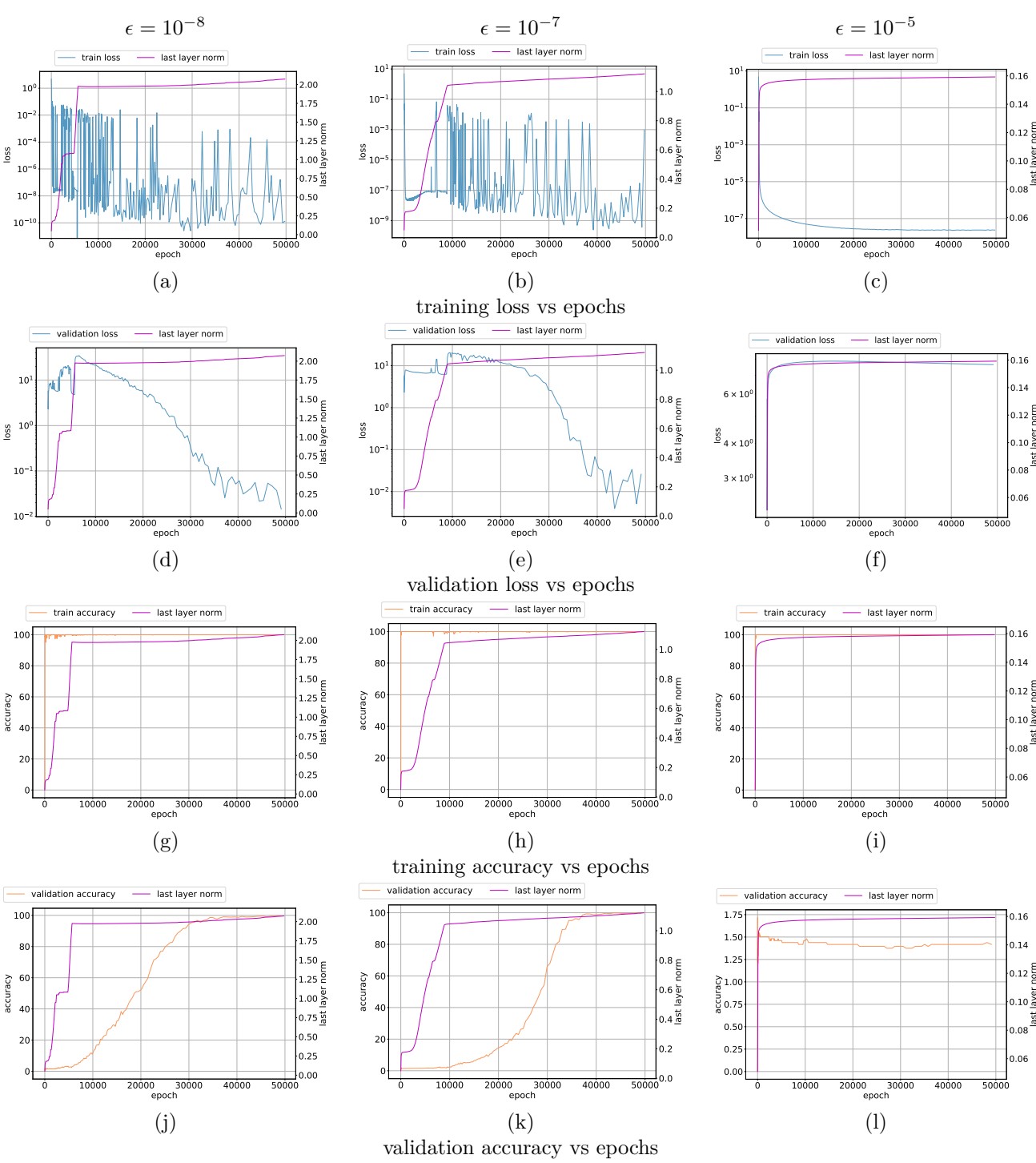

Figure 4: Varying $\epsilon$ in Adam on the Division dataset. Observe that as $\epsilon$ increases, there is no Slingshot Effect or grokking behavior. Figure (a) corresponds to default $\epsilon$ suggested in (Kingma & Ba, 2014) where the model trained with smallest value undergoes multiple Slingshot cycles.

curvature in Appendix A.6.1 to understand what occurs during a phase transition from an optimization perspective.

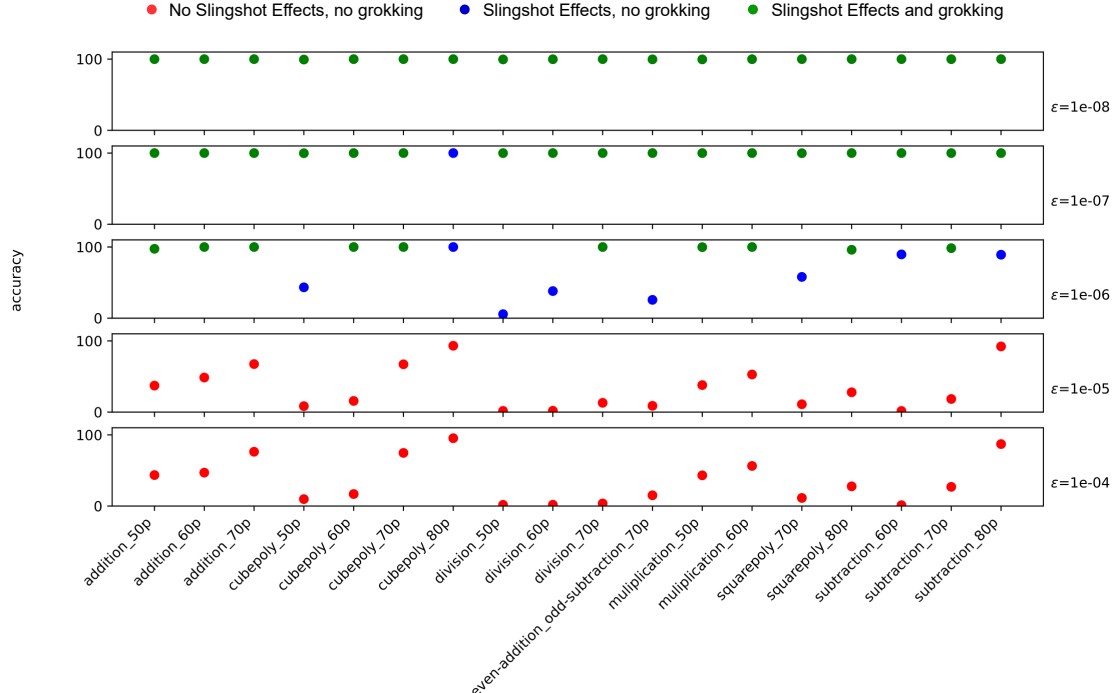

Figure 5: Extended analysis on multiple grokking datasets. Points shown in green represent both Slingshot Effects and grokking, points shown blue indicate Slingshot Effects but not grokking while points in red indicate no Slingshot Effects and no grokking. $\epsilon$ in Adam is varied as shown in text. Observe that as $\epsilon$ increases, there are no Slingshot Effects or grokking behavior.

**Non-Transformer Models with synthetic data** We conduct experiments with MLPs on synthetic data where the synthetic data is a low dimensional embedding projected to higher dimensions via random projections. With this dataset, we show that generalization occurs late in training with Adam. Specifically, we tune $\epsilon$ in Adam and show that the optimizer is highly sensitive to this hyperparameter. These observations are consistent with the behavior reported above with Transformers and on algorithmic datasets as well as standard vision benchmark such as CIFAR-10. We refer the reader to Appendix A.9 for complete description and details of these experiments.

### 3.4 Drawbacks and Limitations

While the Slingshot Effect exposes an interesting implicit bias of Adam that often promotes generalization, due to its arresting of the norm growth and ensuing feature learning, it also leads to some training instability and prolonged training time. In the Appendix we show that it is possible to achieve similar levels of generalization with Adam on the modular division dataset (Power et al., 2021) using the same Transformer setup as above, while maintaining stable learning, in regimes that do not show a clear Slingshot Effect. First we employ weight decay, which causes the training loss values to converge to a higher value than the unregularized model. In this regime the model does not become unstable, but instead regularization leads to comparable generalization, and much more quickly. However, it is important to tune the regularization strength appropriately. Similarly, we find that it is possible to normalize the features and weights using the following scheme to explicitly control norm growth: $w = \frac{w}{\|w\|}, f(x) = \frac{f(x)}{\|f(x)\|}$, where $w$ and $f(x)$ are the weights and inputs to the classification layer respectively, the norm used above is the $L_2$ norm, and $x$ is the input to the neural network. This scheme also results in stable training and similar levels of generalization. In all cases the effects rely on keeping the weight norms from growing uncontrollably, which may be the most important factor for improving generalization. These results suggest that while the Slingshot Effect may be an interesting self-correcting scheme for controlling norm growth, there are likely more efficient ways to leverage adaptive optimizers to

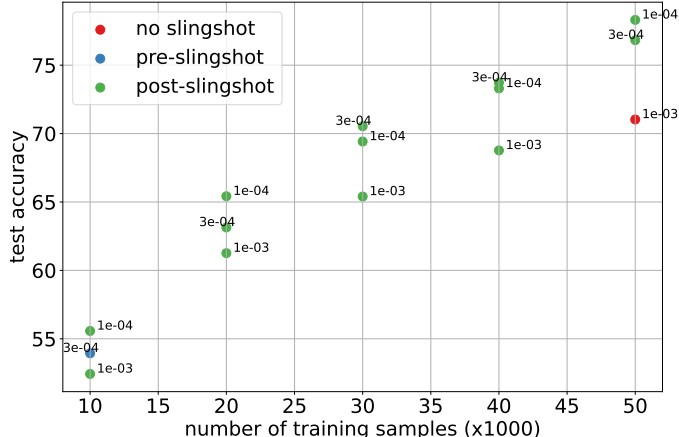

Figure 6: Slingshot Effects on subsets of CIFAR-10 dataset. We train ViTs with multiple learning rates to verify the impact this parameter has on Slingshot. Power et al. (2021) note that grokking occurs over a narrow range of learning rates. Note that the points marked in: (i) green correspond to test accuracy for an experiment after the Slingshot Effect begins, (ii) blue are for trials where best checkpoint is observed prior to start of a Slingshot Effect and (iii) red are for trials with no Slingshot Effect.

similar levels of generalization without requiring the instability that is a hallmark of the Slingshot effect. Finally, we lack a satisfactory theoretical explanation for the Slingshot Effect, and hence removed all attempts at a more rigorous mathematical definition. It is an open problem to formally define and mathematically analyze the mechanism behind Slingshot Effects.

## 4   Conclusion

We have empirically shown that optimizing deep networks with cross entropy loss and adaptive optimizers produces the Slingshot Effect, a curious optimization anomaly unlike anything described in the literature. We have provided ample evidence that Slingshot Effects can be observed with different neural architectures and datasets. Furthermore, we find that Grokking (Power et al., 2021) almost always occurs in the presence of Slingshot Effects and associated regions of instability in the Terminal Phase of Training (TPT). These results in their pure form absent explicit regularization, reveal an intriguing inductive bias of Adam-family of optimizers that becomes salient in the TPT, characterized by cyclic stepwise effects on the optimization trajectory. These effects often promote generalization in ways that differ from non-adaptive optimizers like SGD, and warrant further study to be able to harness efficiently. There are open question remaining to be answered, for instance **1)** What is the causal factor for the weight norm to exit rapid growth and enter a curtailed growth phase? **2)** Are there better ways of promoting generalization without relying on this accidental training instability? Answering these questions will allow us to decouple optimization and regularization, and ultimately to control and improve them independently.

### Acknowledgments

We thank Preetum Nakkiran for insightful feedback and discussions and the Apple Machine Learning Research team for supporting and engaging with this work.

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

# The Slingshot Effect: A Late-Stage Optimization Anomaly in Adam-Family of Optimization Methods - Appendix

## Contents

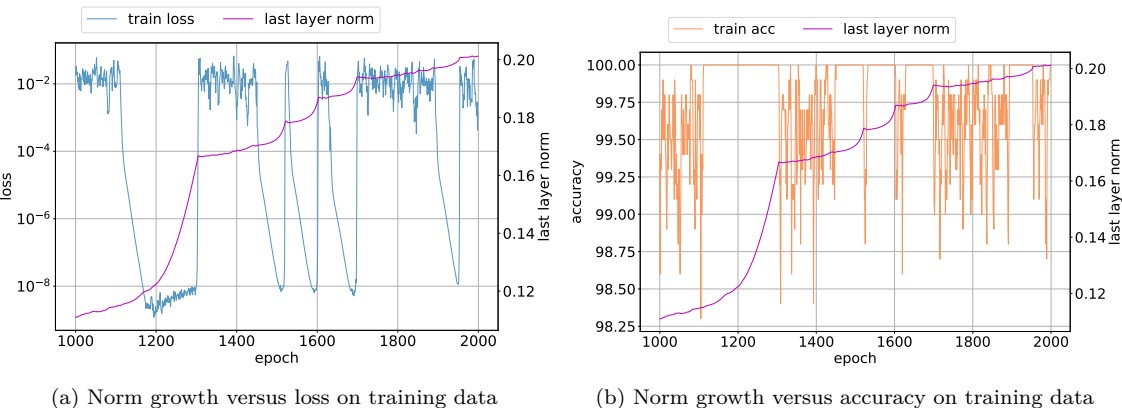

(a) Norm growth versus loss on training data

(b) Norm growth versus accuracy on training data

Figure 7: Vision Transformer trained on 1000 samples from CIFAR-10.

# A  Slingshot Effects across Architectures, Optimizers and Datasets

This section provides further evidence of the prevalence of Slingshot across architectures and optimizers on subsets of CIFAR-10, testing setups beyond the specific setup consider by Power et al. (2021). In these experiments, we focus solely on characterizing the optimization properties of various setups described below. The small sample sizes are used in order to more easily find regimes where different architectures can converge to fit the training data fairly quickly.

We use cross-entropy loss to optimize the models with Adam (Kingma & Ba, 2014) in the following experiments. The following experiments are implemented in PyTorch (Paszke et al., 2019).

## A.1  Vision Transformers on 1000 samples from CIFAR-10

For further validation, we train a Vision Transformer (ViT) (Dosovitskiy et al., 2020) with 12 layers that has 10 million parameters on a small sample of the CIFAR-10 dataset (Krizhevsky, 2009). In this setup, we use a learning rate to 0.001, no weight decay, $\beta_1 = 0.9$, $\beta_2 = 0.95$, $\epsilon = 10^{-8}$ and minibatch size of 128. We choose a sample size of 1000 training samples for computational reasons, as we wish to observe multiple cycles of the Slingshot Effect extremely late in training. The input images are standardized to be in the range $[0, 1]$. No data augmentation is used in our training pipeline. Due to the extremely small sample size, we focus our attention on the training metrics since no generalization is expected. Figure 7a (respectively Figure 7b) shows a plot of training loss (respectively training accuracy) and last layer norm evolution during the latter stages of training. Multiple Slingshot stages are observed in these plots (5 clear cycles), which can be seen by the sharp transition of the weight norm from high growth to a curtailed growth phase.

## A.2  CNN on 200 samples from CIFAR-10

We consider a VGG-like architecture (Simonyan & Zisserman, 2014) that has been adapted for CIFAR-10 dataset.[2] The model is trained with 200 randomly chosen samples from CIFAR-10 training split and with full-batch Adam (Kingma & Ba, 2014; Loshchilov & Hutter, 2017). The hyperparameters used for the optimizer include a learning rate of 0.001, weight decay= 0, $\beta_1 = 0.9$, $\beta_2 = 0.95$, and $\epsilon = 10^{-8}$. As with ViT, no data augmentation is used in these experiments other than standardizing the input to be in the range $[0, 1]$. We observe the presence of multiple Slingshot stages with CNN from Figure 8a and Figure 8b. These experiments suggest that Slingshot effect is not restricted to Transformers architecture alone.

---

[2]We use the VGG11 architecture without batch normalization (Ioffe & Szegedy, 2015) from `https://github.com/kuangliu/pytorch-cifar` in this experiment.

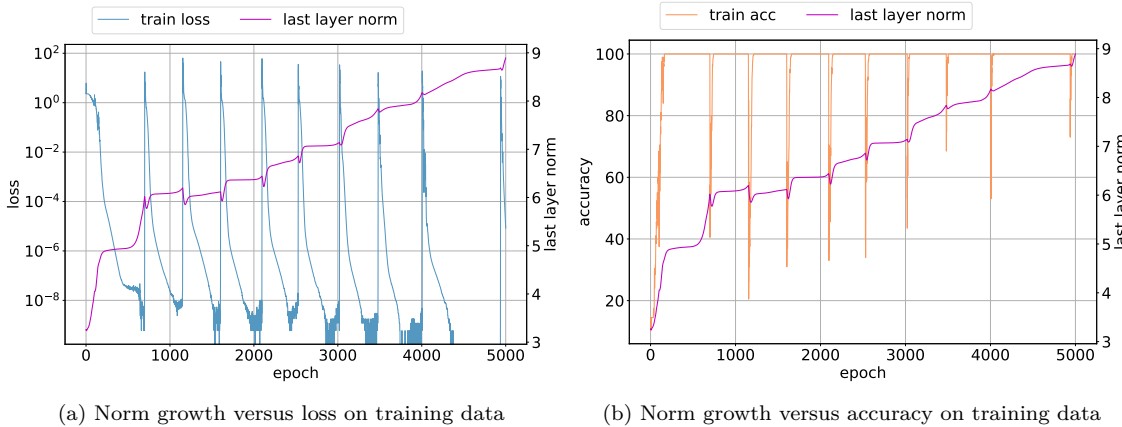

(a) Norm growth versus loss on training data

(b) Norm growth versus accuracy on training data

Figure 8: CNN without batch nomralization trained on 200 samples from CIFAR-10.

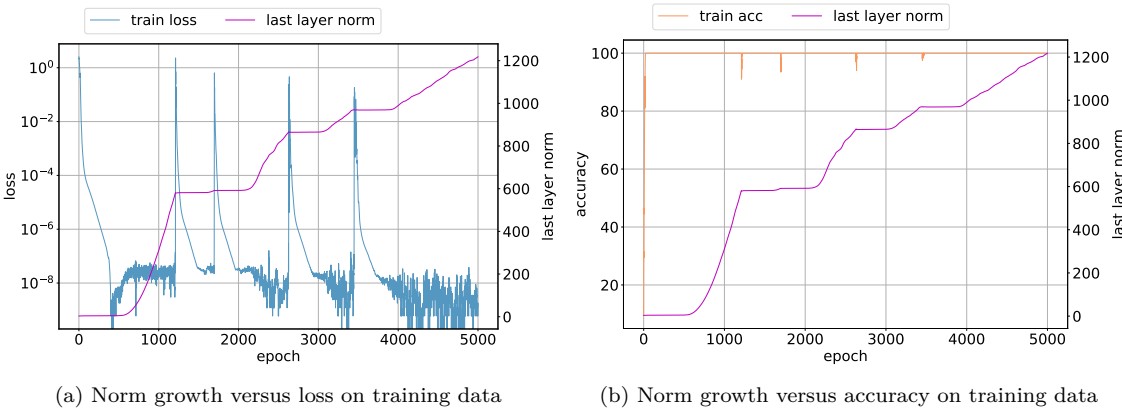

(a) Norm growth versus loss on training data

(b) Norm growth versus accuracy on training data

Figure 9: CNN with batch nomralization trained on 200 samples from CIFAR-10.

**With BatchNorm**  We repeat the CNN-based described above but with a VGG-like model that includes batch normalization (Ioffe & Szegedy, 2015).[3] The training setup is identical to the one described for CNN wihtout batch normalization. We observe the presence of multiple Slingshot stages with CNN from Figure 9a and Figure 9b. The weight norm does not decrease during training as opposed to the weight norm dynamics for CNN without batch normalization seen in Figure 8. These experiments suggest that Slingshot Effects can be seen with standard neural network training components including batch normalization.

### A.3   MLPs on 200 samples from CIFAR-10

The next architecture we consider is a deep (6 layers) fully connected network trained on a small sample of 200 samples belonging to the CIFAR-10 dataset (Krizhevsky, 2009) with full-batch Adam (Kingma & Ba, 2014; Loshchilov & Hutter, 2017) optimizer. The optimizer's hyperparameters are set as following: learning rate = 0.001, weight decay = 0, $\beta_1 = 0.9$, $\beta_2 = 0.95$, and $\epsilon = 10^{-8}$. As with the ViT setup above we do no use data augmentation for training this model. Figure 10a (respectively Figure 10b) shows a plot of training loss (respectively training accuracy) and last layer norm evolution during the latter stages of training. Multiple Slingshot stages are observed in this setup as well. These experiments further suggest that the Slingshot Effect is prevalent in simple models as well.

---

[3]We use the VGG11 architecture with batch normalization (Ioffe & Szegedy, 2015) from `https://github.com/kuangliu/pytorch-cifar` in this experiment.

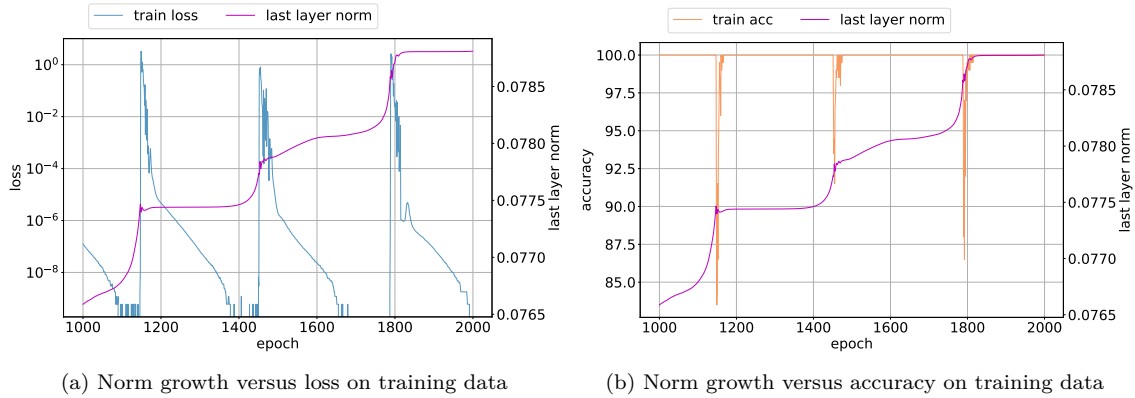

(a) Norm growth versus loss on training data

(b) Norm growth versus accuracy on training data

Figure 10: MLP trained on 200 samples from CIFAR-10.

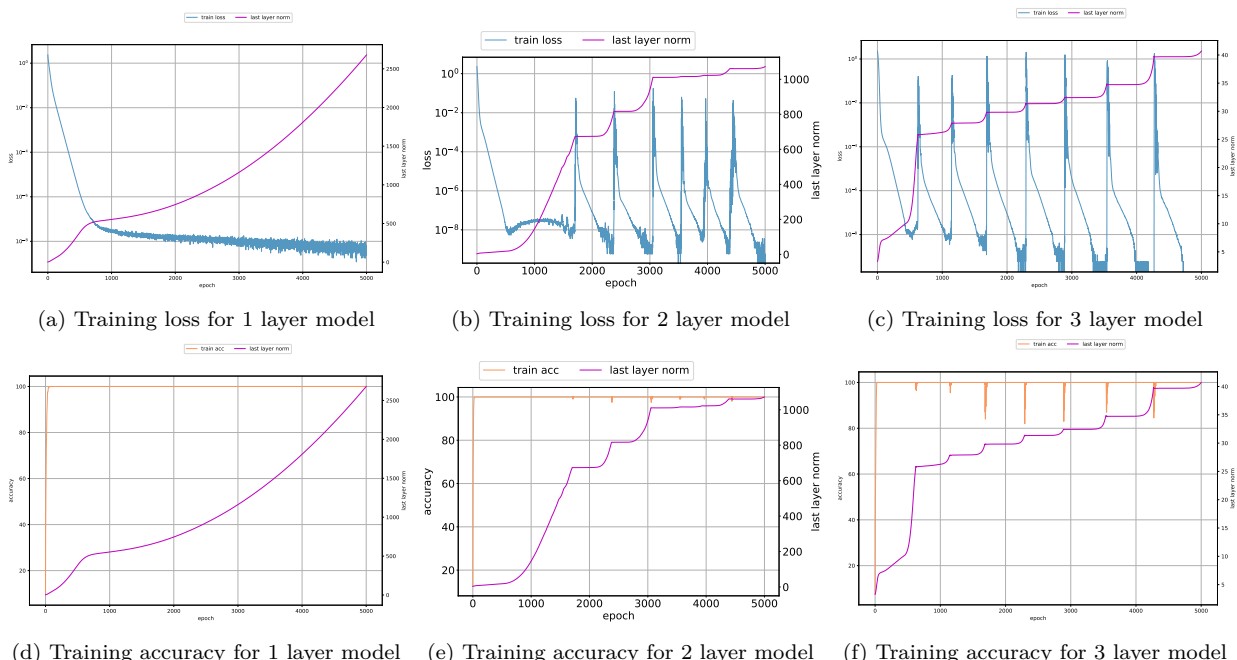

(a) Training loss for 1 layer model

(b) Training loss for 2 layer model

(c) Training loss for 3 layer model

(d) Training accuracy for 1 layer model

(e) Training accuracy for 2 layer model

(f) Training accuracy for 3 layer model

Figure 11: Effect of depth on Slingshots. Norm growth behavior versus training metrics for models whose depth is indicated above. All models are trained with full-batch Adam with learning rate 0.001 on 200 CIFAR-10 samples.

## A.4 Shallow models

We consider the behavior of shallow models including linear, 2- and 3-layer MLPs with Adam optimizer. As with the previous setup, we train these models on a small sample of 200 samples belonging to the CIFAR-10 dataset (Krizhevsky, 2009) with full-batch Adam (Kingma & Ba, 2014) optimizer. The optimizer's hyperparameters are set as following: learning rate = 0.001, weight decay = 0, $\beta_1 = 0.9$, $\beta_2 = 0.95$, and $\epsilon = 10^{-8}$. No data augmentation is used in these experiments as well. Figure 11a, Figure 11b, Figure 11c show the training loss and last layer norm evolution during training for the linear, 2-layer and 3-layer models respectively while Figure 11d, Figure 11e, Figure 11f show the training accuracy and last layer norm evolution. Slingshot Effects are observed in 2-layer and 3-layer MLPs whereas no Slingshot Effects are seen with the linear model. These experiments suggest that depth appears to be a necessary condition to observe Slingshots.

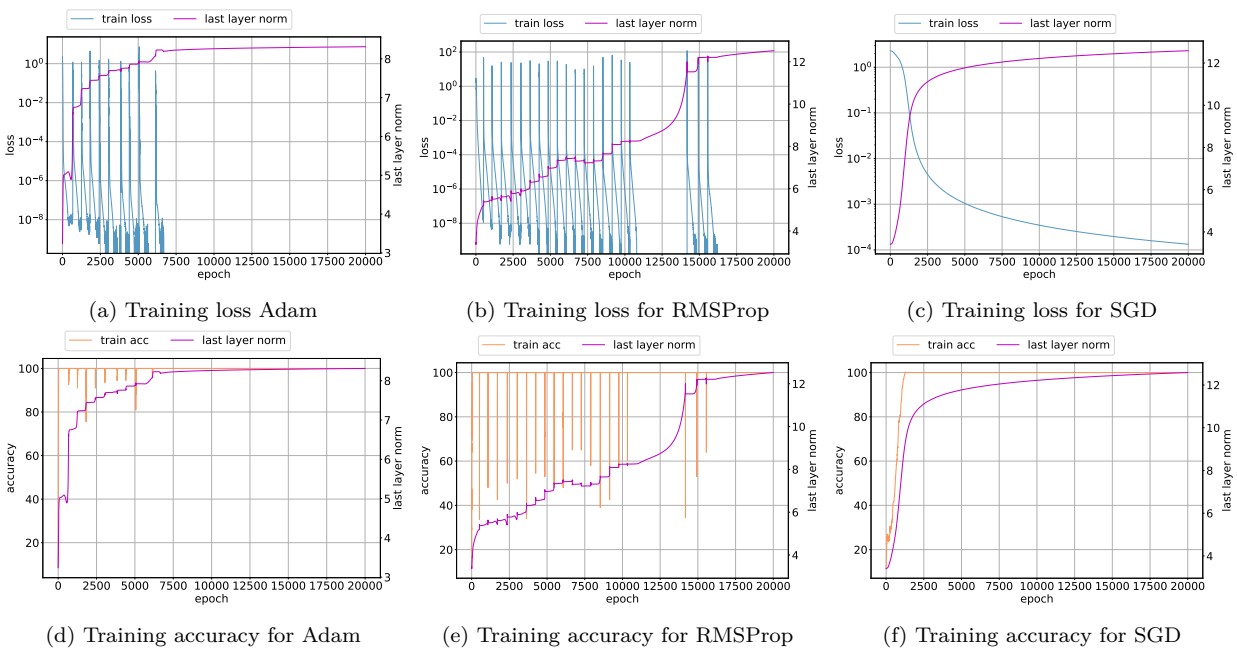

Figure 12: Optimizer choice and Slingshots. All optimizers train a 6-layer linear model full-batch on 200 CIFAR-10 samples.

## A.5 Deep linear models

We train a 6 layer linear model with 200 samples belonging to CIFAR-10 (Krizhevsky, 2009) with full-batch Adam (Kingma & Ba, 2014; Loshchilov & Hutter, 2017). The optimizer's hyperparameters are set as following: learning rate = 0.001, weight decay = 0, $\beta_1 = 0.9$, $\beta_2 = 0.95$, and $\epsilon = 10^{-8}$. Figure 12a and Figure 12d show the training loss and accuracy behavior observed during optimization. Multiple Slingshot stages are observed with this architecture as well.

### A.6 Learning Subset Parities

In this section we use the $k$-sparse parities of $n$ bits task as a test bed. Theoretically, this family of tasks is notoriously challenging since it poses strict computational lower bounds on learning (see (Barak et al., 2022) for more details). For the $(n, k)$ subset parity task, each input is a random $n$ dimensional vector such that each component is randomly sampled from $\sim \text{Unif}\{-1, 1\}$. The label is then given by a parity function over a predefined sparse set of $k \ll n$ bits. For the following experiments, we use $k = 3, n = 50$. For the model, we use a 3 layer MLP with *relu* activations, and the cross entropy loss. We use a dataset of 1000 samples, and a test set of 8000 samples. We train each network with Adam using a batch size of 32, a learning rate of $\eta = \{0.004, 0.003, 0.002\}$ and $\epsilon \in \{10^{-8}, 10^{-7}, 10^{-6}\}$. Our results are summarized in Figures 13, 14 and 15. For $\epsilon = 10^{-8}$, multiple Slingshots appear past the perfect fitting of the training set, with a bump in generalization post most Slingshots. For larger values of $\epsilon$, no Slingshots are observed, while generalization remains poor.

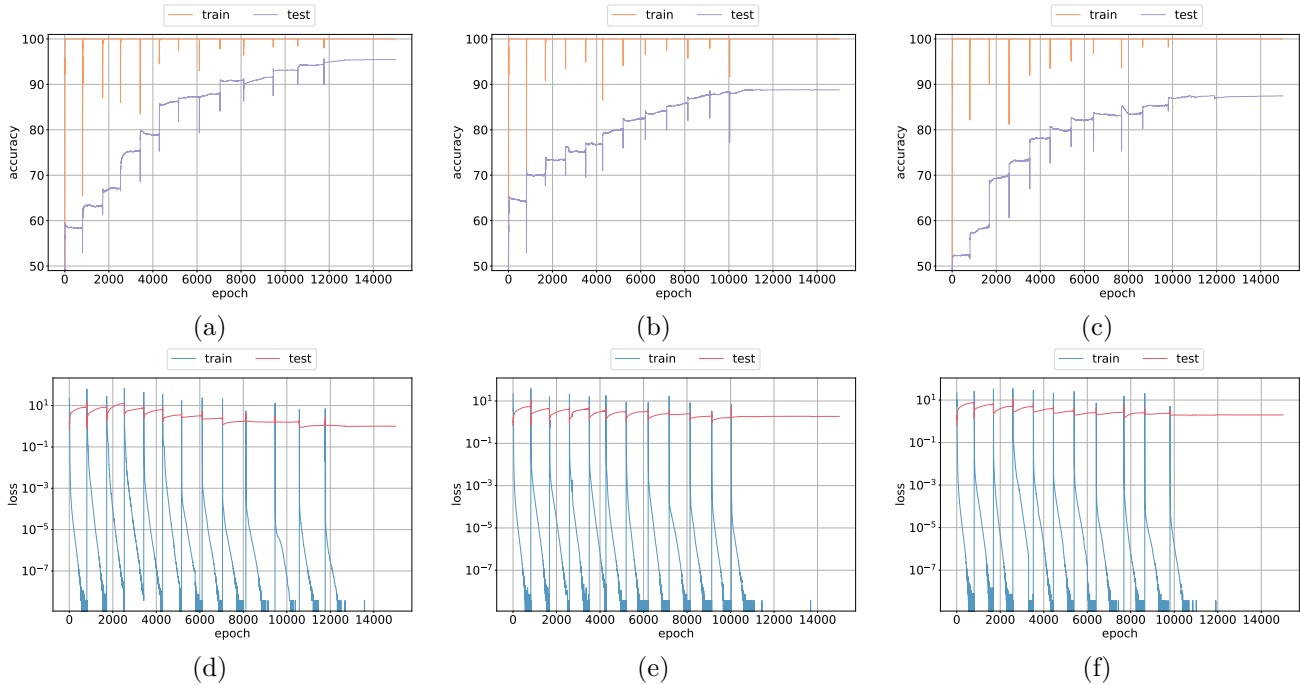

Figure 13: Learning a $(3, 50)$ subset parity with Adam with $\epsilon = 10^{-8}$ and a learning rate of (a),(d) $\eta = 0.004$, (b),(e) $\eta = 0.003$ and (c),(f) $\eta = 0.002$. Multiple Slingshots are visible, resulting in improved generalization.

### A.6.1 Effective Step Size and Curvature Dynamics

A classical results pertaining to optimizing smooth functions with gradient descent states that a sufficient condition for convergence requires that the learning rate does not exceed $\frac{2}{L}$, where $L$ is the Lipschitz constant of the gradient. Due to the sufficiency of the condition, we expect it to be violated at the phase transitions of the slingshots, when the training loss spikes. We quantify the effective step size of a parameter as $\frac{\eta}{\sqrt{V_t^2 + \epsilon}}$ where the terms are defined in Algorithm 1. To approximate $L$ in a local region, we use the maximum eigenvalue of the loss Hessian in this analysis as is done by a series of recent works including Cohen et al. (2021), Ahn et al. (2022) and Arora et al. (2022). We use the same setup described for training parity dataset to conduct this empirical analysis. The hyperparameters used for the optimizer include $\eta = 0.004$, $\epsilon = 10^{-8}$ and $\beta_1 = 0.9$ and $\beta_2 = 0.999$. Figure 16a shows the dynamics of the training and validation loss while Figures 16b, Figure 16c and Figure 16d shows the evolution of the effective step size as well as the maximum allowable step size for a few parameters chosen randomly from the three layers in the neural network. We observe from these plots that the effective step size is smaller than the maximum allowed step size in the vicinity of SlingShot Effects. however, at the phase transitions we clearly see that the effective step

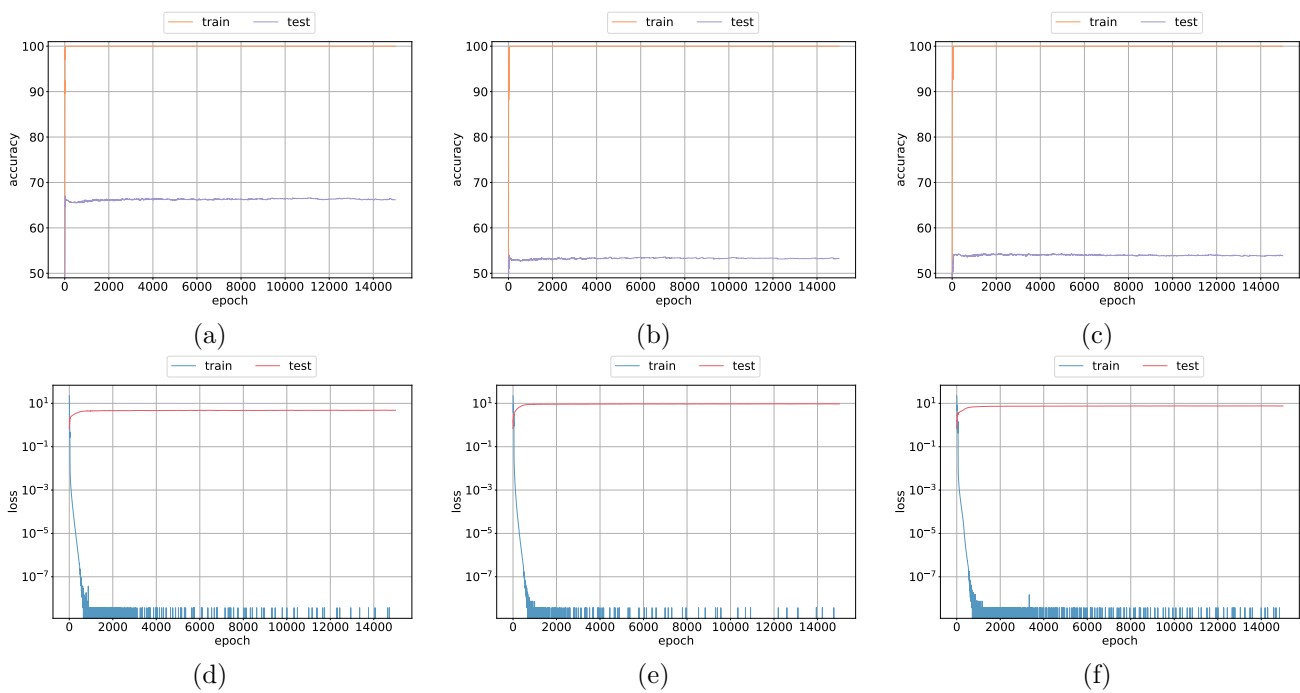

Figure 14: Learning a $(3, 50)$ subset parity with Adam with $\epsilon = 10^{-7}$ and a learning rate of (a),(d) $\eta = 0.004$, (b),(e) $\eta = 0.003$ and (c),(f) $\eta = 0.002$. No Slingshots are visible.

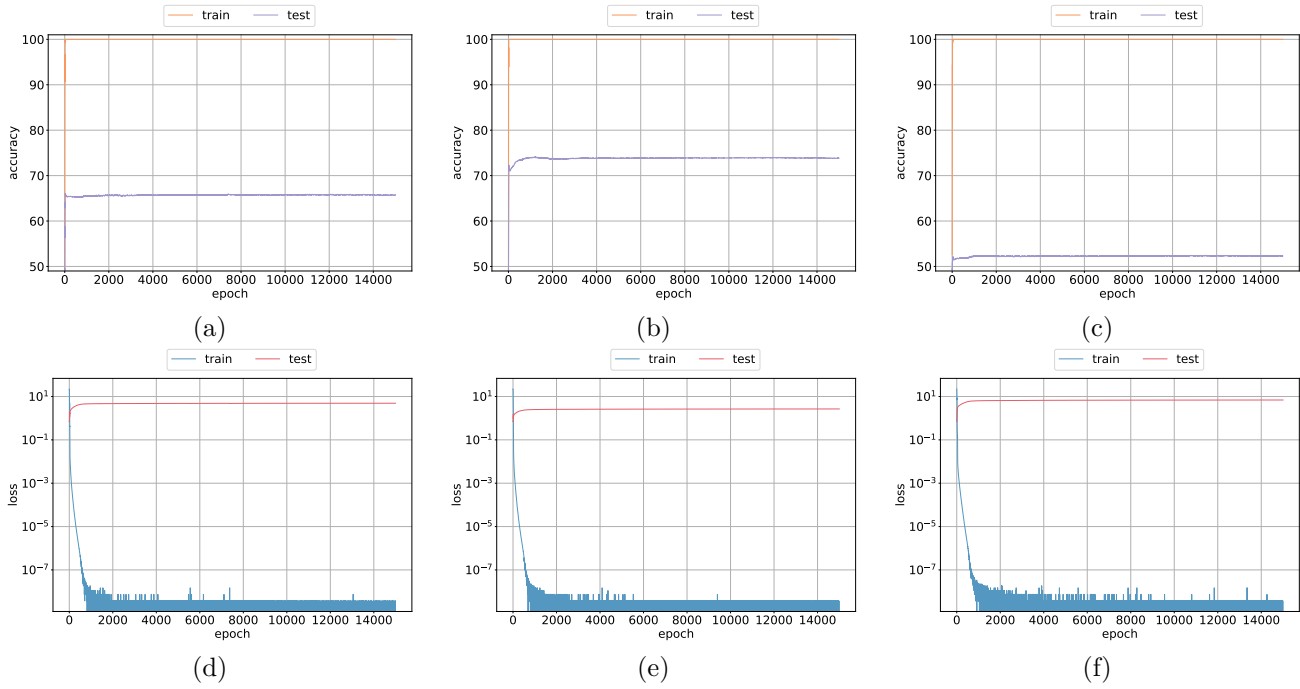

Figure 15: Learning a $(3, 50)$ subset parity with Adam with $\epsilon = 10^{-6}$ and a learning rate of (a),(d) $\eta = 0.004$, (b),(e) $\eta = 0.003$ and (c),(f) $\eta = 0.002$. No Slingshots are visible.

size is larger than the maximum allowed, causing the loss to spike. After a few Slingshot cycles, we observe that the maximum allowed step size increase dramatically, and no additional Slingshots follow.

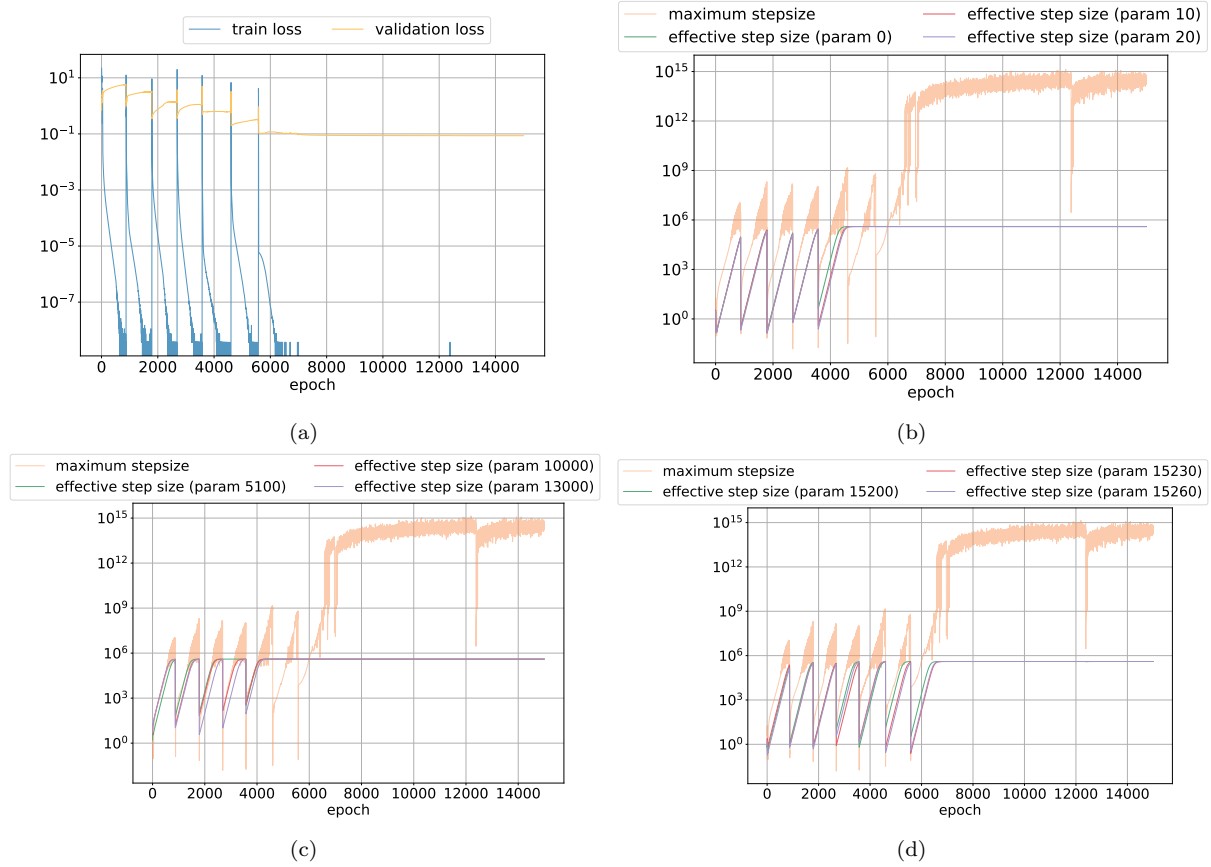

Figure 16: Empirical analysis of the relationship between Slingshot Effects and loss surface sharpness. Above plots include (a) training and validation loss; evolution of effective step size and curvature of parameters from (b) first layer, (c) second layer and (d) classification layer in a 3-layer MLP trained with Adam. At the phase transitions, effective step size is larger than $\frac{2}{L}$, initiating the slingshots. After a few cycles, the Lipschitz constant of the gradients decreases substantially, and the Slingshots cease.

Table 1: Optimizers hyperparameters. Learning rate is set to 0.001 and weight decay to 0 for all optimizers

| Optimizer | Other hyperparameters |
|-----------|----------------------|
| Adam | $\beta_1 = 0.9, \beta_2 = 0.95$ |
| RMSProp | $\alpha = 0.95$, momentum=0.0 |
| GD | momentum=0.9 |

## A.7 Different Optimizers

In this set of experiments, we study the training loss behavior of deep linear models optimized full-batch with Adam (Kingma & Ba, 2014; Loshchilov & Hutter, 2017), RMSProp (Tieleman & Hinton, 2012) and full-batch gradient descent (GD). The six layer model is trained with 200 samples. The hyperparameters used for optimizing the model with various optimizers are described in Table 1. Figure 12 shows the training loss and accuracy behavior of the three optimizers considered in this experiment. We observe Slingshot behavior with Adam and RMSProp from Figure 12 while Slingshot behavior is absent with standard gradient descent. This observation suggests that the normalization used in adaptive optimizers to calculate the update from gradients may lead to Slingshot behavior.

### A.8 Vision Transformers and Full CIFAR-10

In Appendix A, we have empirically shown that the existence of the Slingshot phenomenon on a small subset of CIFAR-10 dataset (Krizhevsky, 2009) with Vision Transformers (ViTs). We now study the impact that Slingshot has on the generalization ability of ViTs by training a model on all 50000 samples in CIFAR-10 training dataset. The ViT used here is a larger model than the one considered in A to account for larger dataset size. The ViT model consists of 12 layers, width 384 and 12 attention heads and is optimized by Adam (Kingma & Ba, 2014; Loshchilov & Hutter, 2017). For this experiment, we set the learning rate to 0.0001, weight decay to 0, $\beta_1 = 0.9$, $\beta_1 = 0.95$ and $\epsilon = 10^{-8}$, minibatch size of 512 and linear learning rate warmup for 1 epoch of optimization. Figure 17 shows the results of experiment with full CIFAR-10 dataset. Multiple Slingshots can be observed in these plots similar to the plots described in Appendix A. We observe from Figure 17d that the test accuracy peaks in epochs following a Slingshot with the maximum recorded test accuracy occurring very late in optimization. This observation suggests that the Slingshot can have a favorable effect on generalization consistent with the behavior observed in the main paper with division dataset.

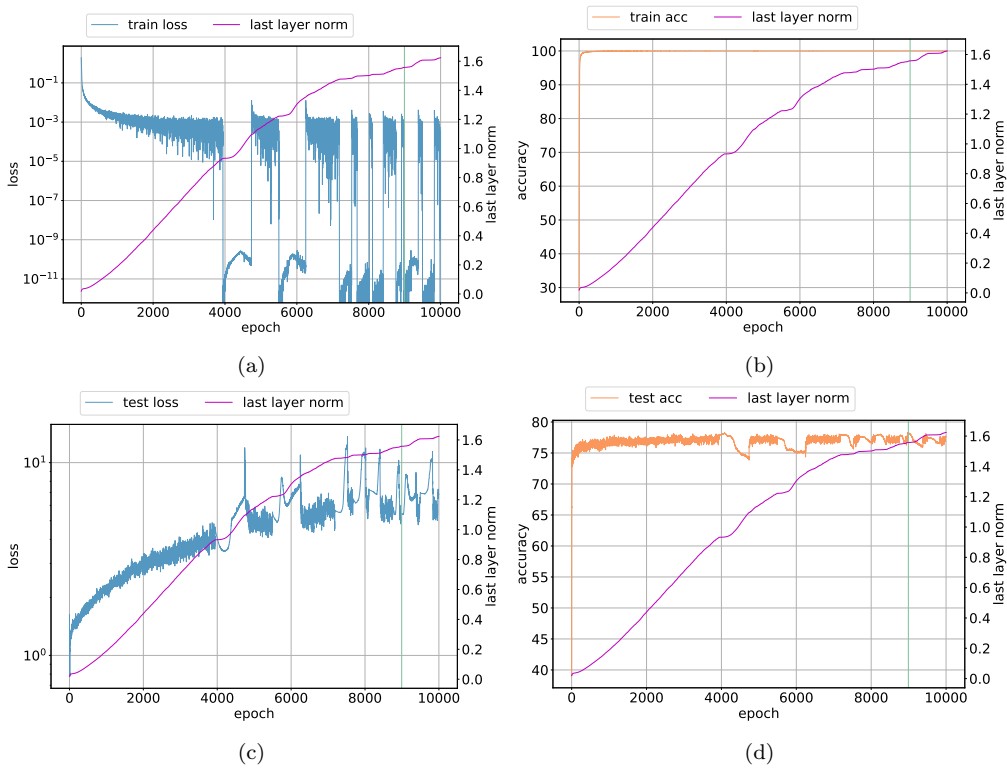

Figure 17: Slingshot generalization on full CIFAR-10 dataset: Norm growth versus a) loss on training data b) accuracy on training data (c) loss on test data d) accuracy on test data.

### A.9 Slingshot with MLP and Synthetic Dataset

In this section, we provide empirical evidence that Slingshot Effects are observed with a synthetic dataset in a fully-connected architecture. The small dimensional dataset, like the Grokking dataset of Power et al. (2021), allows us to easily measure of sharpness, given by $\frac{1}{\|u_t\|^2} u_t^\top \mathcal{H}_t u_t$ where $u_t$ is the optimizer's update vector and $\mathcal{H}_t$ is the Hessian at step $t$, to examine the interplay between Slingshot Effects and generalization.

#### A.9.1 Ablation Study

In this section, we train a toy model on a synthetically generated dataset with the aim of analysing the effect of different hyper parameters on the Slingshot Effect. We construct a 128-dimensional dataset with

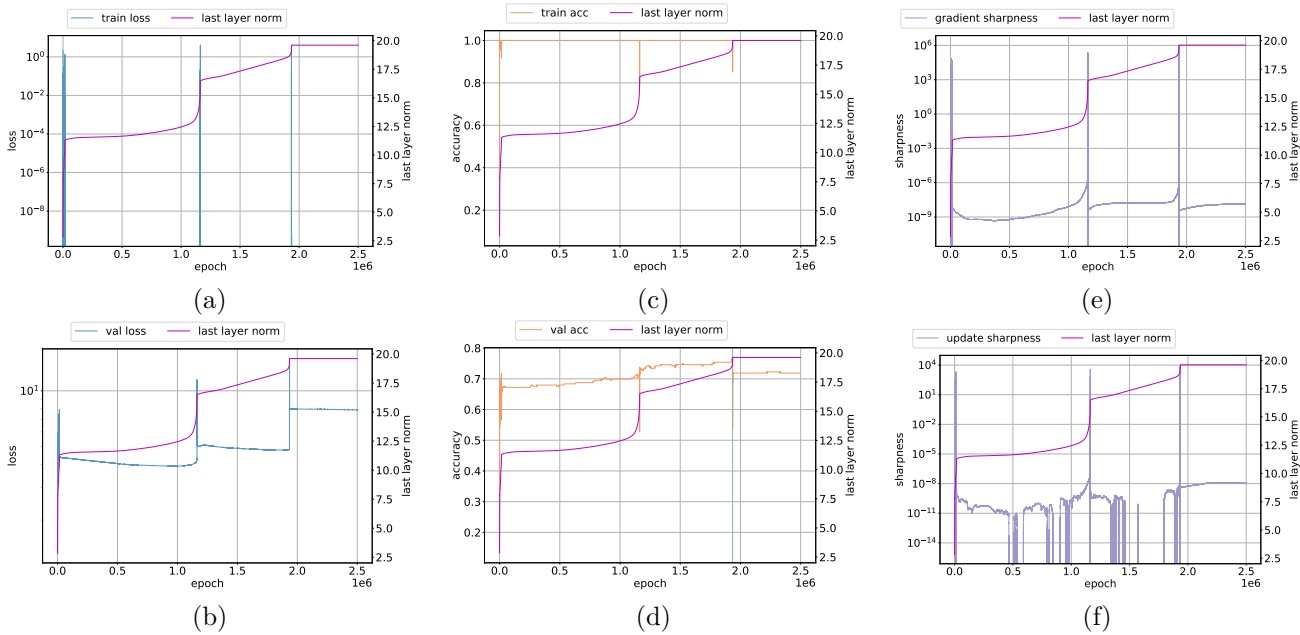

Figure 18: Slingshot generalization on synthetic dataset: Norm growth versus a) loss on training data b) accuracy on training data (c) loss on validation data d) accuracy on validation data. Note that the vertical line in green shows location of maximum test accuracy. Adam hyperparameters are $\beta_1 = 0.9$, $\beta_1 = 0.95$, $\epsilon = 10^{-8}$

Scikit-learn (Pedregosa et al., 2011) that has 3 informative dimensions that represents a 8-class classification problem. The class centers are the edges of a 3-dimensional hypercube around which clusters are data are sampled from a standard normal distribution. The other 125-dimensions are also filled at random to create a high-dimensional dataset used in our experiments. We generate 256 training and validation samples for this dataset and use a minibatch size of 128 in all the experiments described in the following.

**Architecture and Optimizer**  Figure 18 shows the training and validation metrics when we optimize a 4-layer fully-connected network (FCN) with Adam using a learning rate of 0.001, $\beta_1 = 0.9$, $\beta_1 = 0.95$, no weight decay and $\epsilon = 10^{-8}$. Note that we use this value of $\epsilon$ in our first experiment as this is the default value proposed in Kingma and Ba (Kingma & Ba, 2014). These experiments are implemented in JAX (Bradbury et al., 2018).

**Tuning $\epsilon$**  In the next set of experiments with synthetic data, we tune $\epsilon$ value for Adam to understand its impact on test accuracy. Figure 19 shows a plot of the maximum validation accuracy achieved by models trained with Adam as a function of time (epoch). We observe that Adam reaches its best test accuracy late in optimization with $\epsilon = 10^{-5}$ yielding the highest validation accuracy. Furthermore, the best accuracy is achieved with a model that experiences Slingshot during optimization. This observation is consistent with our findings for ViT training with CIFAR-10 dataset described in the main paper and Appendix A.8.

**Influence of $\beta_1$ and $\beta_2$**  In these experiments, we aim to study the impact of Adam optimizer's $\beta_1$ and $\beta_2$ hyperparameters on Slingshot. We use the synthetic data described above and set the learning rate of 0.001 and $\epsilon = 10^{-8}$ for this analysis. Figure 20 and Figure 21 shows the results of this study. We observe from Figure 20 that the Slingshot Effect is fairly robust to the values of $\beta_1$ and $\beta_2$. Figure 20a-Figure 20c show that Slingshot is even observed with $\beta_1$ and $\beta_2$ set to 0 which effectively disables exponential moving averaging of gradient moments in Adam (Kingma & Ba, 2014). Figure 20g-Figure 20i provide an example of hyperparameters that fail to induce Slingshot. We observe from Figure 20 that models that experience Slingshot tend to reach their best test accuracy during the later stages of training. Specifically, we observe from Figure 20b, Figure 20e and Figure 20k that the best validation accuracy occurs after 60000 epochs. These examples provide further evidence about an interesting implicit bias of Adam. Figure 21 shows more

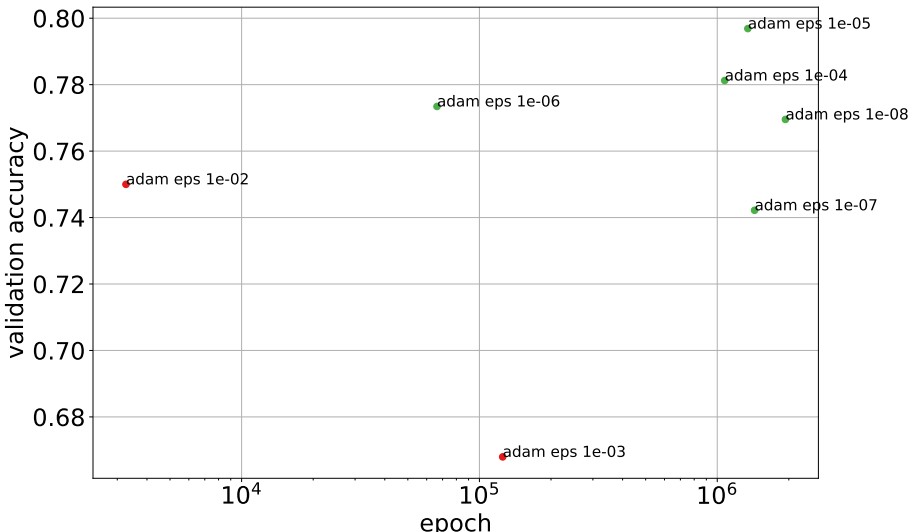

Figure 19: Slingshot on syntehtic dataset. Note that the points marked in: (i) green correspond to Adam-trained models that undergo Slingshot, (ii) red correspond to Adam-trained models that do not experience Slingshot; Adam's hyperparameters are given by $\beta_1 = 0.9$, $\beta_2 = 0.95$, no weight decay and $\epsilon$ shown in parentheses.

examples of hyperparameters that do not induce Slingshot Effects. Finally, we observe from Figure 21 that hyperparameters that provide higher validation accuracy are from models that experience Slingshot Effects.

## B   Slingshot and Grokking

We use the empirical setup described by Power et al. (2021) to describe the Slingshot Effect. The following section describes relevant details including datasets, architecture and optimizer used in our experiments.

**Architecture**   The model used a decoder-only Transformer (Vaswani et al., 2017) with causal attention masking. The architecture used in all our experiments consists of 2 decoder layers with each layer of width 128 and 4 attention heads.

**Optimization**   We train the architecture described above with Adam optimizer (Kingma & Ba, 2014; Loshchilov & Hutter, 2017) in most of our experiments unless noted otherwise. The learning rate is set to 0.001 and with linear learning rate warmup for the first 10 steps. We use $\beta_1 = 0.9$, $\beta_2 = 0.98$ for Adam's hyperparameters. The Transformers are optimized with cross-entropy (CE) loss that is calculated on the output tokens for a given binary operation.

**Algorithmic Datasets**   The Transformer is trained on small algorithmic datasets that consists of sequences that represent a mathematical operation. The following operations are used in our experiments:

$$c = a + b \pmod{p} \text{ for } 0 \le a, b < p$$

$$c = a - b \pmod{p} \text{ for } 0 \le a, b < p$$

$$c = a * b \pmod{p} \text{ for } 0 \le a, b < p$$

$$c = a \div b \pmod{p} \text{ for } 0 \le a, b < p$$

$$c = a^2 + b \pmod{p} \text{ for } 0 \le a, b < p$$

$$c = a^3 + b \pmod{p} \text{ for } 0 \leq a, b < p$$

$$c = a^2 + b^2 \pmod{p} \text{ for } 0 \leq a, b < p$$

$$c = a^2 + b^2 + ab \pmod{p} \text{ for } 0 \leq a, b < p$$

$$c = a^2 + b^2 + ab + b \pmod{p} \text{ for } 0 \leq a, b < p$$

$$c = a^3 + ab \pmod{p} \text{ for } 0 \leq a, b < p$$

$$c = a^3 + ab^2 + b \pmod{p} \text{ for } 0 \leq a, b < p$$

$$c = [a \div b \pmod{p} \text{ if } b \text{ is odd, otherwise } a - b \pmod{p}] \text{ for } 0 \leq a, b < p$$

$$c = a \cdot b \text{ for } a, b \in S_5$$

$$c = a \cdot b \cdot a^{-1} \text{ for } a, b \in S_5$$

$$c = x \cdot b \cdot a \text{ for } a, b \in S_5$$

$$c = [a + b \pmod{p} \text{ if } a \text{ is even, otherwise } a * b \pmod{p}] \text{ for } 0 \leq a, b < p$$

$$c = [a + b \pmod{p} \text{ if } a \text{ is even, otherwise } a - b \pmod{p}] \text{ for } 0 \leq a, b < p$$

where $p = 97$ and with the dataset split in training and validation data. Each equation in the dataset is of the form $(a)(op)(b)(=)c$ where (x) represents the token used to represent x. We refer to Power et al. (2021) for a detailed description of the datasets

## B.1 Analysis of Parameter Dynamics

A common observation is that intermediate representations tend to evolve beyond simple scale increase during phase transitions from rapid norm growth phase to curtailed norm growth phase. In order to empirically quantify this effect, we train the Transformer described in Appendix B with modular addition, multiplication and division datasets using Adam with learning rate set to 0.001 and $\beta_1 = 0.9$ and $\beta_2 = 0.98$. We calculate the cosine distance between the representation and classification parameters from their initial values where the cosine distance is given by

$$d^{repr} = 1.0 - \frac{w_t^{repr}}{\|w_t^{repr}\|} \cdot \frac{w_0^{repr}}{\|w_0^{repr}\|}$$

$$d^{clf} = 1.0 - \frac{w_t^{clf}}{\|w_t^{clf}\|} \cdot \frac{w_0^{clf}}{\|w_0^{clf}\|}$$

where $d^{repr}$ ($d^{clf}$) denotes cosine distance for representation (respectively classification) parameters, $w_t^{repr}$ (resp. $w_t^{clf}$) denotes representation (resp. classification) parameters at time $t$ with $w_0^{repr}$ ($w_t^{clf}$) indicating the initial representation (resp. classification) parameters where the norm used above is the Euclidean norm.

Figure 22 shows the dynamics of the loss, accuracy and cosine distance recorded during training. We observe that the classification parameters move farther away from initialization faster than the representation parameters. More interestingly, we observe from Figure 22c and Figure 22f that the representation parameters travel farther from initialization for training runs that experience Slingshot. These trials use $\epsilon = 10^{-8}$ and $\epsilon = 10^{-7}$ and experience Slingshot Effects. In contrast, we see from Figure 22i and Figure 22l that the representation distance remains low for models trained with $\epsilon = 10^{-5}$ and $\epsilon = 10^{-4}$. The models trained with higher $\epsilon$ values do not experience Slingshot Effects. These results suggest that Slingshot may have a beneficial effect in moving the representation parameters away from initialization which eventually helps with model generalization. Figure 23 and Figure 24 show a similar trend for multiplication and division datasets respectively.

## B.2 SGD Optimization

In this appendix, we show that Slingshot Effects are not seen during Transformer training with stochastic gradient descent (SGD) with momentum to support our claim in the main paper. To this end, we use train the Transformer described in in Appendix B on modular division dataset with a 50/50 train/validation split using SGD with momentum. We use a mini-batch size of 512 which requires the optimizer to take 10 steps per epoch for dataset split described above. We set momentum to 0.9 and use the following learning rates: 0.001, 0.01 and 0.1 and run the optimizer for 1500000 steps. The number of steps used here is 3 times larger than the steps used to run Adam in this work which is chosen to give SGD additional time to reach convergence. Figure 25 shows the usual loss and accuracy metrics calculated on training and validation data as well as the weight norm of the classifier layer. We observe that there is no evidence of Slingshot with SGD. Lastly, we do not see any evidence of Grokking or generalization with this setup as well.

## B.3 Tuning Adam Optimizer's $\beta_2$ Hyperparameter

Zhang et al. (2022) showed in a recent paper that vanilla Adam (Kingma & Ba, 2014) can converege without any modifications to its update equations. A key message in (Zhang et al., 2022) is to tune the $\beta$ hyperparameters that are used to smooth momentum and second order moment terms in Adam (Kingma & Ba, 2014) with a focus on tuning $\beta_2$ which controls the denominator in the update equations. We study the behavior of Adam optimizer (Kingma & Ba, 2014) by training a Transformer described in Appendix B with modular division dataset split evenly into training and validation datasets. Figure 26 shows the behavior of optimizing Transformers with Adam for three values of $\beta_2$ including 0.98 which is the default used both by Power et al. (2021) and in this paper as well as very high $\beta_2$ values including 0.9995 and 0.9997. We observe training instabilities and Slingshot Effects in all of these experiments from Figure 26. These observations suggest that Slingshots can occur during neural networks optimization despite careful tuning of $\beta_2$ hypermater.

## B.4 Slingshots with Additional Datasets

In this appendix, we provide evidence of Slingshot Effects on additional datasets from Power et al. (2021) Grokking work. The datasets are created by a subset of mathematical operations defined in Appendix B. Each operation can have multiple datasets that depends on the train/validation split ratio. We use the training setup described in B on 18 separate datasets. Figure 27 - Figure 44 shows the results the datasets described in this appendix. We observe Slingshot Effects and generalization with all 18 datasets. These results suggest the prevalence of Slingshot Effects when large models are trained with adaptive optimizers, specifically Adam (Kingma & Ba, 2014).

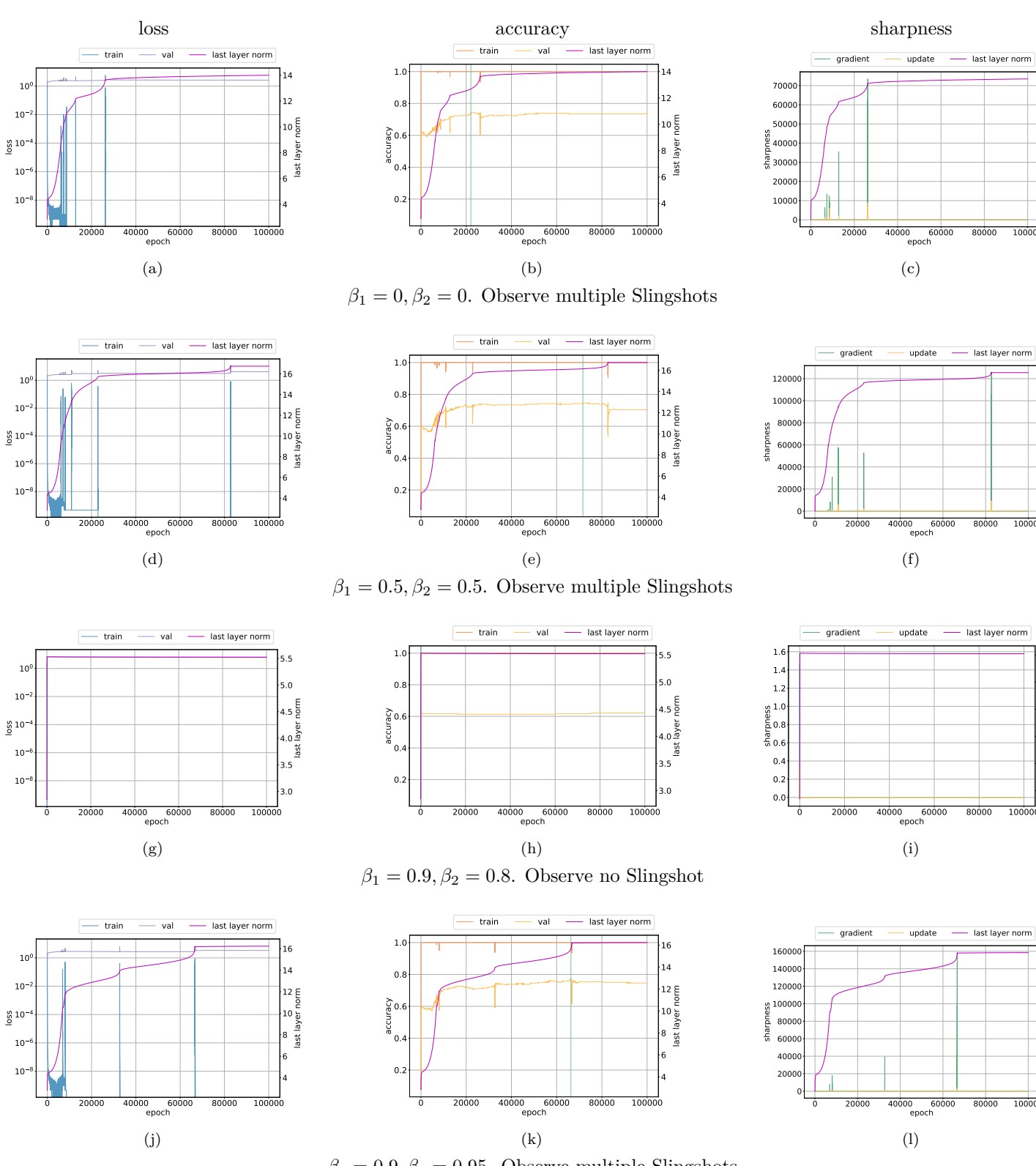

Figure 20: Varying $\beta_1, \beta_2$ in Adam on synthetic dataset. FCN is trained with Adam using learning rate 0.001 and $\epsilon = 10^{-6}$. The validation accuracy of models that experience Slingshot reach their highest accuracy later in training.

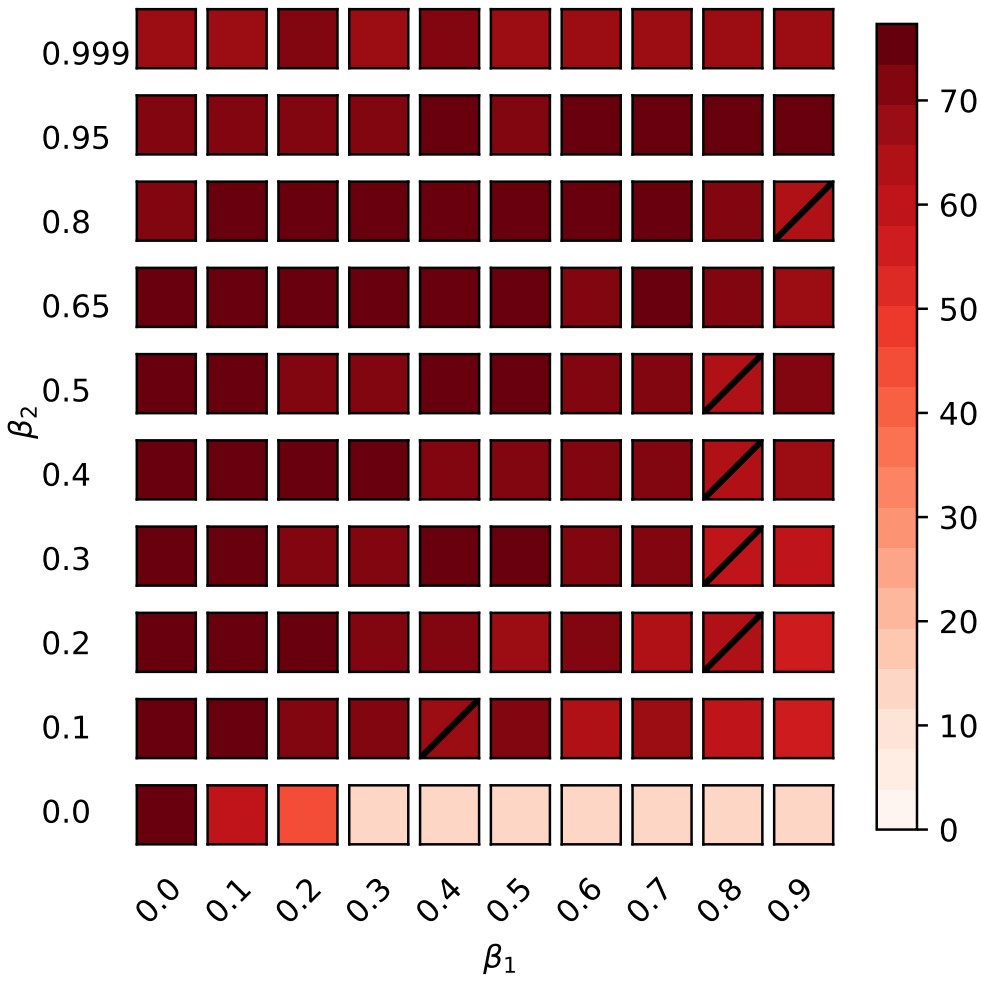

Figure 21: Extended analysis of $\beta_1, \beta_2$ in Adam on synthetic dataset. Plot shows the highest validation accuracy achieved with various values of $\beta_1, \beta_2$ with learning rate set to 0.001 and $\epsilon = 10^{-6}$. Hyperparameters that do not induce Slingshot Effects are marked with a diagonal line in black. Models trained with $\beta_1 > 0.2$ and $\beta_2 = 0$ diverged during training due to instability. These trials have their validation accuracy set to chance level.

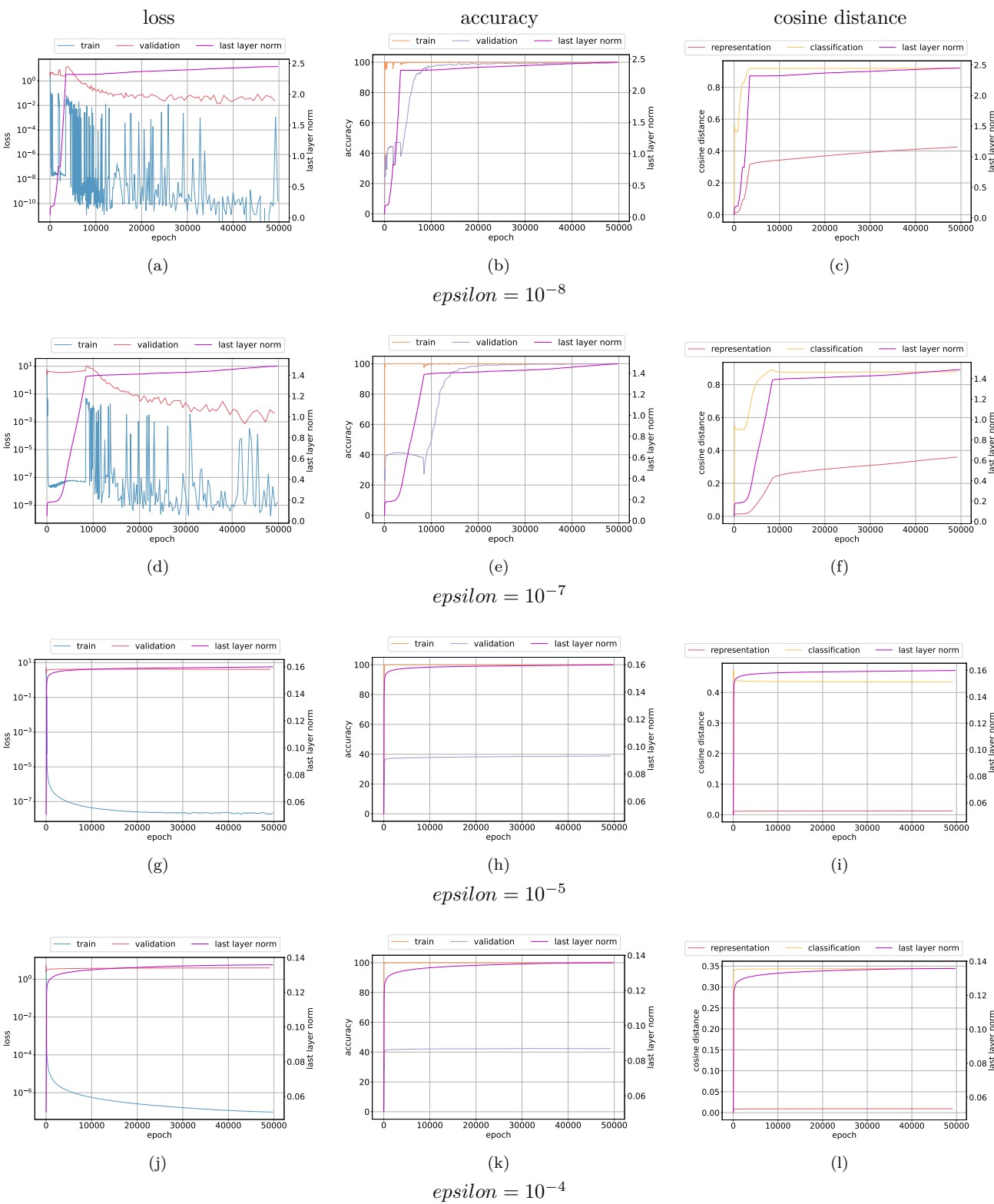

Figure 22: Cosine distance evolution for Transformer described in Appendix B trained on modular addition. Observe that the cosine distance from initialization increases with models that experience Slingshot Effects.

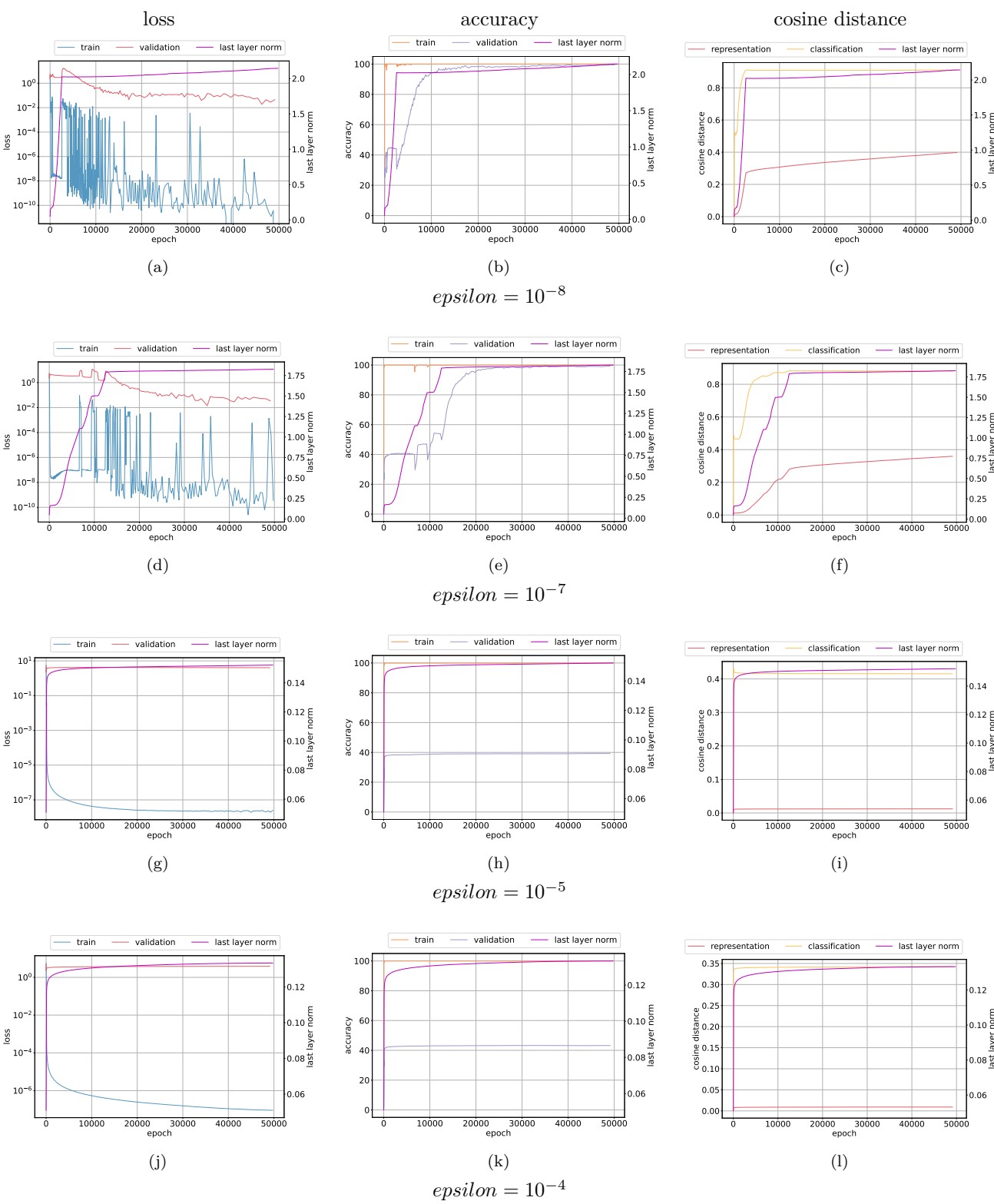

Figure 23: Cosine distance evolution for Transformer described in Appendix B trained on modular multiplication. Observe that the cosine distance from initialization increases with models that experience Slingshot Effects.

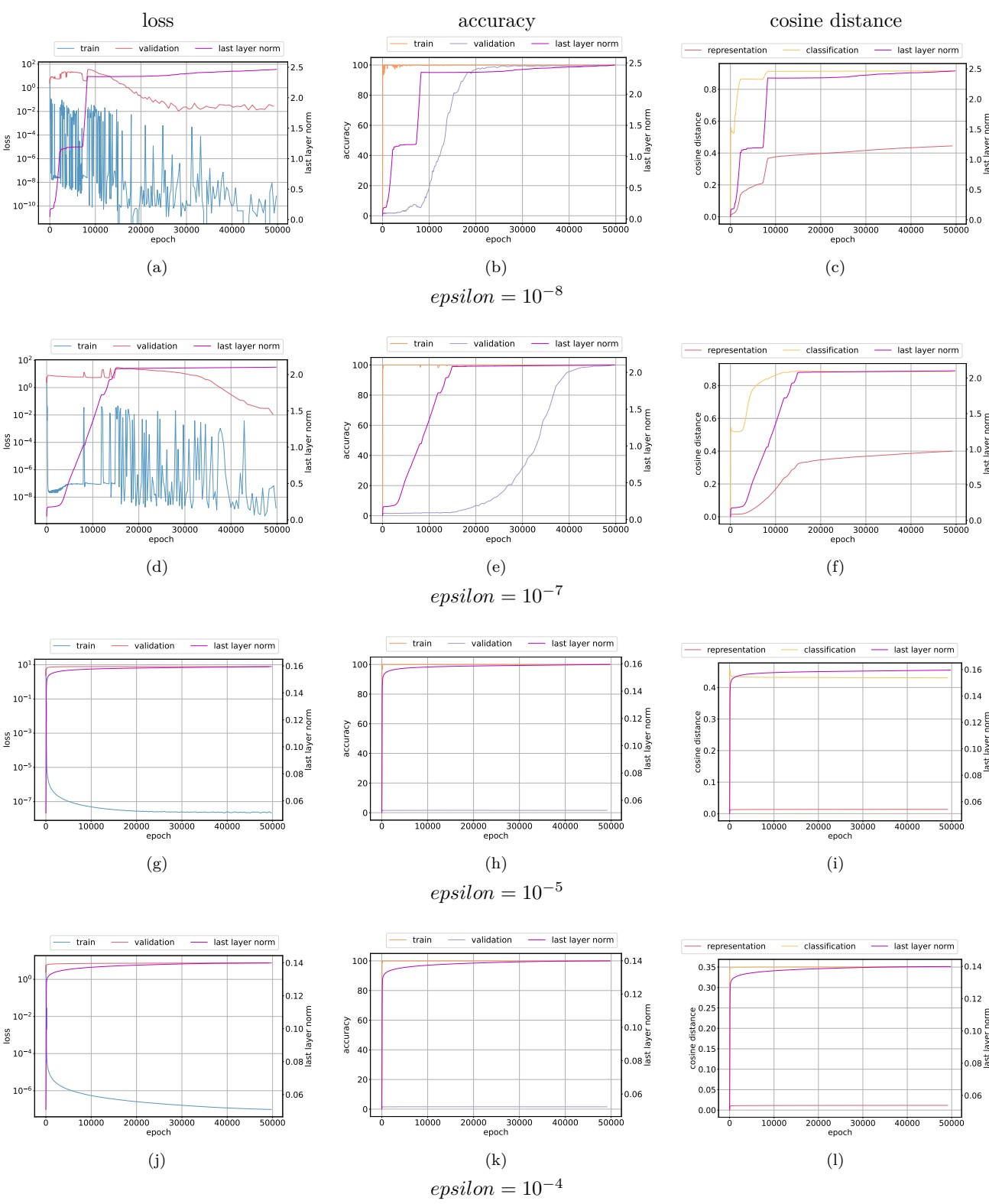

Figure 24: Cosine distance evolution for Transformer described in Appendix B trained on modular division. Observe that the cosine distance from initialization increases with models that experience Slingshot Effects.

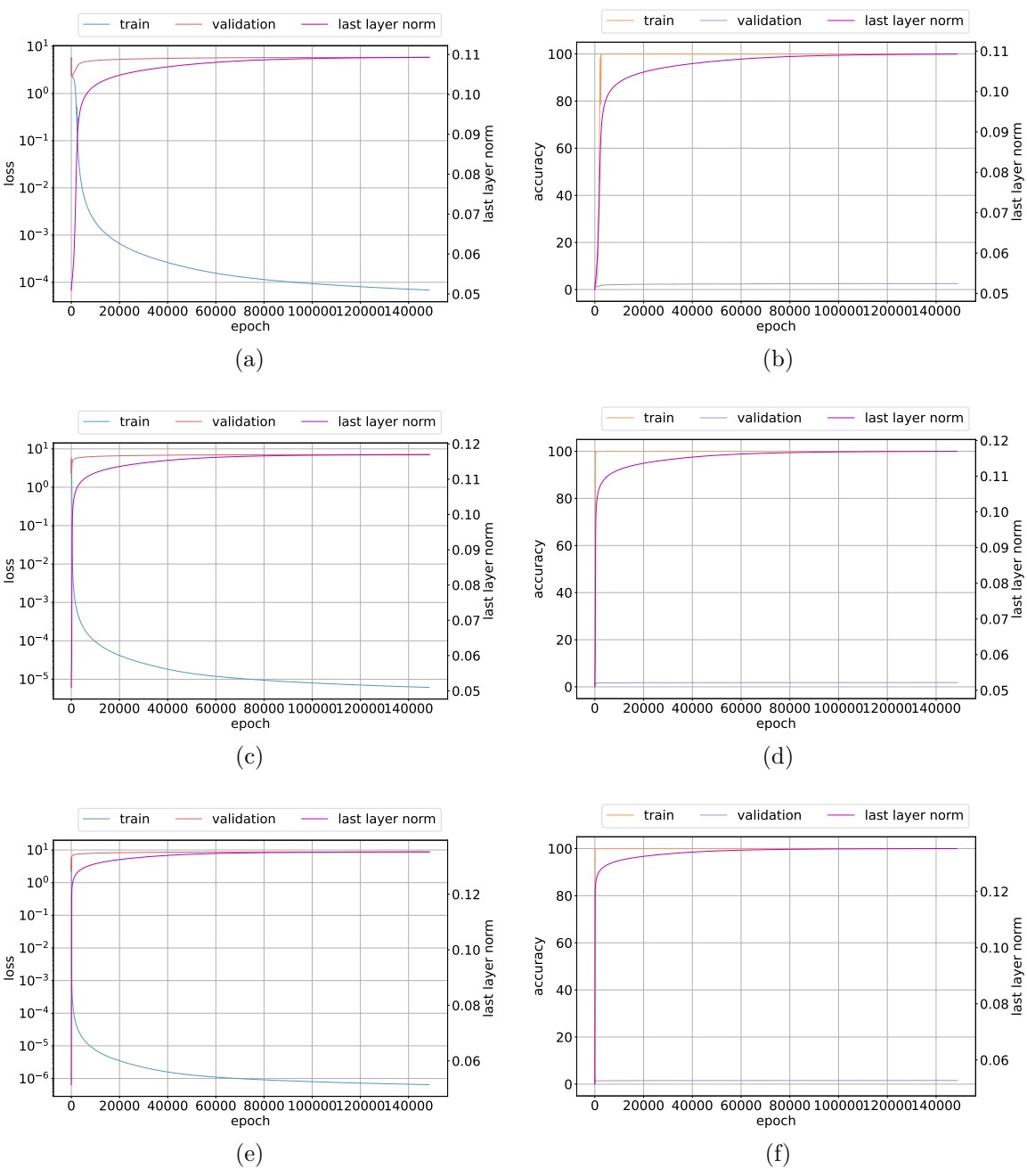

Figure 25: Optimizing a Transformer with SGD on modular division dataset: Norm growth vs (a), (c), (e) training and validation loss, (b), (d), (f) training and validation accuracy. Note the lack of Slingshot Effects, Grokking and generalization seen with Adam optimizer.

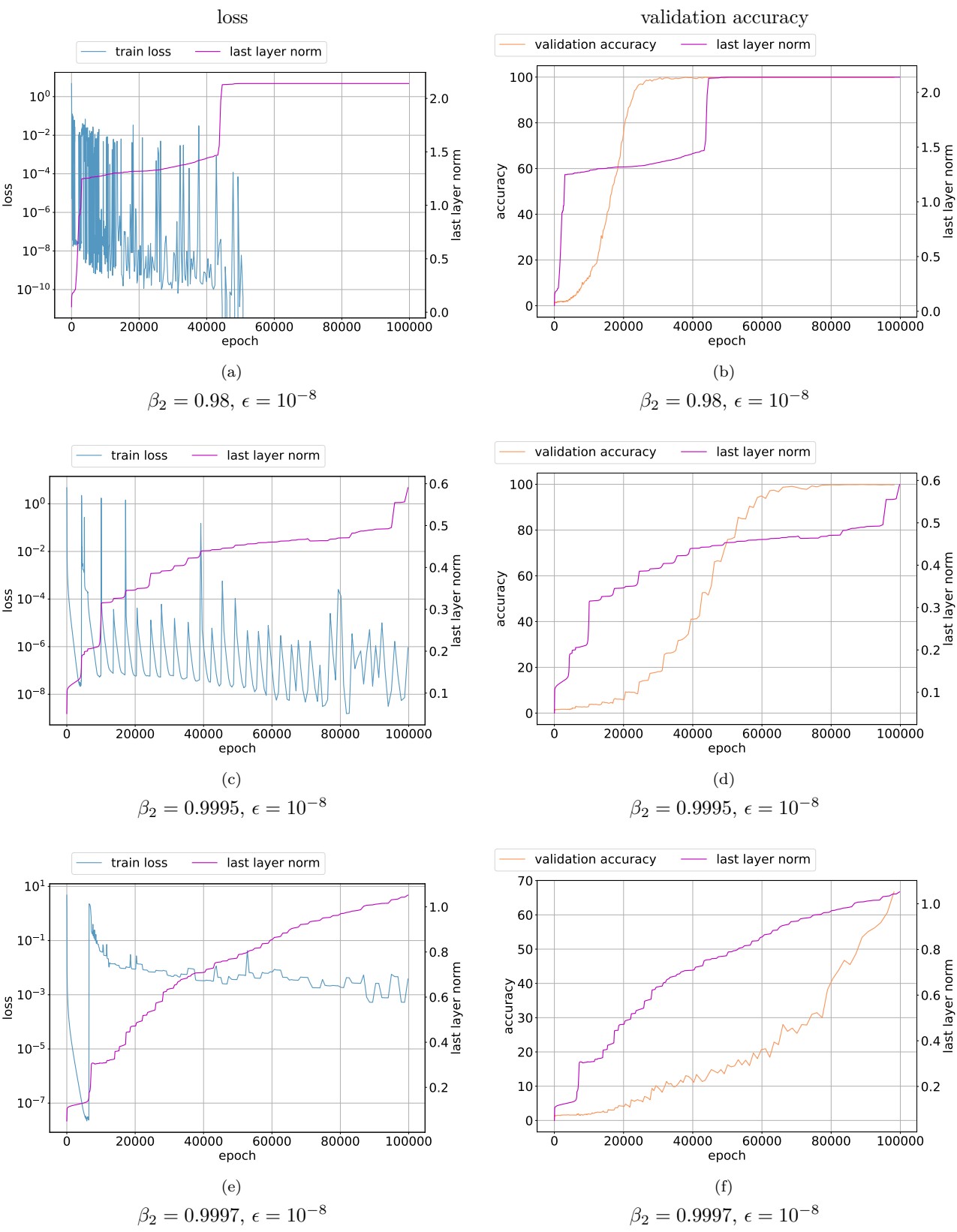

Figure 26: Effect of varying Adam optimizer's $\beta_2$ hyperprameter. Plot shows train loss and validation accuracy of a Transformer trained on modular division dataset described in Appendix B. Observe that Slingshot Effects and training spikes still persists at very high values of $\beta_2$.

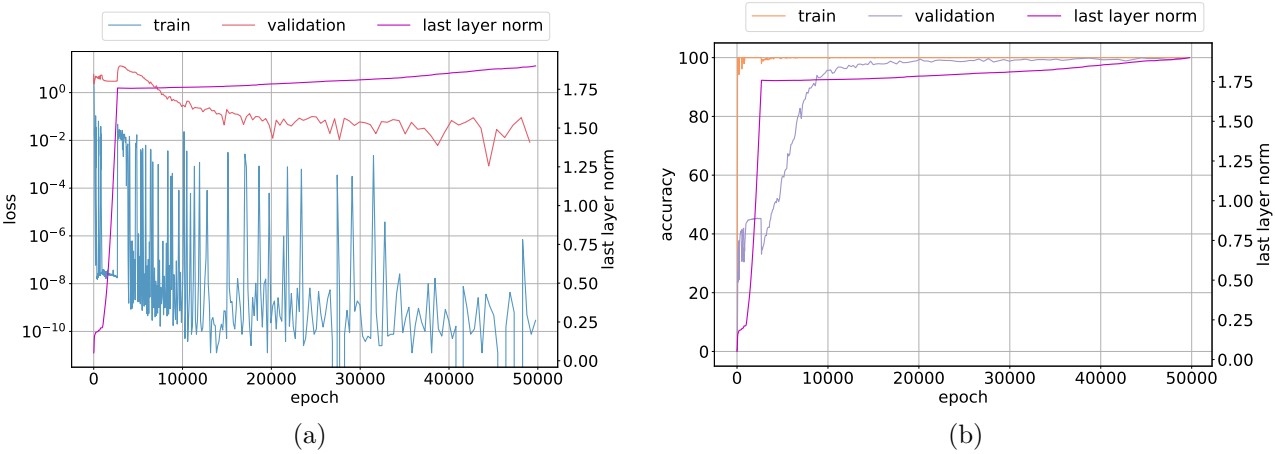

Figure 27: Addition dataset with 50/50 train/validation split. Training and validation (a) loss and (b) accuracy.

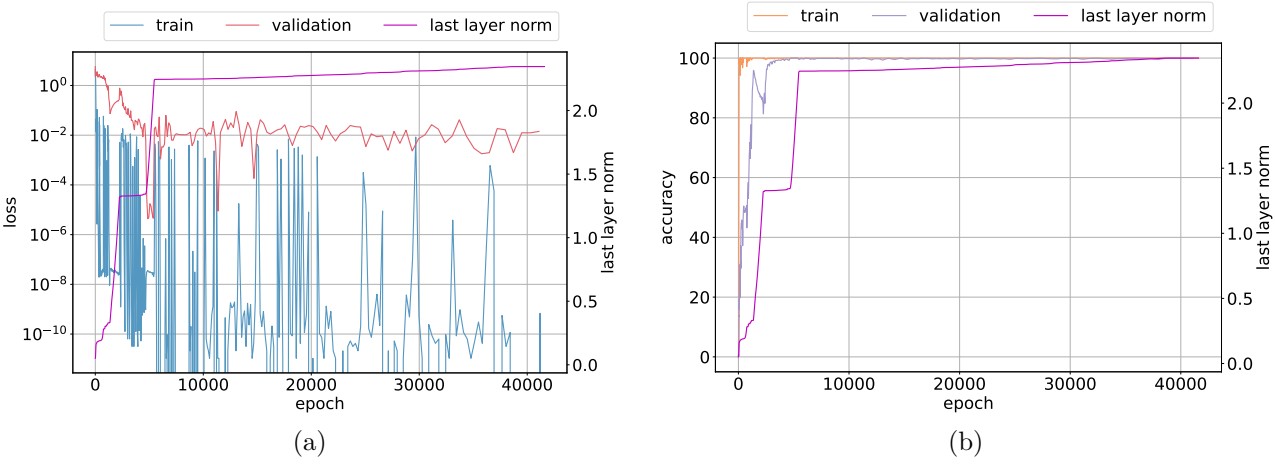

Figure 28: Addition dataset with 60/40 train/validation split. Training and validation (a) loss and (b) accuracy.

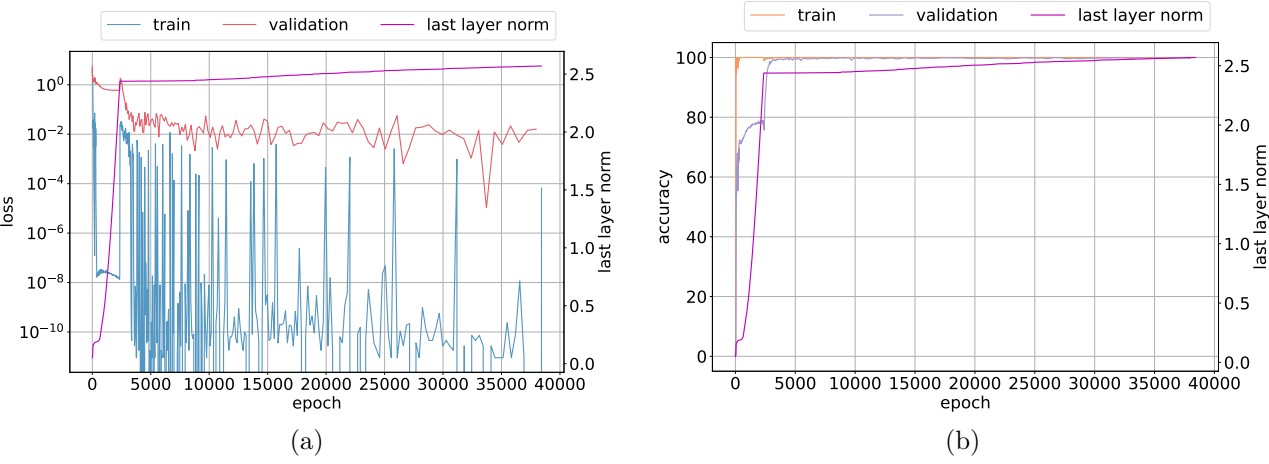

Figure 29: Addition dataset with 70/30 train/validation split. Training and validation (a) loss and (b) accuracy.

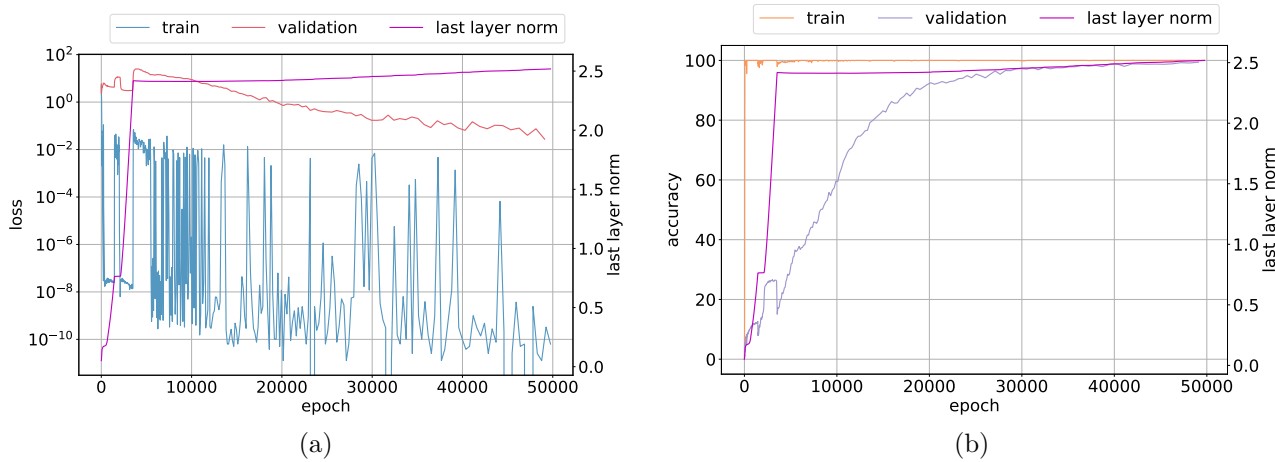

Figure 30: Cubepoly dataset with 50/50 train/validation split. Cubepoly operation is given by $(a^3 + b \pmod{p}$ for $0 \le a, b < p)$. Training and validation (a) loss and (b) accuracy.

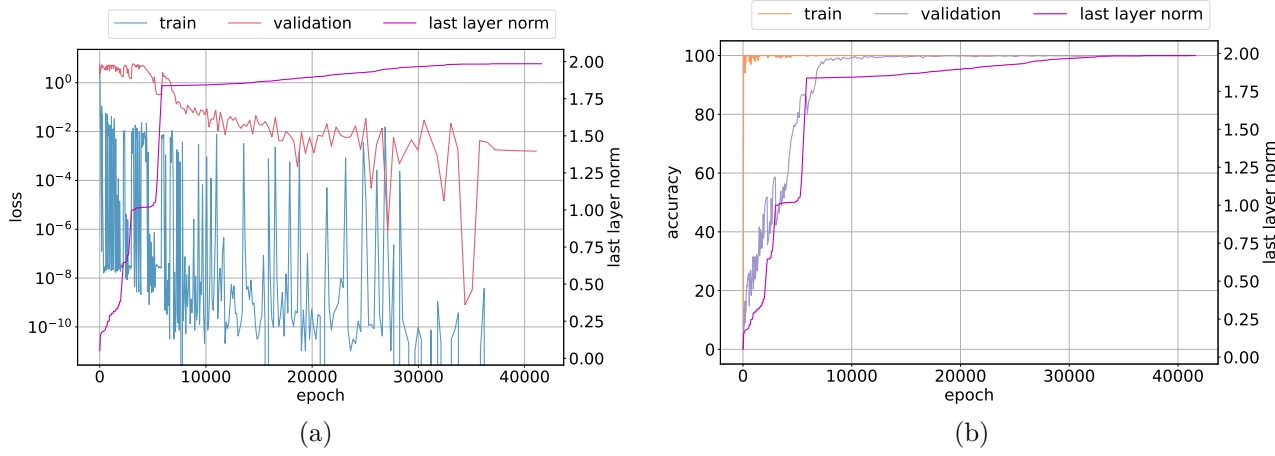

Figure 31: Cubepoly dataset with 60/40 train/validation split. Cubepoly operation is given by $(a^3 + b \pmod{p}$ for $0 \le a, b < p)$. Training and validation (a) loss and (b) accuracy.

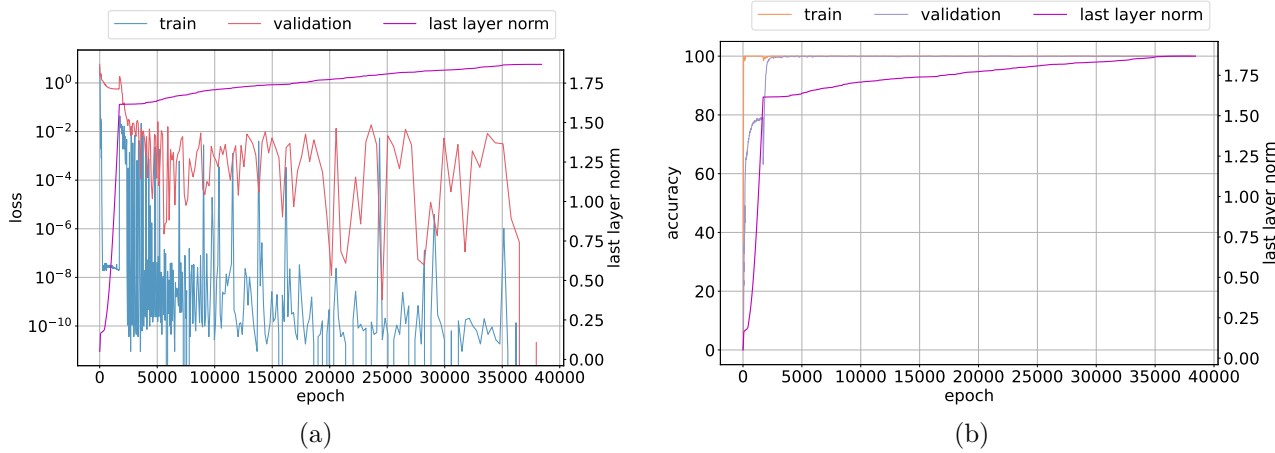

Figure 32: Cubepoly dataset with 70/30 train/validation split. Cubepoly operation is given by $(a^3 + b \pmod{p}$ for $0 \le a, b < p)$. Training and validation (a) loss and (b) accuracy.

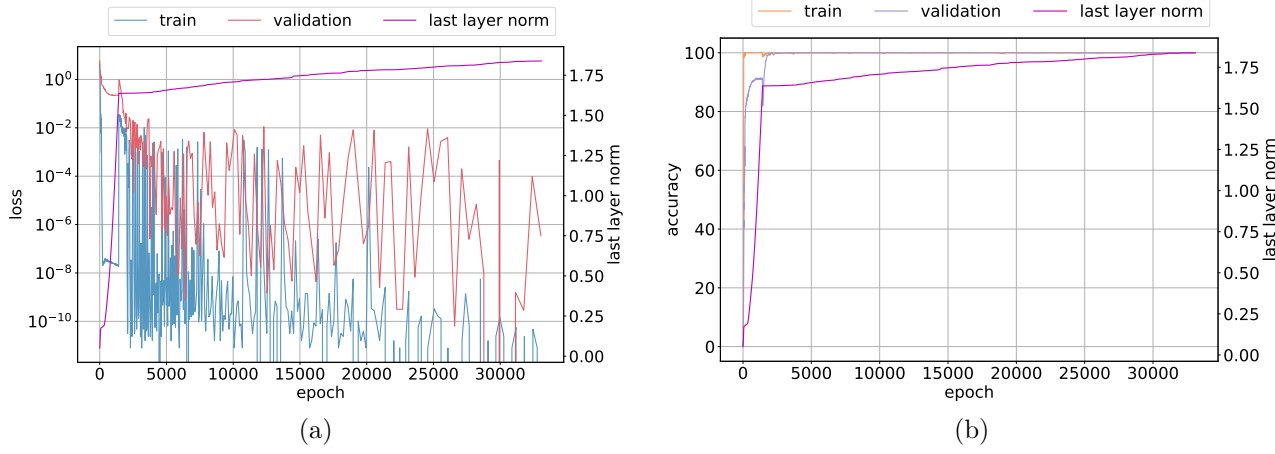

Figure 33: Cubepoly dataset with 80/20 train/validation split. Cubepoly operation is given by $(a^3 + b \pmod{p}$ for $0 \le a, b < p)$. Training and validation (a) loss and (b) accuracy.

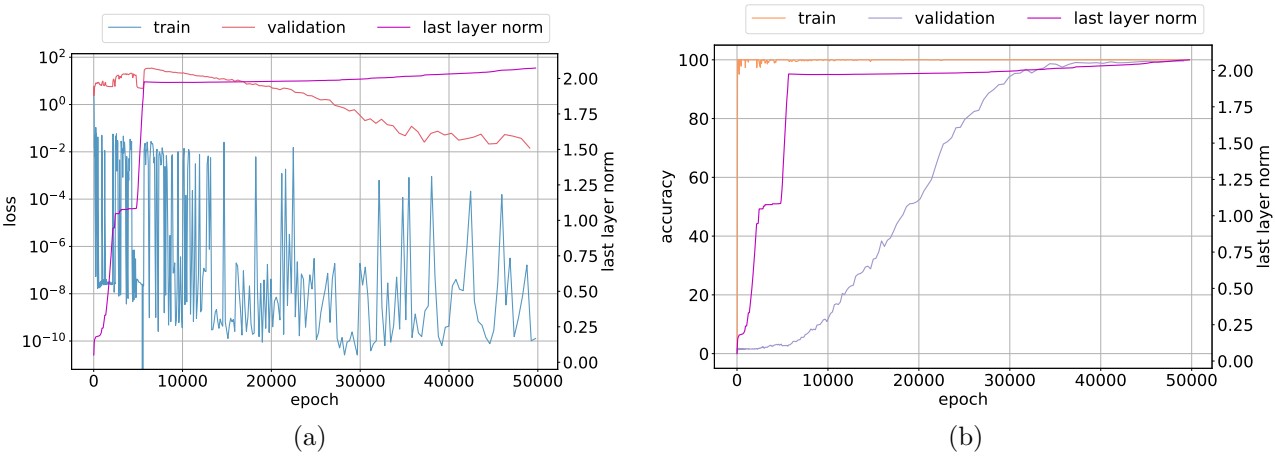

Figure 34: Division dataset with 50/50 train/validation split. Training and validation (a) loss and (b) accuracy.

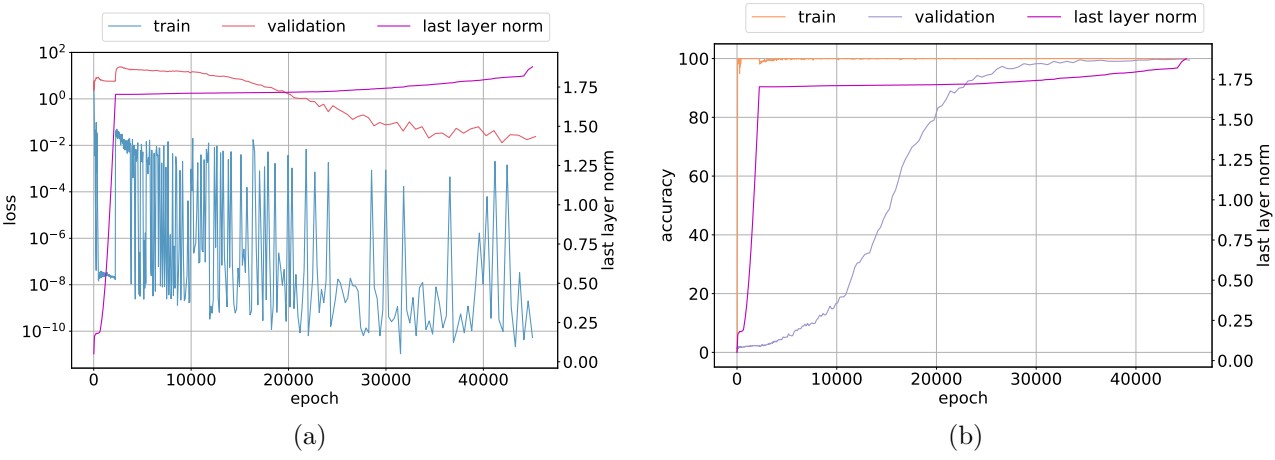

Figure 35: Division dataset with 60/40 train/validation split. Training and validation (a) loss and (b) accuracy.

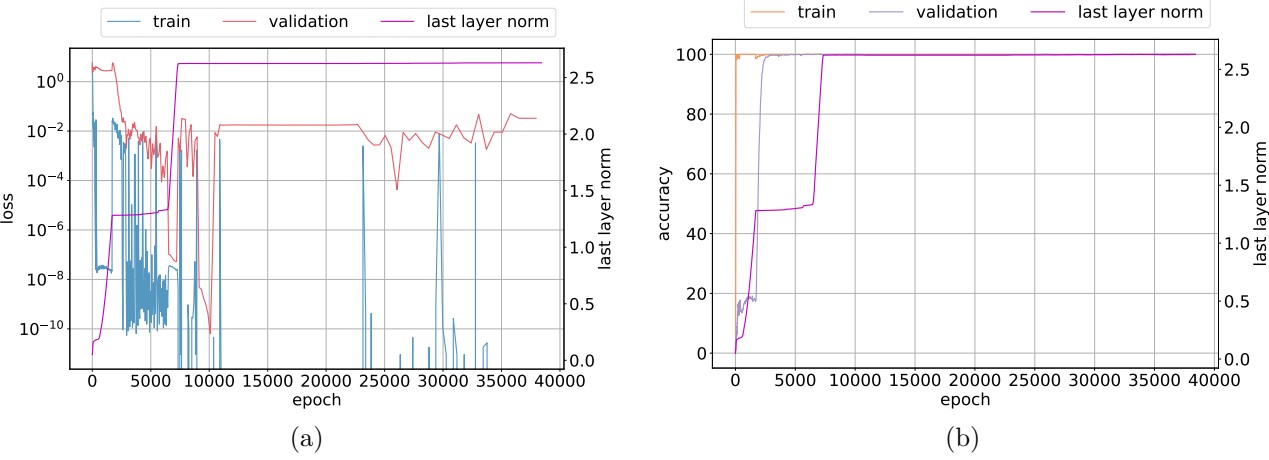

Figure 36: Division dataset with 70/30 train/validation split. Training and validation (a) loss and (b) accuracy.

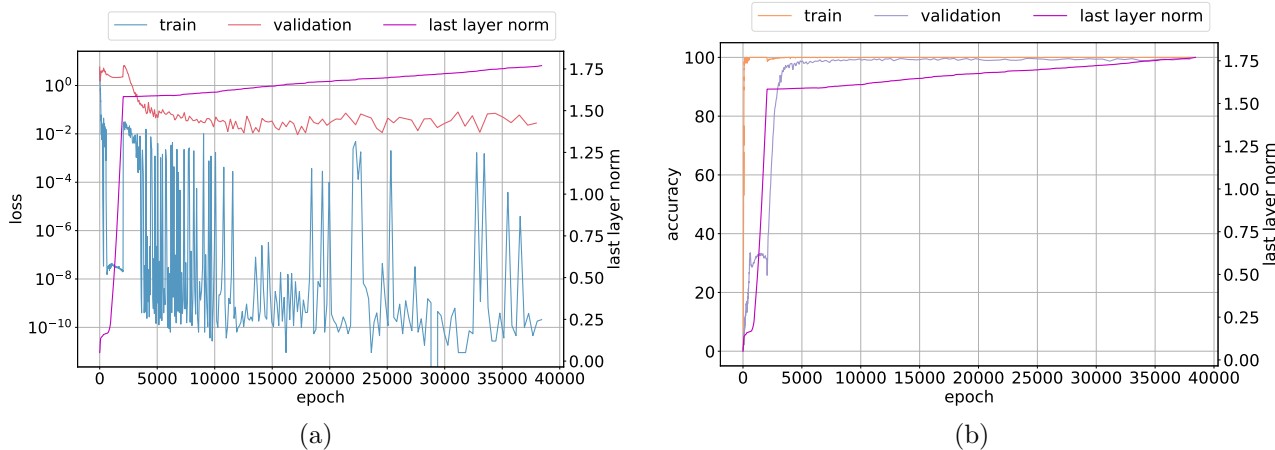

Figure 37: Even-add-odd-subtraction dataset with 70/30 train/validation split. Even-add-odd-subtraction operation is given by $[a + b \pmod{p}$ if $a$ is even, otherwise $a - b \pmod{p}]$ for $0 \leq a, b < p$. Training and validation (a) loss and (b) accuracy.

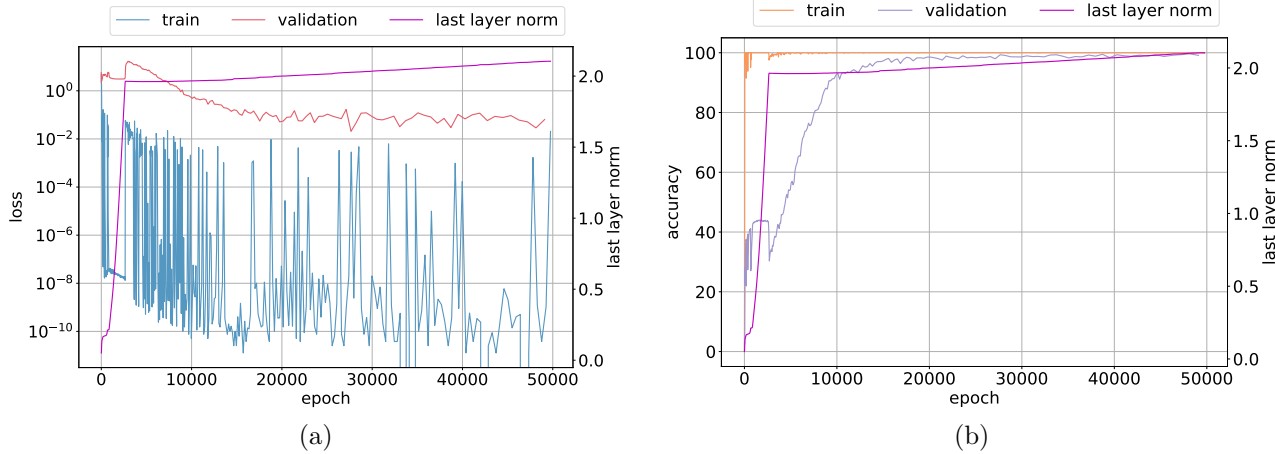

Figure 38: Multiplication dataset with 50/50 train/validation split. Training and validation (a) loss and (b) accuracy.

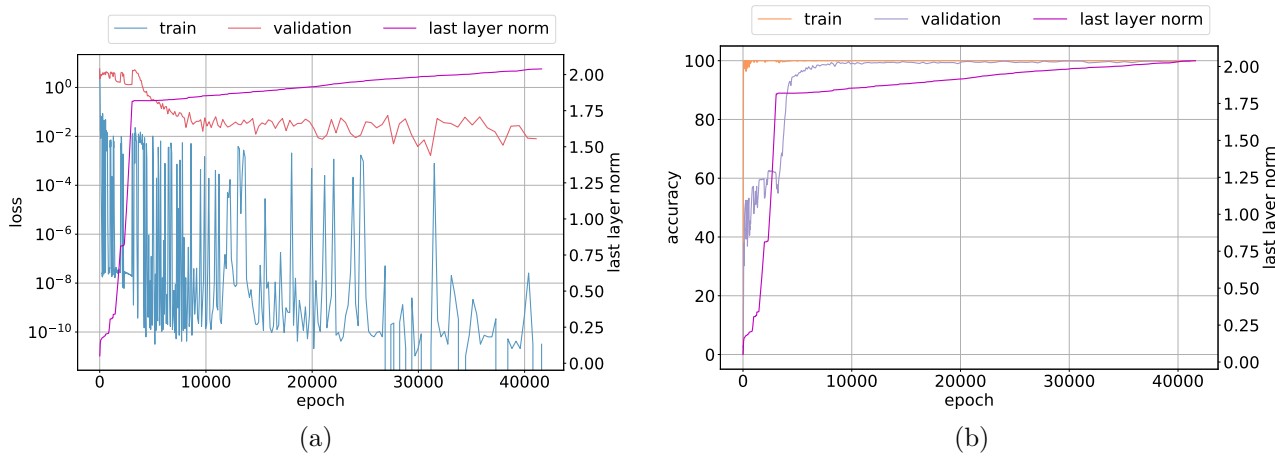

Figure 39: Multiplication dataset with 60/40 train/validation split. Training and validation (a) loss and (b) accuracy.

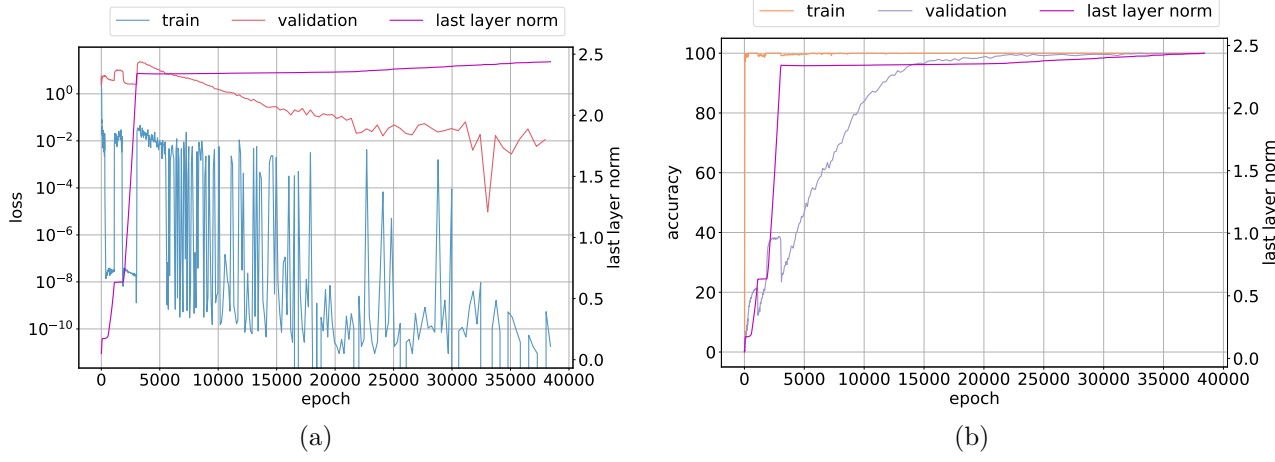

Figure 40: Squarepoly dataset with 70/30 train/validation split. Squarepoly operation is given by $a^2 + b \pmod{p}$ for $0 \leq a, b < p$. Training and validation (a) loss and (b) accuracy.

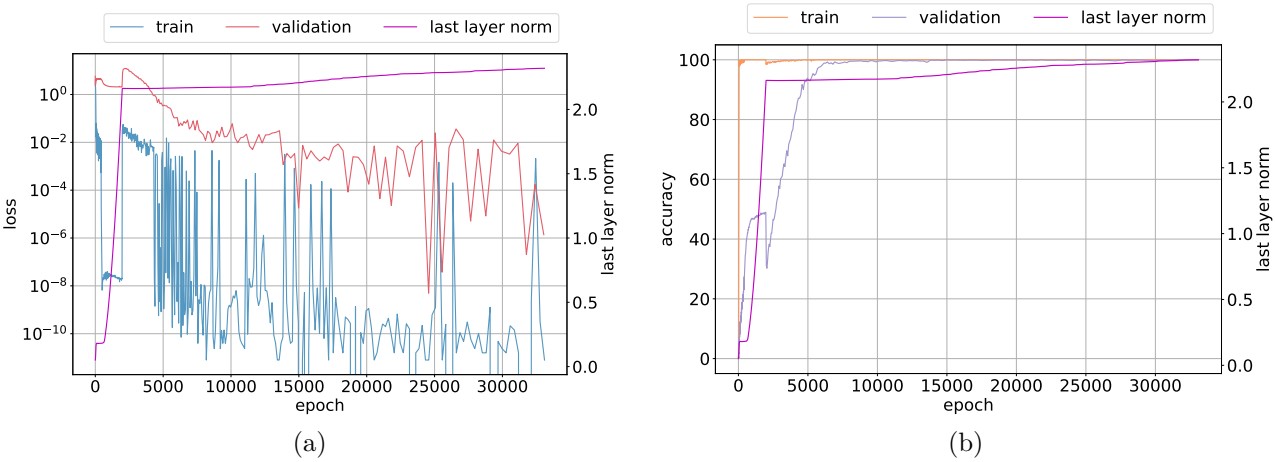

Figure 41: Squarepoly dataset with 80/20 train/validation split. Squarepoly operation is given by $a^2 + b \pmod{p}$ for $0 \le a, b < p$. Training and validation (a) loss and (b) accuracy.

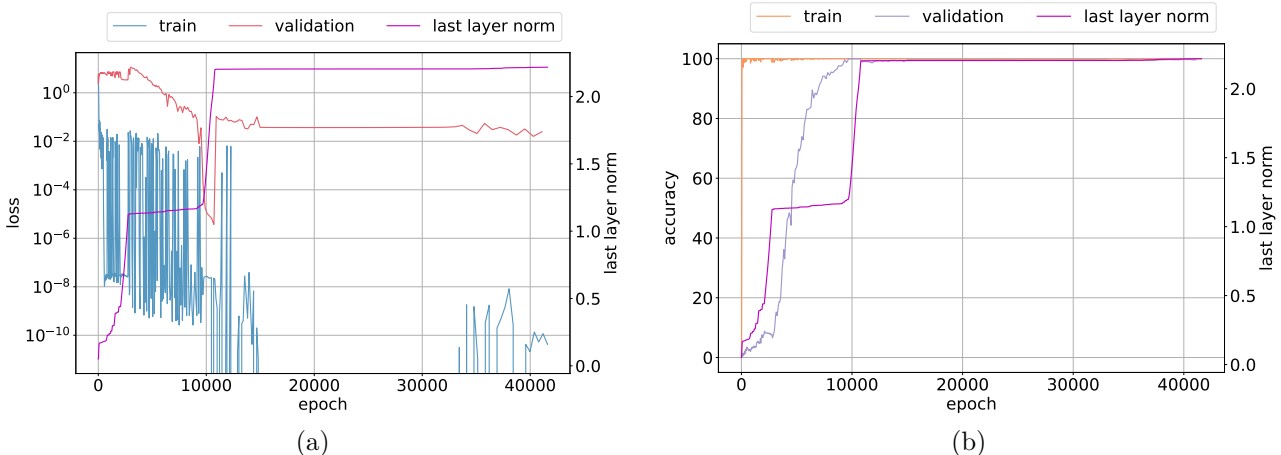

Figure 42: Subtraction dataset with 60/40 train/validation split. Training and validation (a) loss and (b) accuracy.

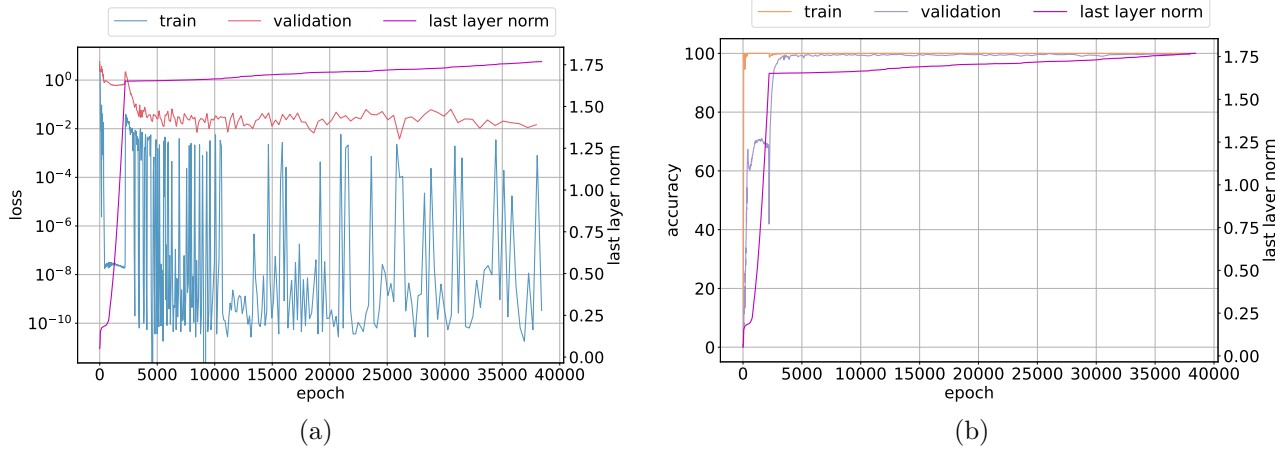

Figure 43: Subtraction dataset with 70/30 train/validation split. Training and validation (a) loss and (b) accuracy.

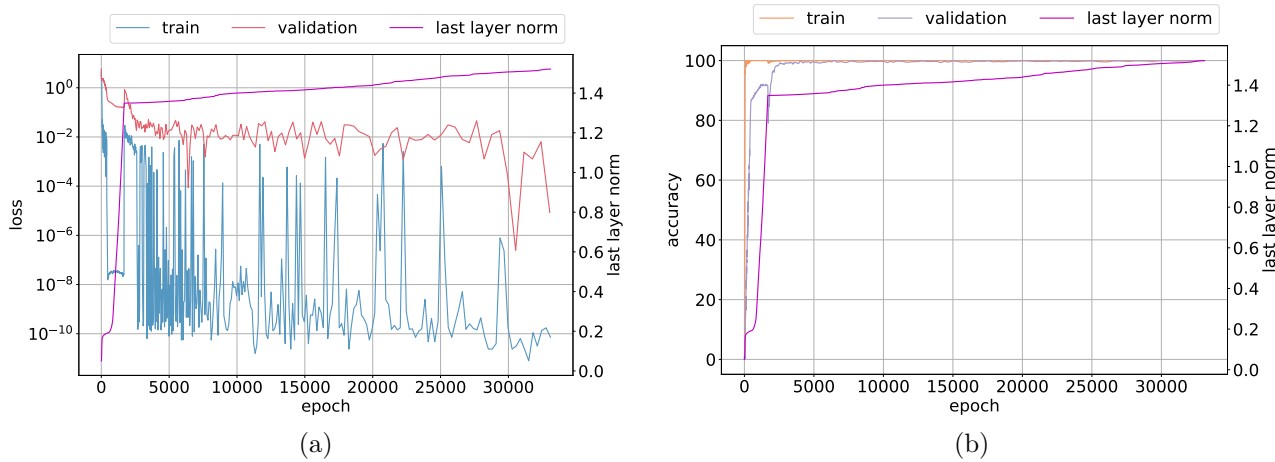

Figure 44: Subtraction dataset with 80/20 train/validation split. Training and validation (a) loss and (b) accuracy.

## C Controlling Instability Through Normalization and Norm Constraints

Training instability is the hallmark of the Slingshot Effect, yet as seen in previous sections, the Slingshot Effect typically results in improved performance, and Grokking. In this section, we explore whether it is possible to maintain stable training, without sacrificing performance. To this end, we explore how constraining and regularizing the weights of the network affect the Slingshot behaviour, and overall performance.

### C.1 Weight decay

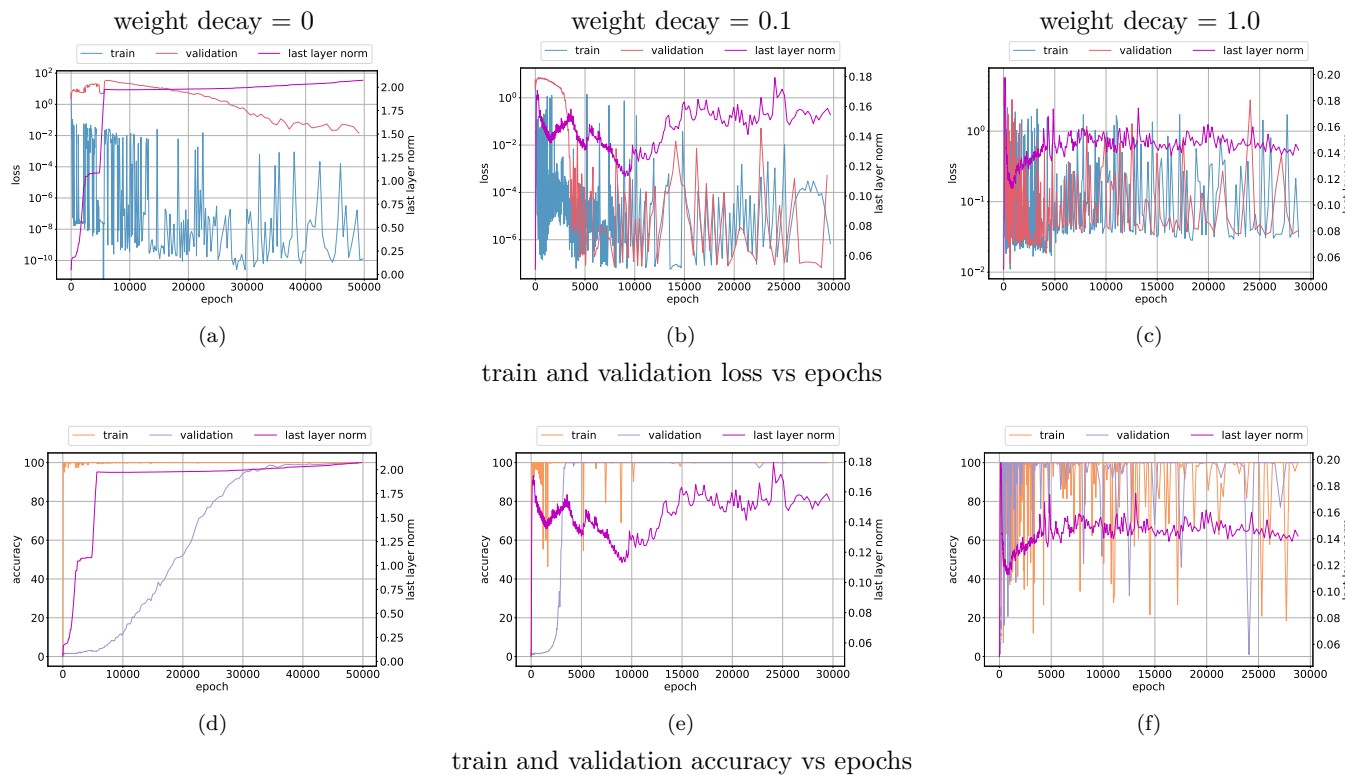

(a)  (b)  (c)

train and validation loss vs epochs

(d)  (e)  (f)

train and validation accuracy vs epochs

Figure 45: Division dataset: Norm behavior with different weight decay values. Training and validation loss vs epochs with weight decay (a) 0.0, (b) 0.1, (c) 1.0; Training and validation accuracy vs epochs shown in (d), (e) and (f). The evolution of classifier weight norm shows instability as increase in weight decay strength.

Weight decay is a commonly used regularization approach to improve the generalization performance of neural networks. Power et al. (2021) show that weight decay has the largest positive effect on alleviating Grokking. Weight decay naturally controls the size of the parameters and consequently their norm growth. We study the effect of weight decay on stability of training Transformers with Grokking datasets in this section. We use weight decay values from $0, 0.1, 0.2, 0.4, 0.6, 0.8 and 1.0$ with AdamW (Loshchilov & Hutter, 2017) optimizer. Figure 45 shows the results for division dataset. We observe from Figure 45 that as weight decay strength increases, both Slingshot Effects and Grokking phenomenon disappear with the model reaching high validation accuracy quickly as seen in Figure 45e and Figure 45f. However, we observe that the model experiences instability as can been seen with the loss plots in Figure 45b and Figure 45c or the accuracy plots in Figure 45e and Figure 45f. A similar trend is observed for addition and multiplication datasets in Figure 46 and Figure 47 respectively.

The results shown above indicate that Slingshot may not be the only way to achieve good generalization. Both Slingshot and weight decay prevent the norms from growing unbounded and achieve high validation accuracy as seen in plots described above. While weight decay shows different weight norm dynamics, this

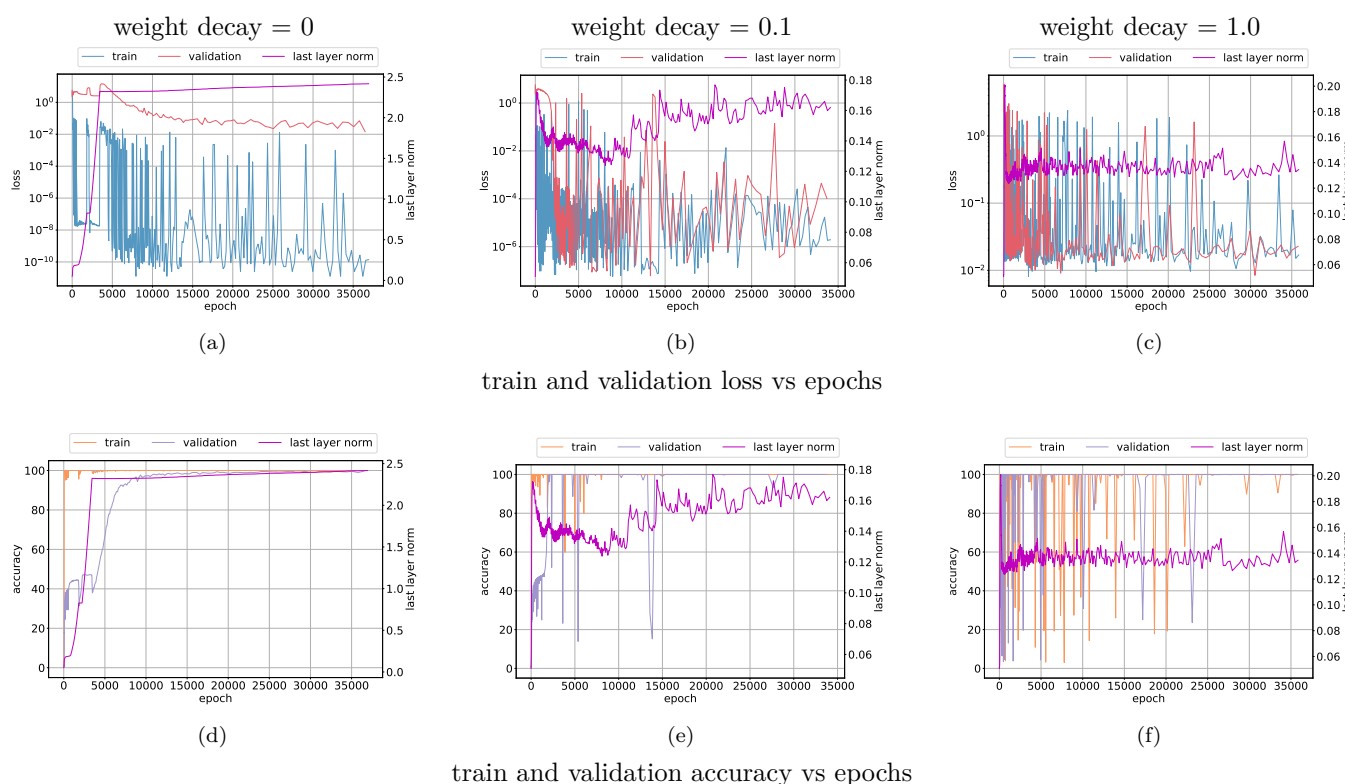

Figure 46: Addition dataset: Norm behavior with different weight decay values. Training and validation loss vs epochs with weight decay (a) 0.0, (b) 0.1, (c) 1.0; Training and validation accuracy vs epochs shown in (d), (e) and (f). The evolution of classifier weight norm shows instability as increase in weight decay strength.

regularization does not decrease training instability. These results suggest the need for alternative approaches to improve training stability.

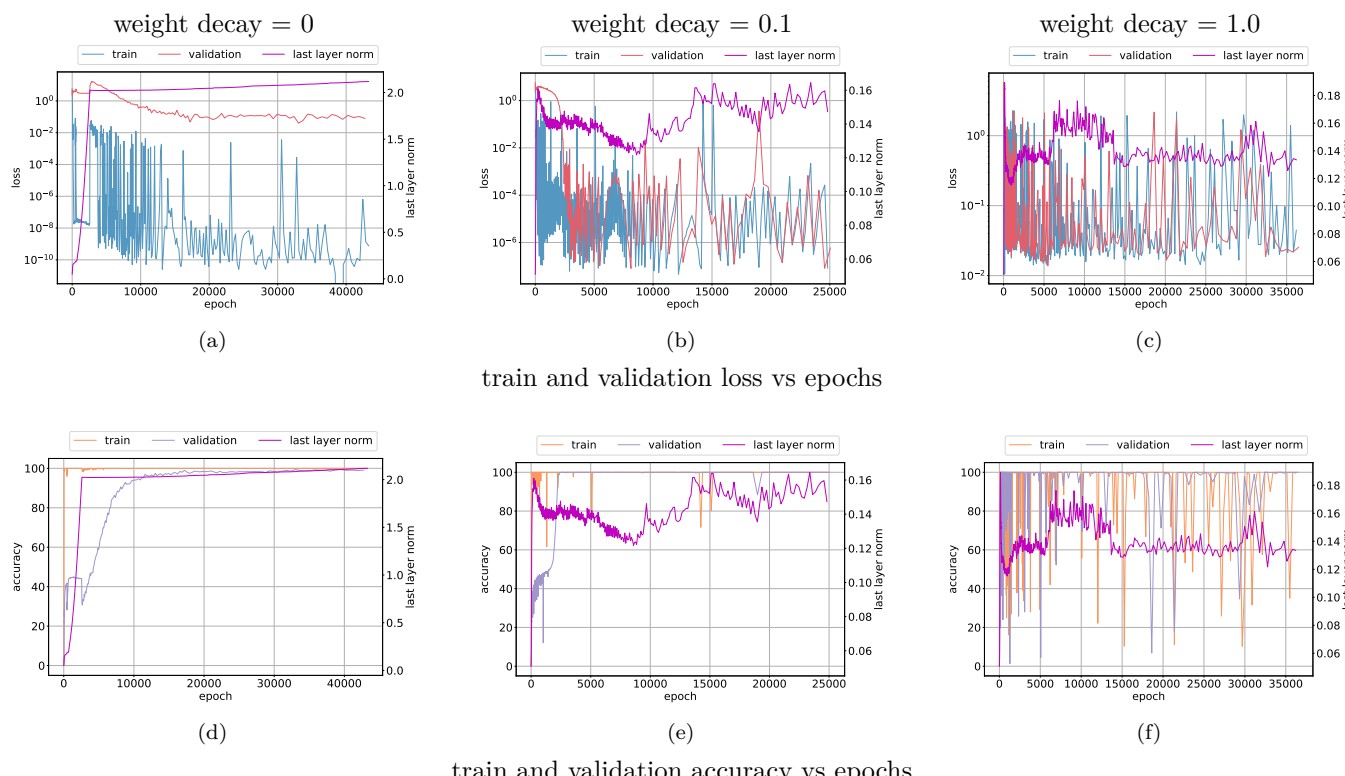

Figure 47: Multiplication dataset: Norm behavior with different weight decay values. Training and validation loss vs epochs with weight decay (a) 0.0, (b) 0.1, (c) 1.0; Training and validation accuracy vs epochs shown in (d), (e) and (f). The evolution of classifier weight norm shows instability as increase in weight decay strength.

### C.2 Features and parameter normalization

A second approach that we use to explicitly control weights and feature norm is by normalizing the features and weights via the following scheme: $w = \frac{w}{\|w\|}, f(x) = \frac{f(x)}{\|f(x)\|}$, where $w$ and $f(x)$ are the weights and inputs to the classification layer respectively, the norm used above is the $L_2$ norm, and $x$ is the input to the neural network. We take the cosine similarity of the normalized weights and features and divide this value by a temperature value that we treat as a hyperparameter in these experiments. The operation is given by: $y = \frac{w \cdot f(x)}{\tau}$ where $\tau$ represents the temperature hyperparameter. We use temperature values from $0.1, 0.25, 0.5, 0.75, 1.0$ for these experiments.

Figure 48 shows the results of Transformer training on division dataset described in Appendix B that is split evenly into train and validation sets. We observe that the model displays training instability evidenced by norm behavior and also loss behavior in Figure 48a at lower temperature values. We observe that $\tau = 0.25$ provides a good compromise between fitting training data while showing no training instability as seen in Figure 48b. This hyperparameter value also results in Grokking as validation accuracy improves late in training as can be seen from Figure 48e. These together suggest that bounding weights and features norm helps stabilize training without sacrificing training performance.

We validate the normalization scheme with two additional datasets namely multiplication and division from Appendix B. Figure 49 shows the results for training Transformers with multiplication dataset that is split evenly into train and validation sets. We observe from Figure 49 that a proper temperature value can stabilize training and with some tuning can provide a compromise between training stability and generalization. Specifically, $\tau = 0.25$ allows the model to fit the training data and reach almost perfect validation accuracy as seen from Figure 49b and Figure 49e.

Finally, we repeat the above experiments with subtraction dataset and show the results in Figure 50. This dataset shows that while a properly tuned temperature can help the model achieve almost perfect generalization, training instability shows up very late in optimization. This observation can be seen from Figure 50b and Figure 50d. This result suggests that more work remains to be done with understanding and stabilizing the training behavior of large neural networks.

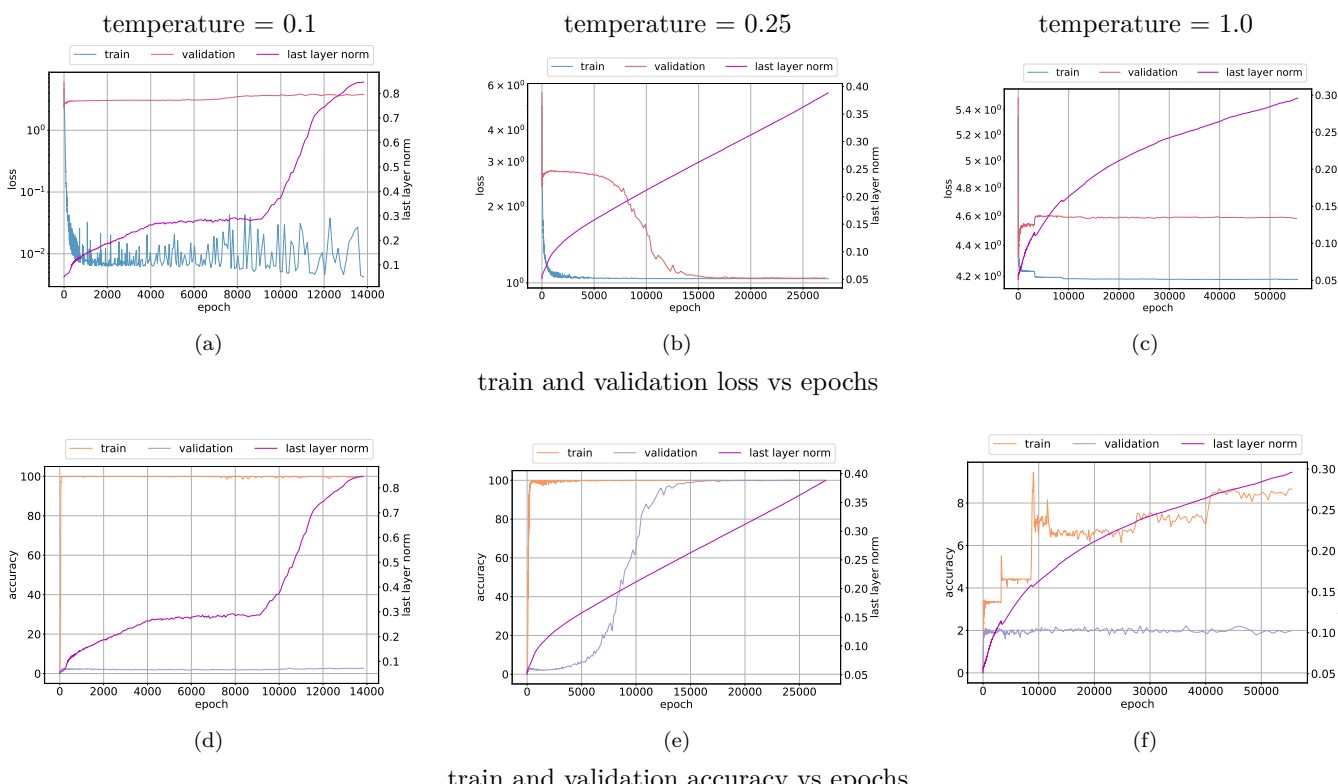

(a)          (b)          (c)

train and validation loss vs epochs

(d)          (e)          (f)

train and validation accuracy vs epochs

Figure 48: Division dataset: Features and parameters normalization. Observe that a smaller temperature allows the model to fit the data better but experiences training instability. Temperature = 0.25 allows the model to fit and achieve high validation accuracy without suffering training instability.

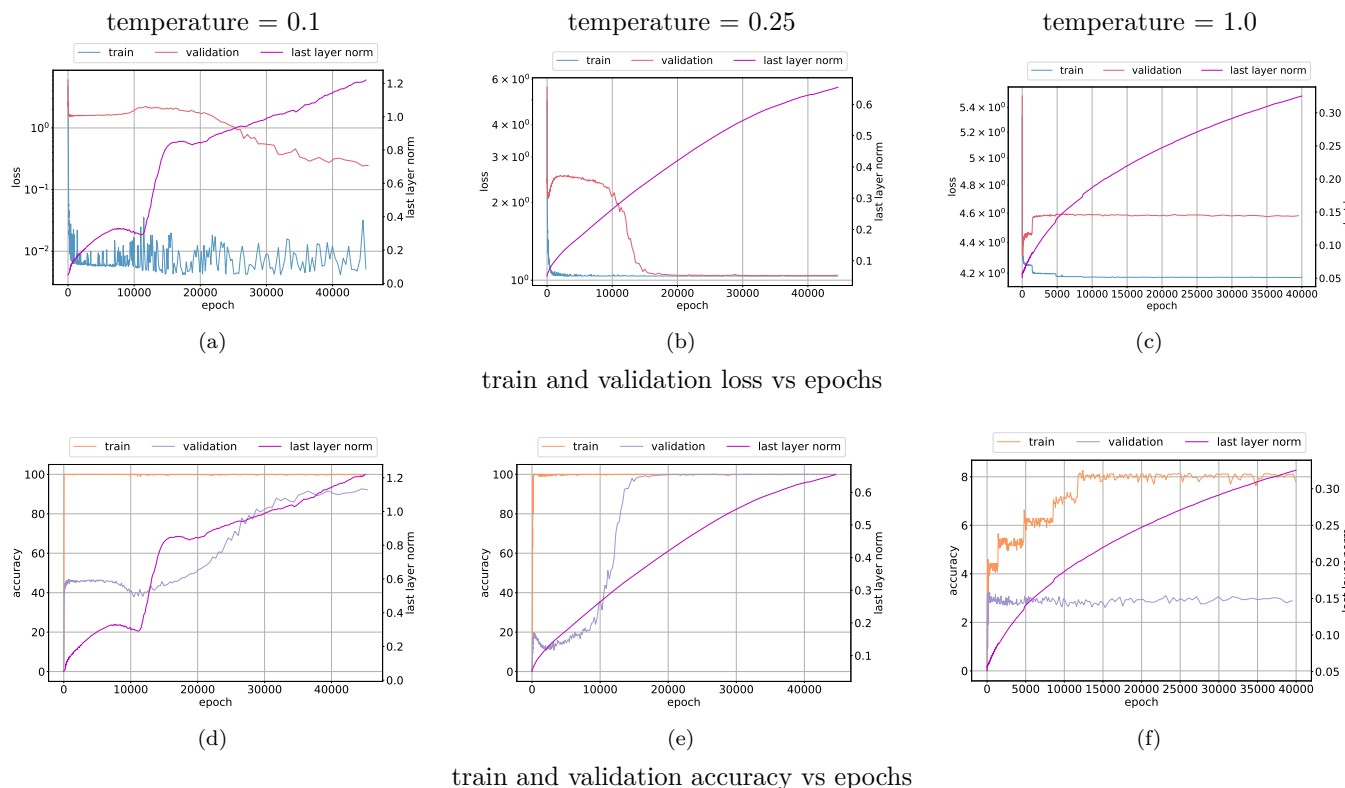

Figure 49: Multiplication dataset: Features and parameters normalization. Observe that a smaller temperature allows the model to fit the data better but experiences training instability. Temperature = 0.25 allows the model to fit and achieve high validation accuracy without suffering training instability.

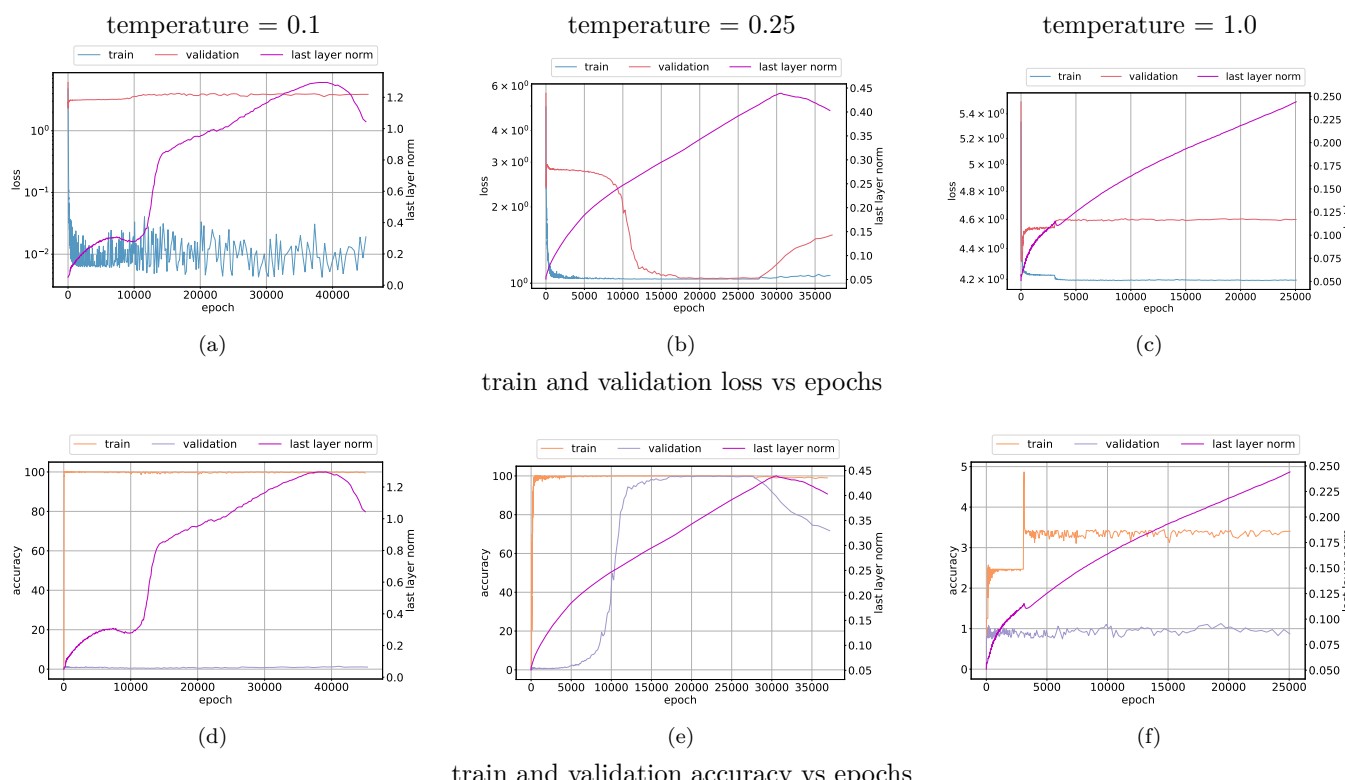

Figure 50: Subtraction dataset: Features and parameters normalization. Observe that a smaller temperature allows the model to fit the data better but experiences training instability. Temperature = 0.25 allows the model to fit and achieve high validation accuracy. However, we observe training instability as can seen with weight norm dynamics.

## D  Slingshot Effects and Linear Mode Connectivity

Recent works (Lubana et al., 2023; Juneja et al., 2023) have used mode connectivity analysis to explore the loss landscapes of neural networks. These works suggest the existence of multiple basins in the loss surface each with its own generalization behavior (Lubana et al., 2023; Juneja et al., 2023). Given the empirical observations we make with Slingshot Effects, one hypothesis to explore is that each Slingshot Effect leads the model to a different basin in the loss landscape that shows potentially differing generalization behavior. In this section, we explore whether Adam optimizer (Kingma & Ba, 2014) explores different basins in the loss landscape by plotting the loss along the linear path between different model checkpoints collected during optimization. We defer a thorough study of the above hypothesis to future work.

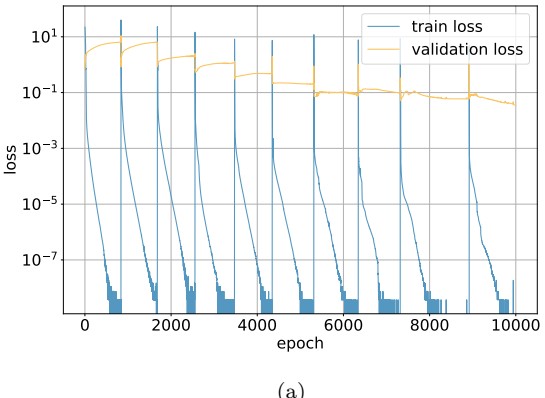
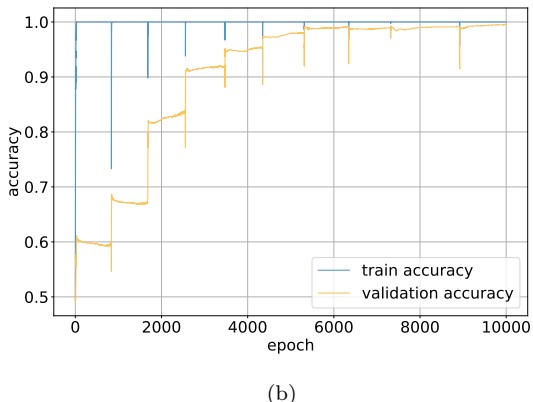

(a)                                                                (b)

Figure 51: Subset parity dataset: train and validation (a) loss and (b) accuracy.

We train a 3-layer MLP with hidden layer width 100 on a $(50, 3)$ subset-parity dataset described in Appendix A.6. The training dataset consists of 1000 samples while the test dataset consists of 8192 samples. We optimize the network with Adam and cross-entropy loss using a learning rate of 0.004, $\beta_1 = 0.9$, $\beta_2 = 0.999$, $\epsilon = 10^{-8}$ and batch size set to 32. The model is optimized with Adam (Kingma & Ba, 2014) for 10000 epochs. Figure 51 shows the train and validation loss and accuracy metrics for this experiment.

The linear mode connectivity (LMC) analysis is conducted by using two checkpoints as input and interpolating between the checkpoints to generate model weights. We choose two checkpoints that are equidistant (as measured in epochs) from a Slingshot Effect event or training loss spike. Specifically, we use two checkpoints that are 25 epochs on either slide of a Slingshot Effect as this value allows us to include periods of low train loss on either side of the loss spike. We tested other values for the number of epochs going as low as 10 epochs and found that this hyperparameter has a negligible effect on our analysis.

We linearly interpolate between the two checkpoints chosen using the method described above and use the interpolated weights to calculate train and validation metrics. Figure 52 shows an example LMC plot that includes the train loss calculated via interpolation (top-left). We observe that the LMC train loss shows a peak that suggests a loss barrier[4] that in turn suggests that the model checkpoints are in different basins. Figure 53 and Figure 54, captured at different points during training, show additional examples of LMC analysis that intuitively suggest that the model jumps between different loss basins.

---

[4]We borrow this terminology from (Lubana et al., 2023; Juneja et al., 2023)

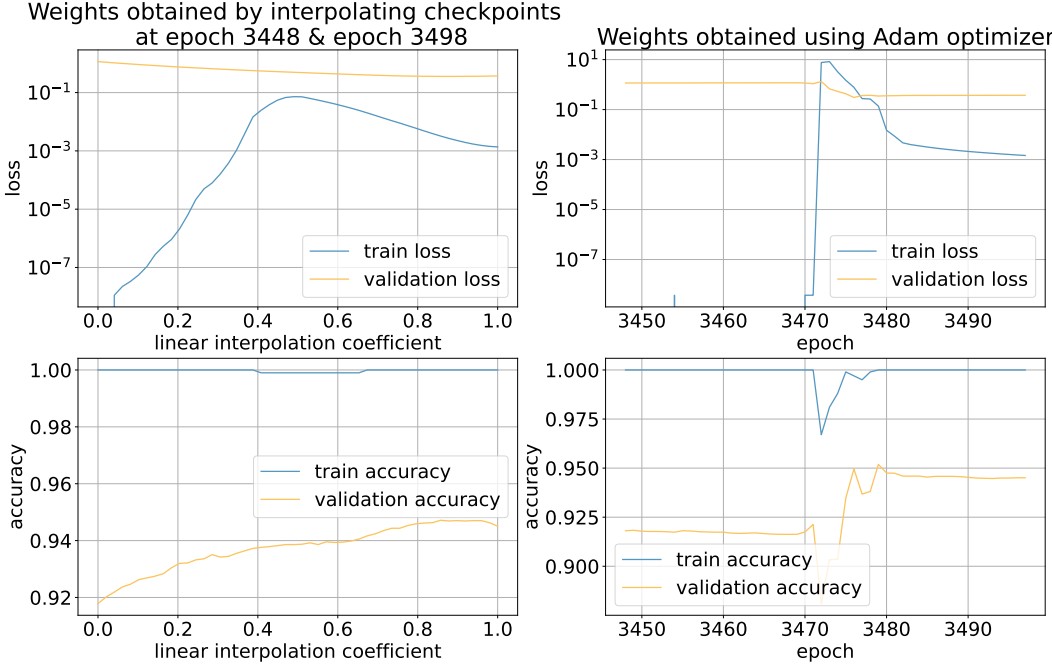

Figure 52: Train and validation metrics calculated with model weights generated via linear interpolation (left) and via Adam optimization (right).

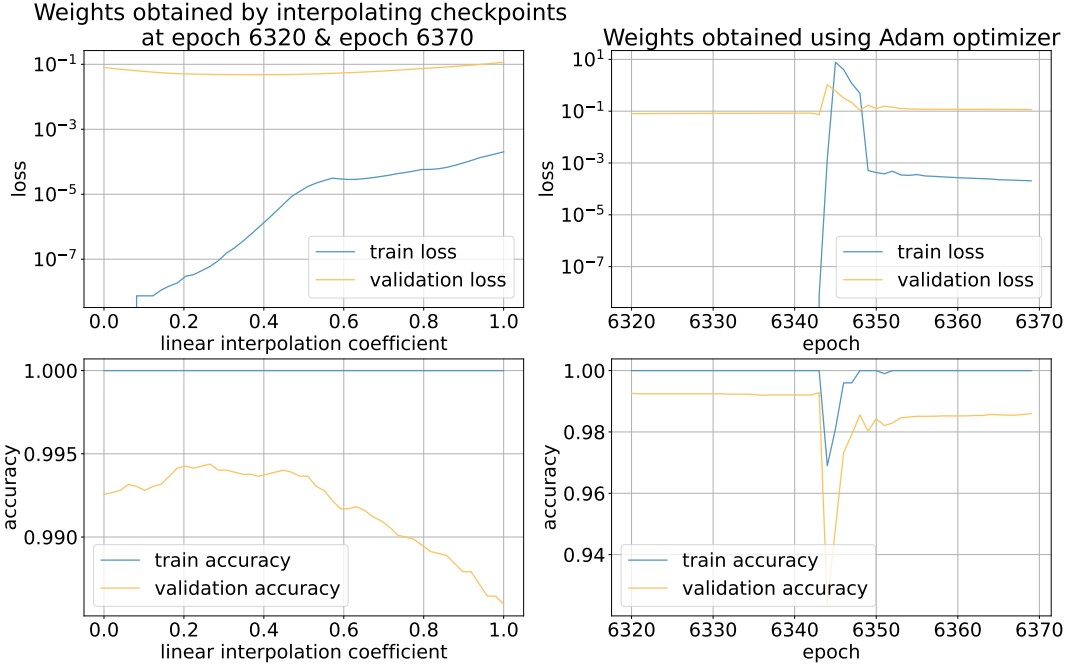

Figure 53: Train and validation metrics calculated with model weights generated via linear interpolation (left) and via Adam optimization (right).

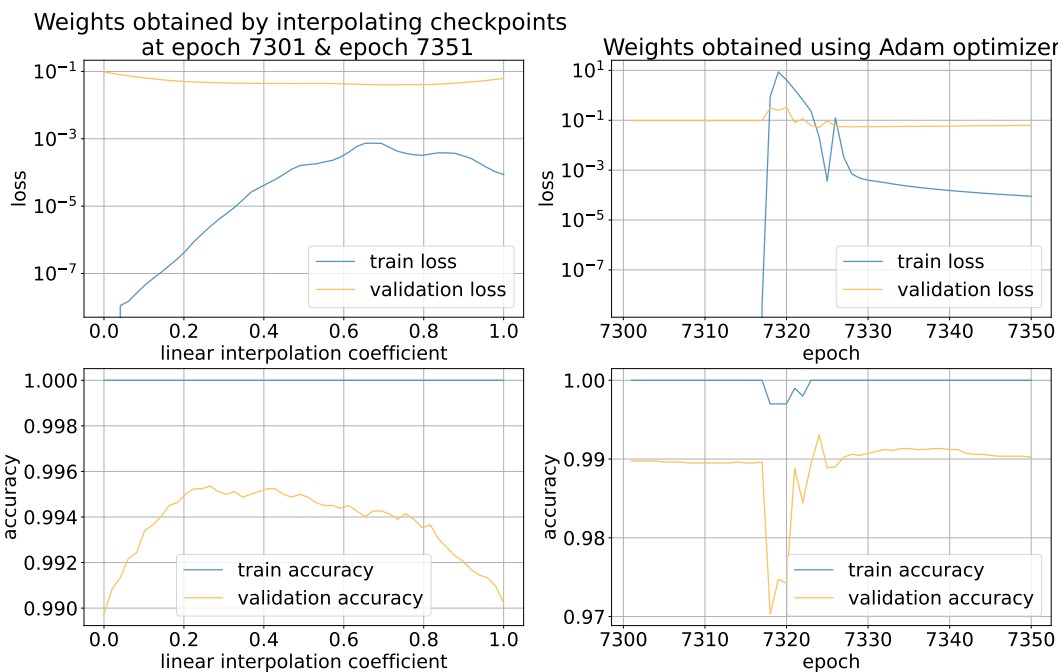

Figure 54: Train and validation metrics calculated with model weights generated via linear interpolation (left) and via Adam optimization (right).

