# OpenReview forum: "The Slingshot Effect: A Late-Stage Optimization Anomaly in Adaptive Gradient Methods"
_TMLR — Accepted by TMLR_

### Review · Reviewer_zYov · 2023-07-24

**Summary Of Contributions:**

The paper empirically verify a interesting pehonoon of Adam, called Slingshot effet, which stands for a repeating phase shifts between stable and unstable training regimes after the model overfits the training set.  As claimed by authors, slingshot is  a unique effect of Adam family. Further, as shown in Figure 6, the model seem to have better generalization power after the slingshot performance. As such, authors belive that the above discovery reveal an intriguing inductive bias of adaptive gradient optimizers.



On the algorithmic datasets by Power et al. , The authors also discuss the intersting relation between slingshot and  Grokking effect (Power et al., 2021) .




**Audience:**

Yes

**Claims And Evidence:**

Yes

**Requested Changes:**


**Major Questions:**

1. **Universial training phenomenon?**  I am not sure whether the Slingshot Effect is a universal phenomenon. Authors only verify this phenomenon on special toy data set by  (Power et al., 2021) and small data set like cifar 10. We are not sure whether the pheononm exists in other real language or vision tasks. Further,  we are not sure whether sliingshot is related to the spike effect in LLM as reported by Chowdhery et al., 2022, either.





2. **Relation with generalization power?**  The author claims that  "the model has better generalization power after the Slingshot phase". However, again, this claim is only verified on cifar10 and toy datasets. Without further evidence, the connection between Slingshot and generalization is quite weak, making the script less interesting than it claimed.



Remark: Although both questions are related to the limited experiments demonstrated in the script, question1 and 2 are two fundamentally different questions. The first focus on training,  and the 2nd focus on generalization.  The authors need to provide more evidence to support their claim on "discovering an important and universal training phenomenon" and "its relation to implicit bias of Adam/ generalization power"

**Other questions:**

1. The coloring in Figure 6 is confusing. Correct me if wrong: From my understanding ,   every column

(# samples)  should  have  three balls **with distinct color**, right?  But now many columns has three green balls.


2. Is the instability phenomenon due to the improper choice of beta2 in Adam? As proved in recent work [1], Adam might not work well if beta2 is not sufficiently large. I suggest the authors spent some effort tuning beta2 according to the guidance in [1], then see if the spiking phenomenon will be mitigated. For instance, simply try larger beta2 like 0.9995, 0.9999 to see if the spike disappears.

[1] adam can converge without any modification on update rules, Zhang et al., 2022


**Writing suggestion:**

The logic of the paper is a bit confusing. I suggest the author introduce the messages one by one as follows: 1. discover a new training phenomenon with spikes, and provide evidence. 2. Discuss the relation with generalization power, and provide evidence (the cifar 10 part). 3. Discuss the relation with Grokking, and provide evidence.

Currently, 12 and 3 are all mixed up without clear logical order. It is difficult to read for general readers.



**Strengths And Weaknesses:**

See below

---

> ### Author Response · Authors · 2023-10-31
> **Reply to Reviewer zYov Part 1**
>
> We thank the reviewer for their valuable time reviewing out manuscript and proving comments. Please see our responses below
>
> > Universial training phenomenon? I am not sure whether the Slingshot Effect is a universal phenomenon. Authors only verify this phenomenon on special toy data set by (Power et al., 2021) and small data set like cifar 10. We are not sure whether the pheononm exists in other real language or vision tasks.
>
> We do not make any claim about universality of Slingshot Effects. We view the Slingshot effect as an optimization anomaly and provide setups that are used in practice to study the optimization and generalization behavior of neural networks including algorithmic datasets, subset parity and CIFAR-10 data. We believe it is beyond the scope of the work to study these effects in larger practical models that include many other bells and whistles including regularizers that likely obscure the effect. Nevertheless, we believe that reporting this phenomenon is valuable to the community as it likely becomes apparent in cases far into the terminal phase of training where the training loss is very low.
>
>
> > Further, we are not sure whether sliingshot is related to the spike effect in LLM as reported by Chowdhery et al., 2022, either.
>
> Chowdhery et al. (2022) report loss spikes in the logit layer which is also noted by Wortsman et al. recently (Wortsman, 2023). Our observed Slingshot Effects also occurs in the last layer which is consistent with the observations made by Chowdhery et al. and Worstman et al.
>
>
> > Relation with generalization power? The author claims that "the model has better generalization power after the Slingshot phase". However, again, this claim is only verified on cifar10 and toy datasets. Without further evidence, the connection between Slingshot and generalization is quite weak, making the script less interesting than it claimed.
>
> We appreciate the reviewer’s point that that the range of settings used in the paper to gather empirical evidence about Slingshots and generalization appear to be limited. We address this concern by first noting that we view the Slingshot effect as an optimization anomaly and its precise connection to generalization is at this point unclear. Instead we report our observations that generalization occurs at the onset of Slingshot effects with algorithmic datasets used in Grokking — and strikingly also on subset parity, another important benchmark used to study the generalization behavior of neural networks (Barak 2022). We believe that these results are interesting to the machine learning research community as similar toy model setups have been used by Nanda et. al. (Nanda 2023), Liu et. al. (Liu 2022, Liu 2023) and references there-in to explain Grokking via mechanistic interpretability. Our work exposes an optimization artifact not seen in the above works as these works rely on weight decay while we point out to the existence of Slingshots in a setup with no weight decay that show interesting generalization patterns as reported by Power et. al. (Power 2022) and also observed in our experiments. We have been careful about not making any claims about behavior with larger models/setups and leave these as topics of future work. We are happy to edit the wording further to clarify the scope of our claims

---

> ### Author Response · Authors · 2023-10-31
> **Reply to Reviewer zYov Part 2**
>
> Other questions:
> > The coloring in Figure 6 is confusing. Correct me if wrong: From my understanding , every column
> (# samples) should have three balls with distinct color, right? But now many columns has three green balls.
>
> We apologize for the confusion. The dots in the figure need not necessarily be distinct. The colors are used to indicate whether the best test accuracy was observed after Slingshot Effects (green), before Slingshot Effects (blue) or no Slingshot Effects (red) were seen.
>
>
> > Is the instability phenomenon due to the improper choice of beta2 in Adam? As proved in recent work [1], Adam might not work well if beta2 is not sufficiently large. I suggest the authors spent some effort tuning beta2 according to the guidance in [1], then see if the spiking phenomenon will be mitigated. For instance, simply try larger beta2 like 0.9995, 0.9999 to see if the spike disappears.
>
> The reviewer raises an excellent question. We observed Slingshot Effects with default settings (suggested by the authors) used in many experiments reported in the paper and put forth a hypothesis that Slingshots can be controlled via \epsilon value in Adam and verified this hypothesis empirically. We have run more experiments by varying \beta_{2} as suggeted by the reviewer and can confirm the prescence of Slingshot Effects. The plots are included in the updated draft in Appendix B.3 and Figure 26.
>
>
> > The logic of the paper is a bit confusing. I suggest the author introduce the messages one by one as follows: 1. discover a new training phenomenon with spikes, and provide evidence. 2. Discuss the relation with generalization power, and provide evidence (the cifar 10 part). 3. Discuss the relation with Grokking, and provide evidence.
> Currently, 12 and 3 are all mixed up without clear logical order. It is difficult to read for general readers.
>
> We are grateful to the reviewer for carefully reading our draft and providing feedback to improve our presentation. We will update the draft in a subsequent revision by taking into account the feedback provided above as well as all other reviews to ensure our clear presentation.
>
> - [Barak 2022] https://arxiv.org/abs/2207.08799
> - [Chowdhery 2022] https://arxiv.org/abs/2204.02311
> - [Liu 2022] https://arxiv.org/abs/2205.10343
> - [Liu 2023] https://arxiv.org/abs/2210.01117
> - [Nanda 2023] https://arxiv.org/abs/2301.05217
> - [Wortsman 2023] https://arxiv.org/abs/2309.14322

---

### Review · Reviewer_aRAs · 2023-08-08

**Summary Of Contributions:**

This paper have shown an interesting phenomenon named the Slingshot effect during the optimization of deep neural networks using adaptive optimizers, specificially Adam (and RMSProp). The authors have conducted extensive experiments to show that this effect is predominant in different deep learning senarios, with models ranging from MLP to Transformers, and datasets from synthetic Gaussian data to real-life datasets such as CIFAR10. The authors have also shown an interesting connection between Grokking and the Slingshot Effect.

**Audience:**

Yes

**Claims And Evidence:**

Yes

**Requested Changes:**

Please refer to the weakness part. Specifically, I would consider accepting this paper if:

1. Adam is highlighted instead of adaptive optimization algorithms in general.
2. Theoretical understanding/intuition is provided. It does not have to be very hard. Even linear regression is fine.

**Strengths And Weaknesses:**

Strengths:
1. The paper is easy to follow and well-written.
2. The observed Slingshot effect is interesting and has potential impact in understanding Adam.
3. The authors have conducted numerous experiments to show how common the Slingshot effect is.

Weaknesses:

1. First of all, I think the title (and of course, many parts of the paper) should be changed because most of the experiments in this paper are solely concentrated on "Adam". RMSProp is just a special case of Adam with first-order momentum removed ($\beta_1 = 0$ in Adam). Therefore, claiming "a late-stage optimization anomaly in adaptive gradient methods" would be over-selling. "Adaptive Optimization Algorithms" is a term used to describe optimization algorithms that have the ability to change its stepsizes based on certain criterions during the optimization process. For example, AdaGrad, AMSGrad, AdaX, AdaBelief, and AdaBound are quite famous, just to name a few. (Please let me know if you need references for them) These works should not only be mentioned in the related works, but also conducted experiments with. Of course, it would be infeasible to conduct experiments with all of them, but some of them should be fine. The authors have also mentioned in page 6 that AdaGrad does not have the Slingshot effect, so claiming that such effect is universal is overselling.

2. The paper is completely an experimental and observational work, with no theories to explain the proposed effect. As I mention in Weakness 1, it would be infeasible to try all the adaptive optimization algorithms. Therefore, a theoretical study of the proposed Slingshot effect should be desirable. Given the current paper, I cannot tell the reasons behind the proposed Slingshot effect, why it happens for Adam and not for AdaGrad, and how to prevent it/reproduce it.

3. The authors have mentioned that in the experiments, weight decay is set to 0. Does weight decay prevent the proposed Slingshot effect? If so, I would doubt the importance of this effect since weight decay is commonly used in practice. Moreover, in page 6, the authors have mentioned AdamW. If weight decay is not used, AdamW is basically the same as Adam. Therefore I am quite confused by the usage of weight decay in this paper.

---

> ### Author Response · Authors · 2023-10-31
> **Reply to Reviewer aRAs Part 1**
>
> We thank the reviewer for carefully reading our paper and providing feedback aimed at improving the quality of our presentation. We find it encouraging that the reviewer found the paper easy to read and appreciates the opportunities the paper creates for improving our understanding of Adam optimizer.
>
>
> > First of all, I think the title (and of course, many parts of the paper) should be changed because most of the experiments in this paper are solely concentrated on "Adam". RMSProp is just a special case of Adam with first-order momentum removed ( in Adam). Therefore, claiming "a late-stage optimization anomaly in adaptive gradient methods" would be over-selling. "Adaptive Optimization Algorithms" is a term used to describe optimization algorithms that have the ability to change its stepsizes based on certain criterions during the optimization process. For example, AdaGrad, AMSGrad, AdaX, AdaBelief, and AdaBound are quite famous, just to name a few. (Please let me know if you need references for them) These works should not only be mentioned in the related works, but also conducted experiments with. Of course, it would be infeasible to conduct experiments with all of them, but some of them should be fine. The authors have also mentioned in page 6 that AdaGrad does not have the Slingshot effect, so claiming that such effect is universal is overselling.
>
> We thank the reviewer for bringing up this point. and agree with the reviewer to limit our claim to "Adam-family of Optimization methods". We have edited the paper to reflect this change and will ensure that any remaining text that are not consistent with our updated claim will be fixed in subsequent revisions.
> We would like to point out that the introduction of our paper states: "Finally, we focus on Adam~\citep{kingma2014adam} optimization method in the main paper, and relegate all experiments with additional optimizers to the appendix that suggest that our observations and conclusions hold for other methods under the Adam-family of optimizers.". In the Appendix, in addition to the RMSProp experiments, we included an ablation of Adam's hyperparameters (by running experiments over a range of $\beta$s in Appendix A.9). We are happy to edit our wording in the paper further to clarify the scope of our claims.
>
>
> > For example, AdaGrad, AMSGrad, AdaX, AdaBelief, and AdaBound are quite famous, just to name a few. (Please let me know if you need references for them) These works should not only be mentioned in the related works, but also conducted experiments with. Of course, it would be infeasible to conduct experiments with all of them, but some of them should be fine.
>
> We thank the reviewer for the above algorithms. We have added references to the methods mentioned above in our paper. We are happy to further edit our claims to ensure that the claims correctly reflect the evidence provided in the paper.
>
>
> > The paper is completely an experimental and observational work, with no theories to explain the proposed effect. As I mention in Weakness 1, it would be infeasible to try all the adaptive optimization algorithms. Therefore, a theoretical study of the proposed Slingshot effect should be desirable. Given the current paper, I cannot tell the reasons behind the proposed Slingshot effect, why it happens for Adam and not for AdaGrad, and how to prevent it/reproduce it.
>
> This is a valid point raised by the reviewer. We view the Slingshot effect as an optimization anomaly, but make no attempt to precisely characterize its connections to generalization. However, due to the generality of the phenomena and the wide-spread use of Adam and Adam-like adaptive optimizers, we feel our paper would provide a valuable setting to study training instabilities that others have reported in large-scale settings [Chowdhery et al.]. Finally, we argue that providing adequate explanations of interesting empirical phenomena should not pose a barrier to publication in the field, which could result in limiting exposure to important findings. Instead, we believe that the findings may spur the community to develop a deeper understanding of tools that are commonly used to optimize large scale models.

---

> ### Author Response · Authors · 2023-10-31
> **Reply to Reviewer aRAs Part 2**
>
> > The authors have mentioned that in the experiments, weight decay is set to 0. Does weight decay prevent the proposed Slingshot effect? If so, I would doubt the importance of this effect since weight decay is commonly used in practice. Moreover, in page 6, the authors have mentioned AdamW. If weight decay is not used, AdamW is basically the same as Adam. Therefore I am quite confused by the usage of weight decay in this paper.
>
> The reviewer is indeed correct that AdamW with weight decay set to 0 is Adam. We have updated our paper to state that we use Adam in our experiments where we observe Slingshot Effects. We apologize for the confusion caused here.
> We would like to point out to the reviewer that we do have experiments with AdamW and non-zero weight decay in the Appendix. In Appendix C.1, we explore whether the training instabilities which is a hallmark of Slingshot Effects can be controlled via weight decay. We explore weight decay as our analysis exposes a built-in norm control behavior that accompanies Slingshot Effects. Our experimental results suggest that  using weight decay can remove Slingshot Effects but not prevent overall training instability.
>
> As to the importance of our work, we believe that these results are interesting to the machine learning research community as similar toy model setups have been used by Nanda et. al. (Nanda 2023), Liu et. al. (Liu 2022, Liu 2023) and references there-in to explain Grokking via mechanistic interpretability. These papers are considered as important references by researchers that study Grokking. These references use weight decay in their study while we point out to the existence of Slingshots in a setup with no weight decay that show interesting generalization patterns as reported by Power et. al. (Power 2022) and also observed in our experiments. We believe that our work is still of interest to researchers that study Grokking. Additionally, our work may provide an interesting small-scale setup similar to the work of Wortsman et. al. (Worstman 2023) to study instabilities observed while training neural networks.
>
> - [Barak 2022] https://arxiv.org/abs/2207.08799
> - [Chowdhery 2022] https://arxiv.org/abs/2204.02311
> - [Liu 2022] https://arxiv.org/abs/2205.10343
> - [Liu 2023] https://arxiv.org/abs/2210.01117
> - [Nanda 2023] https://arxiv.org/abs/2301.05217
> - [Wortsman 2023] https://arxiv.org/abs/2309.14322

---

### Review · Reviewer_5SKJ · 2023-08-18

**Summary Of Contributions:**

This paper describes a slingshot effect that occurs during the terminal phase of training models to minimize the unregularized cross entropy loss using adaptive optimizers. During the terminal phase of training, they identify that the training loss undergoes periodic spikes, which occur simultaneously with an explosion in the norm of the model. Following the explosion, the norm growth is curtailed and the training loss is minimized until the next "slingshot" occurs. Each subsequent slingshot also seems to be accompanied by an improvement in the validation loss/accuracy, which is reminiscent of the "grokking" phenomenon where changes in the validation loss can occur long after the training loss has been minimized.

The authors attempt to provide explanations for the emergence of the slingshot phenomenon by showing that variations in the stability parameter $\epsilon$ of an adaptive gradient algorithm can control the emergence/disappearance of the slingshot phenomenon.

The authors also claim that the slingshot effect is indicative of better generalization and compare models that exhibit this effect.

**Audience:**

Yes

**Claims And Evidence:**

Yes

**Requested Changes:**

1. Establish the slingshot effect in terms of whether adaptive optimizers progressively explore different basins in the loss landscape by plotting the loss along the linear path between different models.

2. How do the basins differ in generalization? Are there differences in the features learned at different stages?

**Strengths And Weaknesses:**

Extensive experimentation with different choices of architecture, optimizer, and dataset in deep learning has been an important driver of progress in our understanding. One of the contributions of empirical research in understanding deep learning is to highlight phenomena that are unexplained by existing theories and models. In this paper the authors are able to identify the situations in which the slingshot effect emerges and show that varying the $\epsilon$ parameter controls its emergence. The authors could however incorporate some theoretical insights to sharpen their description of the slingshot effect and isolate the exact anomaly in optimization with adaptive gradient methods.

1. **Optimization observations:** Gradient based minimization (SGD/GD/adaptive gradient methods) using the cross-entropy loss should naturally increase the norm of models. This is because the minimizers of the unregularized cross entropy loss are at $\infty$, and a simple model of the dynamics of SGD [1, Lemmas 1 and 2] also predicts the norm growth. Hence the dynamics of "norm curtailment" are not surprising.

The observation that is actually interesting (in my opinion) is the fact that adaptive gradient methods display sharp spikes in the training loss and norm. A hypothesis that could explain these observations is that in these large model/small dataset setups, adaptive gradient methods jump from one local/global minimum basin to another. This could indicate two regimes of operation for adaptive gradient methods - a) where the gradients are large (relative to $\epsilon$) and SGD type dynamics dominate to find the local minimum of a basin and b) where the gradients are small (relative to $\epsilon$) and the adaptive methods are pushed out of the local basin that they are currently in. More careful sharpness measurements could help uncover whether this is the case. I will note here that the observations about feature change (Fig 2) are interesting and could be an avenue for further theoretical/empirical investigation.

2. **Generalization observations:** The authors claim that the slingshot effect is accompanied by improved generalization, and while they are careful to not claim that this is a causal link, they do not posit any mechanisms for how this improvement in generalization could be happening. Connecting back to the previous observation about different basins in the loss landscape, one mechanism for improving generalization is that the different basins have different generalization properties (similar to [2,3]). Exploring this hypothesis would motivate the slingshot effect better. The current exploration of connections to generalization (Figures 5,6) do not have enough models in which the slingshot effect is absent in order to draw effective conclusions.


My broad concern about this paper is that as it is framed currently, the slingshot effect could be explained entirely in terms of jumping between different basins around local minima with different generalization capabilities, and this hypothesis is under-explored in favor of tracking norm dynamics, which are reasonably well understood and will not be explanatory for generalization.

[1] Poggio, Tomaso, Andrzej Banburski, and Qianli Liao. "Theoretical issues in deep networks." Proceedings of the National Academy of Sciences 117.48 (2020): 30039-30045.

[2] Juneja, Jeevesh, Rachit Bansal, Kyunghyun Cho, João Sedoc, and Naomi Saphra. "Linear connectivity reveals generalization strategies." arXiv:2205.12411, ICLR 2023

[3] Lubana, Ekdeep Singh, Eric J. Bigelow, Robert P. Dick, David Krueger, and Hidenori Tanaka. "Mechanistic mode connectivity." In International Conference on Machine Learning, pp. 22965-23004. PMLR, 2023.

---

> ### Author Response · Authors · 2023-10-31
> **Reply to Reviewer 5SKJ**
>
> We thank the Reviewer for carefully reading our paper and for an accurate and crisp summary of our work. We find it very encouraging that the viewer found our experiments extensive and appreciates the work we put into identify situations where slingshots occur.
>
>
> > Optimization observations: Gradient based minimization (SGD/GD/adaptive gradient methods) using the cross-entropy loss should naturally increase the norm of models. This is because the minimizers of the unregularized cross entropy loss are at , and a simple model of the dynamics of SGD [1, Lemmas 1 and 2] also predicts the norm growth. Hence the dynamics of "norm curtailment" are not surprising.
>
>
> We agree with the reviewer that the lemmas in [1] suggest norm growth. However, unless we are completely mistaken we believe that [1] suggest that the norm continues to grow (for all layers) monotonically while we empirically observe “norm curtailment” followed by renewed norm growth that coincides with Slingshot Effects. We believe this effect is a surprise, not documented or explained by previous literature, that should be of interest to the machine learning research community, especially since it appears to be a property specific to adaptive optimizers that are common in current practice.
>
>
> >The observation that is actually interesting (in my opinion) is the fact that adaptive gradient methods display sharp spikes in the training loss and norm. A hypothesis that could explain these observations is that in these large model/small dataset setups, adaptive gradient methods jump from one local/global minimum basin to another. This could indicate two regimes of operation for adaptive gradient methods - a) where the gradients are large (relative to ) and SGD type dynamics dominate to find the local minimum of a basin and b) where the gradients are small (relative to ) and the adaptive methods are pushed out of the local basin that they are currently in. More careful sharpness measurements could help uncover whether this is the case. I will note here that the observations about feature change (Fig 2) are interesting and could be an avenue for further theoretical/empirical investigation.
>
> >Generalization observations: The authors claim that the slingshot effect is accompanied by improved generalization, and while they are careful to not claim that this is a causal link, they do not posit any mechanisms for how this improvement in generalization could be happening. Connecting back to the previous observation about different basins in the loss landscape, one mechanism for improving generalization is that the different basins have different generalization properties (similar to [2,3]). Exploring this hypothesis would motivate the slingshot effect better. The current exploration of connections to generalization (Figures 5,6) do not have enough models in which the slingshot effect is absent in order to draw effective conclusions.
>
> > My broad concern about this paper is that as it is framed currently, the slingshot effect could be explained entirely in terms of jumping between different basins around local minima with different generalization capabilities, and this hypothesis is under-explored in favor of tracking norm dynamics, which are reasonably well understood and will not be explanatory for generalization.
>
> The reviewer raises a valid point above for avenues of further investigation be it theoretical or empirical work. We view the Slingshot effect as an optimization anomaly, and its precise connection to generalization is at this point unclear — yet curiously in the various settings in which we observed slingshot effects, these consistently resulted in improved generalization, indicating there is a connection for future work to try to characterize — the most compelling case being that of parity, where for each slingshot cycle a corresponding consistent generalization improvement was seen . However, due to the generality of the phenomena and the popularity of adaptive optimizers, we feel our paper would function as a valuable resource for studying training instabilities and their impact on generalization in a controlled manner (along the lines of (Wortsman 2023)). Finally, we submit that our primary focus in this paper is to provide several setups that cover various datasets including algorithmic datasets, subset parity dataset and vision (CIFAR-10) that demonstrate Slingshot Effects and leave a full exploration of the connections between Slingshot Effects and generalization to follow up work.
>
> [Wortsman 2023] https://arxiv.org/abs/2309.14322

---

### Author Response · Authors · 2023-10-31
**General Response**

Dear Reviewers and AE,

We once again thank you for granting an extension to rebuttal deadline. We have updated the draft of our paper in response to comments and concerns raised by the reviewers and uploaded it as a PDF file. We have added the following:
- Narrowed the focus of our paper to cover Adam or Adam-family of optimization algorithms where Adam-family of algorithms more or less cover the update rules of vanilla Adam (Kingma and Ba). Varying \beta values allows us to include RMSProp as well

- Added a new experiments that including tuning \beta_{2} Adam hypeparameter as suggested by  Reviewer zYov

- Several edits in response to  comments made by reviewers below. We have responded to each comment individually below

We look forward to hearing your feedback about our updated work.

---

### Decision · Action_Editor_ubJq · 2024-01-23

**Recommendation:** Accept with minor revision

**Comment:**

The claims made in the paper are well supported by extensive experiments and two of the reviewers have recommended acceptance after the author response. Reviewer aRAs is learning towards reject -- they acknowledge that the paper identifies an interesting phenomenon but are concerned about the lack of any theoretical understanding about it in the paper. While the point on the lack of theoretical understanding is valid, the paper does acknowledge this under the limitations section. Since the paper supports all the claims with extensive experiments as acknowledged by the reviewers and the slingshot phenomenon itself is deemed as interesting by all three reviewers, AE thinks the paper meets the [TMLR criteria for acceptance](https://www.jmlr.org/tmlr/acceptance-criteria.html).

Minor change: Authors should add the experiment suggested by the Reviewer 5SKJ on plotting the loss along the linear path between different models to provide intuition to the readers from the perspective of jumping between different loss basins.

**Audience:**

The slingshot phenomenon where the training loss and layer layer weight norm undergo cycles of spikes followed by decay / stabilization for Adam like optimizers is going to be of interest to the community, particularly researchers studying the interplay between memorization and generalization (eg, grokking). However the AE believes the findings in the paper are going to be more interesting from a theoretical perspective, since slingshot instabilities can be avoided in practice by using mechanisms that control the norm growths (such as weight decay or normalization).

**Claims And Evidence:**

The paper identifies a phenomenon in Adam family of optimizers without weight decay, called as Slingshot effect, that occurs as an anomaly during the terminal phase of training where the training loss undergoes cycles of sudden spikes and decay. This is accompanied by the spikes in norm of the last layer weights followed by norm stabilization. It is also connected to the grokking phenomenon identified in an earlier work where validation loss can decrease long after the training loss has converged. Authors also clarify that it is possible to reach comparable generalization performance without letting the model go through instabilities of slingshots by adding regularization such as weight decay.

All reviewers have acknowledged the extensive experiments in the paper in support of the claims about slingshot anomaly and about its connection to grokking (under no weight decay). Authors have also addressed the concern about scope of the paper by clarifying in the paper that results are for Adam family of optimizers and do not apply for all adaptive optimizers.